**Coupled eco-hydrology and biogeochemistry algorithms enable simulation of water table**
**depth effects on boreal peatland net CO$_2$ exchange**
Mohammad Mezbahuddin*[1,2], Robert F. Grant[2], and Lawrence B. Flanagan[3]
[1]*Environmental Stewardship Branch, Alberta Agriculture and Forestry, Edmonton, AB, Canada*
[2]*Department of Renewable Resources, University of Alberta, Edmonton, AB, Canada*
[3]*Department of Biological Sciences, University of Lethbridge, AB, Canada*
*corresponding author. Email addresses: symon.mezbahuddin@gov.ab.ca,
mezbahud@ualberta.ca.

**Abstract**

Water table depth (WTD) effects on net ecosystem $CO_2$ exchange of boreal peatlands are largely mediated by hydrological effects on peat biogeochemistry, and eco-physiology of peatland vegetation. Lack of representation of these effects in carbon models currently limits our predictive capacity for changes in boreal peatland carbon deposits under potential future drier and warmer climates. We examined whether a process-level coupling of a prognostic WTD with 1) oxygen transport which controls energy yields from microbial and root oxidation-reduction reactions, and 2) vascular and non-vascular plant water relations could explain mechanisms that control variations in net $CO_2$ exchange of a boreal fen under contrasting WTD conditions i.e. shallow vs. deep WTD. This coupling of eco-hydrology and biogeochemistry algorithms in a process-based ecosystem model *ecosys* was tested against net ecosystem $CO_2$ exchange measurements in a Western Canadian boreal fen peatland over a period of drier weather driven gradual WTD drawdown. A May-October WTD drawdown of ~0.25 m from 2004 to 2009 hastened oxygen transport to microbial and root surfaces, enabling greater microbial and root energy yields, and peat and litter decomposition, which raised modelled ecosystem respiration ($R_e$) by 0.26 µmol $CO_2$ $m^{-2}$ $s^{-1}$ per 0.1 m of WTD drawdown. It also augmented nutrient mineralization, and hence root nutrient availability and uptake, which resulted in improved leaf nutrient (nitrogen) status that facilitated carboxylation, and raised modelled vascular gross primary productivity (GPP) and plant growth. The increase in modelled vascular GPP exceeded declines in modelled non-vascular (moss) GPP due to greater shading from increased vascular plant growth, and moss drying from near surface peat desiccation, thereby causing a net increase in modelled growing season GPP by 0.39 µmol $CO_2$ $m^{-2}$ $s^{-1}$ per 0.1 m of WTD drawdown. Similar increases in GPP and $R_e$ left no significant WTD effects on modelled seasonal and

interannual variations in net ecosystem productivity (NEP). These modelled trends were
corroborated well by eddy covariance measured hourly net $CO_2$ fluxes (modelled vs. measured:
$R^2$~0.8, slopes~1±0.1, intercepts~0.05 μmol m$^{-2}$ s$^{-1}$), hourly measured automated chamber net
$CO_2$ fluxes (modelled vs. measured: $R^2$~0.7, slopes~1±0.1, intercepts~0.4 μmol m$^{-2}$ s$^{-1}$), and
other biometric and laboratory measurements. Modelled drainage as an analog for WTD
drawdown induced by climate change driven drying showed that this boreal peatland would
switch from a large carbon sink (NEP~160 g C m$^{-2}$ yr$^{-1}$) to carbon neutrality (NEP~10 g C m$^{-2}$
yr$^{-1}$) should water table deepen by a further ~0.5 m. This decline in projected NEP indicated that
a further WTD drawdown at this fen would eventually lead to a decline in GPP due to water
limitation. Therefore, representing the effects of interactions among hydrology, biogeochemistry
and plant physiological ecology on ecosystem carbon, water, and nutrient cycling in global
carbon models would improve our predictive capacity for changes in boreal peatland carbon
sequestration under changing climates.

## 1. Introduction

Northern boreal peatlands have been accumulating carbon (C) at a rate of about 20-30 g $m^{-2}$ $yr^{-1}$ over several thousand years (Gorham, 1991; Turunen et al., 2002). Drier and warmer future climates can affect the resilience of long-term boreal peatland C stocks by lowering water table (WT) that can halt or even reverse the C accumulation in boreal peatlands (Limpens et al., 2008; Dise, 2009; Frolking et al., 2011). To maintain and protect the C sequestration potentials of boreal peatlands we need an improved predictive capacity of how these C stocks would behave under future drier and warmer climates. However, boreal peatland C processes are currently under-represented in global C models largely due to inadequate simulation of hydrologic feedbacks to C cycles (St-Hilaire et al., 2010; Sulman et al., 2012). This can be overcome by integrating interactions between eco-hydrology of peatland vegetation, and peat biogeochemistry into finer resolution process models that can eventually be scaled up into larger spatial and temporal scale C models (Waddington et al., 2015).

The hydrologic feedbacks to boreal peatland C processes are largely mediated by water table depth (WTD) variation and its effects on peat-microbe-plant-atmosphere exchanges of C, energy, water and nutrients (Grant et al., 2012). WTD drawdown can affect net ecosystem productivity (NEP) of boreal peatlands through its effects on ecosystem respiration ($R_e$) and gross primary productivity (GPP). Receding WT can cause peat pore drainage that enhances microbial $O_2$ availability, energy yields, growth and decomposition and hence increases $R_e$ (Sulman et al., 2009, 2010; Cai et al., 2010; Flanagan and Syed, 2011; Peichl et al., 2014). The rate of increase in $R_e$ due to the WTD drawdown may vary with peat moisture retention and quality of peat forming substrates (Preston et al., 2012). For instance, peats with low moisture retention exhibit more rapid pore drainage than those with high moisture retention thus causing

more increase in $R_e$ for similar WTD drawdowns (Parmentier et al., 2009; Sulman et al., 2009,
2010; Cai et al., 2010). Peats formed from *Sphagnum* mosses degrade at rates slower than those
formed from remains of vascular plants (Moore and Basiliko, 2006). So for similar WTD
drawdowns, moss peats would generate less increase in microbial decomposition and hence $R_e$
than would sedge, reed or woody peats (Updegraff et al., 1995). Continued WTD drawdown can
also cause near surface peat desiccation from inadequate recharge through capillary rise from
deeper WT. Desiccation of near surface or shallow peat layers can cause a reduction in microbial
decomposition that can partially or fully offset the increased decomposition in the deeper peat
layers thereby yielding indistinct net effects of WTD drawdown on $R_e$ (Dimitrov et al., 2010a).
The interactions between WTD and GPP vary across peatlands depending upon peat
forming vegetation. For instance, increased aeration due to WTD drawdown enhances root $O_2$
availability and growth in vascular plants (Lieffers and Rothwell, 1987; Murphy et al., 2009).
Enhanced root growth is also associated with greater root nutrient availability and uptake from
more rapid mineralization facilitated by greater microbial energy yields, growth and
decomposition under deeper WT (Choi et al., 2007). Greater root nutrient uptake in turns
increases the rate of vascular $CO_2$ fixation and hence GPP (Sulman et al., 2009, 2010; Cai et al.,
2010; Flanagan and Syed, 2011; Peichl et al., 2014). WTD drawdown, however, does not affect
the non-vascular (e.g., moss) GPP in the same way it does the vascular GPP (Lafleur et al.,
2005). Non-vascular plants mostly depend upon the water available for uptake in the near surface
or shallow peat layers (Dimitrov et al., 2011). These layers can drain quickly with receding WT
and thus have to depend on moisture supply through capillary rise from deeper WT (Dimitrov et
al., 2011; Peichl et al., 2014). If recharge through the capillary rise is not adequate, near surface
peat desiccation occurs which slows moss water uptake, causes eventual drying of mosses and
reduces moss GPP (Lafleur et al., 2005; Riutta 2008; Sonnentag et al., 2010; Sulman et al., 2010;
Dimitrov et al., 2011; Kuiper et al., 2014; Peichl et al., 2014). Near surface peat desiccation also
suppresses vascular root water uptake from the desiccated layers (Lafleur et al., 2005; Dimitrov
et al., 2011). But enhanced root growth and elongation facilitated by improved $O_2$ status in the
newly aerated deeper peat layers under deeper WT enables vascular roots to take up water from
wetter deeper layers (Dimitrov et al., 2011). If deeper root water uptake offsets the reduction in
water uptake from desiccated near surface layers, vascular transpiration ($T$), canopy stomatal
conductance ($g_c$) and hence GPP are sustained under deeper WT (Dimitrov et al., 2011). But if
the WT falls below certain threshold level under which deeper root water uptake can no longer
sustain vascular $T$, $g_c$ and hence vascular GPP declines (Lafleur et al., 2005; Wu et al., 2010).

WTD variation can thus affect boreal peatland NEP through its effects on peat moisture

and aeration and consequent root and microbial oxidation-reduction reactions and energy yields.
To predict how boreal peatlands would behave under future drier and warmer climates, a
peatland C model thus needs to simulate WTD dynamics that determine the boundary between
aerobic and anaerobic zones and controls peat biogeochemistry. However, most of the current
process-based peatland C models either do not simulate a continuous anaerobic zone below a
prognostic WT (e.g., Baker et al., 2008; Schaefer et al., 2008; Tian et al., 2010), or do not
simulate peat saturation since any water in excess of field capacity is drained in those models
(e.g., Gerten et al., 2004; Krinner et al., 2005; Weng and Luo, 2008). Moreover, instead of
explicitly simulating the above-described hydrological and biological interactions between peat
aeration and biogeochemistry, most of those models use scalar functions of soil moisture
contents to inhibit $R_e$ and GPP under low or high moisture conditions (e.g., Frolking et al., 2002;
Zhang et al., 2002; Bond-Lamberty et al., 2007; St-Hilaire et al., 2010; Sulman et al., 2012).
Consequently, those peatland C models could not simulate declines in GPP and $R_e$ due to
shallow WT while simulating WTD effects on $CO_2$ exchange of peatlands across northern US
and Canada (Sulman et al., 2012). Furthermore, the approach of using scalar functions to
simulate moisture limitations to GPP and $R_e$ requires site-specific parameterization of model
algorithms which reduces scalabitiy of those peatland C models.

**1.1. Objective and rationale**

In this study, we tested a process-based ecosystem model *ecosys* against eddy covariance
(EC) net $CO_2$ fluxes measured over a drying period from 2004 to 2009 in a western Canadian
boreal fen peatland in Alberta, Canada (will be termed as WPL hereafter) (Syed et al., 2006;
Flanagan and Syed, 2011). The objective was to test whether the coupling of a dynamic WTD
that arises from vertical and lateral water fluxes as a function of a soil moisture retention scheme
with 1) oxygen transport, 2) microbial and root oxidation-reduction reactions and energy yields,
and 3) root, microbial and plant growth and uptake within a soil-plant-microbe-atmosphere
water, C and nutrient (nitrogen, phosphorus) scheme in an ecosystem process model could
explain underlying processes that govern hydrological effects on net $CO_2$ exchange of a northern
boreal fen under contrasting WTD conditions (e.g., shallower vs. deeper WTD). This study
would reconcile our knowledge on the feedback mechanisms among hydrology, eco-physiology,
and biogeochemistry of peatlands which are predominantly based upon inferences drawn from
EC-gap filled values that include empirically modelled estimates. It would also provide us with a
better insight into- and an improved predictive capacity of- how carbon deposits in northern
boreal peatlands would behave under changing climates. Rigorous site-scale testing of coupled
eco-hydrology and biogeochemistry algorithms in ecosystem process models such as *ecosys*
would also provide us with important insights on how to improve large-scale representation of
these processes into next generation land surface models.

**1.2. Hypotheses**

In an eddy covariance (EC) study, Flanagan and Syed (2011) found no net effect of a
weather driven WTD drawdown on NEP of WPL over 2004-2009. From the regressions of EC-
derived GPP and $R_e$ on site measured WTD, they inferred that the absence of a net WTD effect
on NEP was caused by similar increases in GPP and $R_e$ with WTD drawdown. We hypothesized
that coupled eco-hydrology and biogeochemistry algorithms in *ecosys* would be able to simulate
and explain underlying mechanisms of these effects of WTD drawdown on GPP and $R_e$ and
hence NEP at the WPL. We tested the following four central hypotheses:
(1) WTD drawdown would increase $R_e$ of the northern fen at the WPL. This effect of WTD
drawdown on $R_e$ would be modelled by simulation of peat pore drainage and improved peat
aeration that would increase the energy yields from aerobic microbial decomposition and hence
would increase $R_e$.
(2) WTD drawdown would increase GPP of the northern fen at the WPL. This effect of WTD
drawdown on GPP would be modelled by simulating enhanced microbial activity due to WTD
drawdown that would cause more rapid nutrient mineralization and greater root nutrient
availability and uptake, greater leaf nutrient concentrations and hence increased GPP.
(3) Increase in $R_e$ with WTD drawdown (hypothesis 1) would cease should WTD fall below a
threshold depth. This threshold WTD effect on $R_e$ would be modelled by simulating inadequate
recharge of the near surface peat layers through capillary rise from the deeper WTD below the
threshold level that would cause desiccation of those layers. Drying of near surface peat layers
and the surface residue would reduce near surface and surface peat respiration that would
partially offset the increase in deeper peat respiration due to aeration.
(4) Net effect of threshold WTD on GPP would be driven by the balance between how WTD
would affect vascular vs. non-vascular GPP. This threshold WTD effect on GPP would be
modelled by simulating vascular vs. non-vascular water relations under deeper WTD below the
threshold level. Near surface peat desiccation in hypothesis 3 would reduce peat water potential
and hydraulic conductivity and hence vascular and non-vascular water uptake from desiccated
near surface layers. Since non-vascular mosses depend mainly on near surface peat layers for
moisture supply, reduction in moss water uptake would cause a reduction in moss water potential
and hence moss GPP. On the contrary, suppression of vascular root water uptake from desiccated
near surface layers under deeper WT would be offset by increased deeper root water uptake from
newly aerated deeper peat layers with higher water potentials that would sustain vascular canopy
water potential ($\psi_c$), canopy stomatal conductance ($g_c$) and GPP.

**2. Methods**

**2.1. Model development**

*Ecosys* is a process-based ecosystem model which simulated 3D water, energy, carbon

and nutrient (nitrogen, phosphorus) cycles in different peatlands (Dimitrov et al., 2011; Grant et
al., 2012; Sulman et al., 2012; Mezbahuddin et al., 2014, 2015, 2016). *Ecosys* algorithms that
govern the modelled effects of WTD variations on peatland net $CO_2$ exchange are described
below. These algorithms in *ecosys* are derived from published independent basic research which
describe eco-hydrological and biogeochemical mechanisms that govern carbon, nutrient (N, P),
water, and energy balance of a typical boreal peatland ecosystem (Fig. 1). *Ecosys* algorithms
which are related to our hypotheses are depicted as a flowchart (Fig. 1) and cited as equations
within the text. These equations are also listed in sections S1-S4 of the supplementary material
with references to their sources for further clarification. These site-independent basic ecosystem
process algorithms in *ecosys* are thus not parameterized for each peatland site. Instead the
coupled algorithms are fed with peatland-specific measurable soil, weather, vegetation and
management inputs to simulate C, nutrient (N, P), water and energy balance of a particular
peatland ecosystem.

**2.1.1. Water table depth (WTD)**

The WTD in *ecosys* is calculated at the end of each time step as the depth to the top of the

saturated zone below which air-filled porosity is zero (Eq. D32). It is the depth at which lateral
water flux is in equilibrium with the difference between vertical influxes (precipitation) and
effluxes (evapotranspiration). Lateral water transfer between modelled grid cells in *ecosys* and
the adjacent ecosystem occurs to and from a set external WTD ($WTD_x$) over a set distance ($L_t$)
(Fig. 2). The $WTD_x$ represents average watershed WTD with reference to average hummock
surface. The WTD in *ecosys* is thus not prescribed, but rather controls, and is controlled by
lateral and vertical surface and subsurface water fluxes (Eqs. D1-D31). More detail about how
peatland WTD, vertical and lateral soil water flow, and soil moisture retention are modelled in
*ecosys* can be found in Dimitrov et al. (2010b) and Mezbahuddin et al. (2015, 2016).

**2.1.2. Heterotrophic respiration and WTD**

WTD fluctuation in *ecosys* determines the boundary between and the extent of aerobic vs.

anaerobic soil zones. So WTD fluctuation affects *ecosys*'s algorithms of organic oxidation-
reduction transformations and microbial energy yields, which drive microbial growth, substrate
decomposition and uptake (Fig. 1) (Eqs. A1-A30). Organic transformations in *ecosys* occur in a
residue layer and in each of the user defined soil layers within five organic matter-microbe
complexes i.e., coarse woody litter, fine non-woody litter, animal manure, particulate organic C
and humus (Fig. 1). Each of the complexes has three decomposition substrates i.e., solid organic
C, sorbed organic C and microbial residue C; the decomposition agent i.e., microbial biomass;
and the decomposition product i.e., dissolved organic C (DOC) (Fig. 1). Rates of the
decomposition and resulting DOC production in each of the complexes is a first-order function
of the fraction of substrate colonized by active biomasses ($M$) of diverse microbial functional
types (MFTs). The MFTs in *ecosys* are obligate aerobes (bacteria and fungi), facultative
anaerobes (denitrifiers), obligate anaerobes (fermenters), heterotrophic (acetotrophic) and
autotrophic (hydrogenotrophic) methanogens, and aerobic and anaerobic heterotrophic
diazotrophs (non-symbiotic $N_2$ fixers) (Fig. 1) (Eqs. A1-A2, A4). Biomass ($M$) growth of each of
the MFTs (Eq. A25a) is calculated from its DOC uptake (Fig. 1) (Eq. A21). The rate of $M$
growth is driven by energy yield from growth respiration ($R_g$) (Eq. A20) that is calculated by
subtracting maintenance respiration ($R_m$) (Eq. A18) from heterotrophic respiration ($R_h$) (Eq.
A11). The values of $R_h$ are driven by oxidation of DOC (Eq. A13). DOC oxidation may be
limited by microbial $O_2$ reduction (Eq. A14) driven by microbial $O_2$ demand (Eq. A16) and
constrained by $O_2$ diffusion calculated from aqueous $O_2$ concentrations in soil ($[O_{2s}]$) (Eq. A17).
Values of $[O_{2s}]$ are maintained by convective-dispersive transport of $O_2$ from the atmosphere to
gaseous and aqueous phases of the soil surface layer (Eq. D41), by convective-dispersive
transport of $O_2$ through gaseous and aqueous phases in adjacent soil layers (Eqs. D42, D44), and
by dissolution of $O_2$ from gaseous to aqueous phases within each soil layer (Eq. D39).

Shallow WTD in *ecosys* can cause lower air-filled porosity ($\theta_g$) in the wetter peat layers

above the WT. Lower $\theta_g$ reduces $O_2$ diffusivity in the gaseous phase ($D_g$) (Eq. D44) and gaseous
$O_2$ transport (Eqs. D41-D42) in these layers. Peat layers below the WT have zero $\theta_g$ that prevents
gaseous $O_2$ transport in these layers. So, under shallow WT, $[O_{2s}]$ relies more on $O_2$ transport
through the slower aqueous phase (Eq. D42) which causes a decline in $[O_{2s}]$. Decline in $[O_{2s}]$
slows $O_2$ uptake (Eq. A17) and hence $R_h$ (Eq. A14), $R_g$ (Eq. A20) and growth of $M$ (Eq. A25).
Lower $M$ slows decomposition of organic C (Eqs. A1-A2) and production of DOC which further
slows $R_h$ (Eq. A13), $R_g$ and growth of $M$ (Fig. 1). Although some MFTs can sustain DOC
oxidation by reducing alternative electron acceptors (e.g., methanogens reducing acetate or $CO_2$
to $CH_4$, and denitrifiers reducing $NO_x$ to $N_2O$ or $N_2$), lower energy yields from these reactions
reduce $R_g$ (Eq. A21), and hence $M$ growth, organic C decomposition and subsequent DOC
production (Fig. 1). Slower decomposition of organic C under low $[O_{2s}]$ also causes slower
decomposition of organic nitrogen (N) and phosphorus (P) (Eq. A7) and production of dissolved
organic nitrogen (DON) and phosphorus (DOP), which causes slower uptake of microbial N and
P (Eq. A22) and growth of $M$ (Eq. A29) (Fig. 1). Slower $M$ growth causes slower mineralization
(Eq. A26), and hence lowers aqueous concentrations of $NH_4^+$, $NO_3^-$ and $H_2PO_4^-$ (Fig. 1).
WTD drawdown can increase $\theta_g$ that results in greater $D_g$ (Eq. D44) and more rapid
gaseous $O_2$ transport. A consequent rise in $[O_{2s}]$ increases $O_2$ uptake (Eq. A17) and $R_h$ (Eq. A14),
$R_g$ (Eq. A20) and growth of $M$ (Eq. A25). Larger $M$ hastens decomposition of organic C (Eqs.
A1-A2) and production of DOC which further hastens $R_h$ (Eq. A13), $R_g$ and growth of $M$. More
rapid decomposition of organic C under adequate $[O_{2s}]$ in this period also causes more rapid
decomposition of organic N and P (Eq. A7) and production of DON and DOP, which increases
uptake of microbial N and P (Eq. A22) and growth of $M$ (Eq. A29) (Fig. 1). Rapid $M$ growth
causes rapid mineralization (Eq. A26), and hence greater aqueous concentrations of $NH_4^+$, $NO_3^-$
and $H_2PO_4^-$ (Fig. 1).
When WTD recedes below a certain threshold level, capillary rise from the WT can no
longer support adequate recharge of the near surface peat layers and the surface litter (Eqs. D9,
D12). It causes desiccation of the residue and the near surface peat layers thereby causing a
reduction in water potential ($\psi_s$) and an increase in aqueous microbial concentrations ([$M$]) in
each of these layers (Eq. A15). Increased [$M$] caused by the peat desiccation reduces microbial
access to the substrate for decomposition in each of the desiccated layers and reduces $R_h$ (Eq.
A13). Reduction in $R_h$ is calculated in *ecosys* from competitive inhibition of microbial exo-
enzymes with increasing concentrations (Eq. A4) (Lizama and Suzuki, 1991).
**2.1.3. WTD effects on vascular gross primary productivity**

*Ecosys* simulates effects of WTD variation on vascular GPP from WTD variation effects

on root $O_2$ and nutrient availability and root growth and uptake. Root $O_2$ and nutrient uptake in
*ecosys* are coupled with a hydraulically driven soil-plant-atmosphere water scheme. Root growth
in each vascular plant population in *ecosys* is calculated from its assimilation of the non-
structural C product of $CO_2$ fixation ($\sigma_C$) (Eq. C20). Assimilation is driven by $R_g$ (Eq. C17)
remaining after subtracting $R_m$ (Eq. C16) from autotrophic respiration ($R_a$) (Eq. C13) driven by
oxidation of $\sigma_C$ (Eq. C14). Oxidation in roots may be limited by root $O_2$ reduction (Eq. C14b)
which is driven by root $O_2$ demand to sustain C oxidation and nutrient uptake (Eq. C14e), and
constrained by $O_2$ uptake controlled by concentrations of aqueous $O_2$ in the soil ([$O_{2s}$]) and roots
([$O_{2r}$]) (Eq. C14d). Values of [$O_{2s}$] and [$O_{2r}$] are maintained by convective-dispersive transport
of $O_2$ through soil gaseous and aqueous phases and root gaseous phase (aerenchyma)
respectively and by dissolution of $O_2$ from soil and root gaseous to aqueous phases (Eqs. D39-
D45). $O_2$ transport through root aerenchyma depends on species-specific values used for root air-
filled porosity ($\theta_{pr}$) (Eq. D45). Shallow WTD and resultant high peat moisture content in *ecosys*
can cause low $\theta_g$ that reduces soil $O_2$ transport, forcing root $O_2$ uptake to rely more on $[O_{2r}]$ and
hence on root $O_2$ transport determined by $\theta_{pr}$. If this transport is inadequate, decline in $[O_{2r}]$
slows root $O_2$ uptake (Eqs. C14c-d) and hence $R_a$ (Eq. C14b), $R_g$ (Eq. C17) and root growth (Eq.
C20b) and root N and P uptake (Eqs. C23b, d, f) (Fig. 1). Root N and P uptake under shallow
WT is further slowed by reductions in aqueous concentrations of $NH_4^+$, $NO_3^-$ and $H_2PO_4^-$ (Eqs.
C23a, c, e) from slower mineralization of organic N and P (Fig. 1). Slower root N and P uptake
reduces concentrations of non-structural N and P products of root uptake ($\sigma_N$ and $\sigma_P$) with
respect to that of $\sigma_C$ in leaves (Eq. C11), thereby slowing $CO_2$ fixation (Eq. C6) and GPP.

WTD drawdown facilitates rapid $D_g$ which allows root $O_2$ demand to be almost entirely

met from $[O_{2s}]$ (Eqs. C14c-d) and so enables more rapid root growth and N and P uptake (Eqs.
C23b, d, f). Increased root growth and nutrient uptake is further stimulated by increased aqueous
concentrations of $NH_4^+$, $NO_3^-$ and $H_2PO_4^-$ (Eqs. C23a, c, e) from more rapid mineralization of
organic N and P during deeper WT (Fig. 1). Greater root N and P uptake increases
concentrations of $\sigma_N$ and $\sigma_P$ with respect to $\sigma_C$ in leaves (Eq. C11), thereby facilitating rapid $CO_2$
fixation (Eq. C6) and GPP. When WT falls below a certain threshold, inadequate capillary rise
(Eqs. A9, A12) from deeper WT causes near-surface peat desiccation that reduces soil water
potential ($\psi_s$) and raises soil hydraulic resistance ($\Omega_s$) (Eq. B9), thereby forcing lower root water
uptake ($U_w$) from desiccated layers (Fig. 1) (Eq. B6). However, deeper rooting facilitated by
increased $[O_{2s}]$ under deeper WT can sustain $U_w$ (Eq. B6) from wetter deeper peat layers with
higher $\psi_s$ and lower $\Omega_s$ (Eq. B9). If $U_w$ from the deeper wetter layers cannot offset the
suppression in $U_w$ from desiccated near surface layers, the resultant net decrease in $U_w$ causes a
reduction in root, canopy and turgor potentials ($\psi_r$, $\psi_c$ and $\psi_t$) (Eq. B4) and hence $g_c$ (Eq. B2b) in
*ecosys* when equilibrating $U_w$ with transpiration ($T$) (Eq. B14). Lower $g_c$ reduces $CO_2$ diffusion
into the leaves thereby reducing $CO_2$ fixation (Eq. C6) and GPP (Eq. C1) (Fig. 1).
**2.1.4. WTD effects on non-vascular gross primary productivity**

*Ecosys* simulates non-vascular plants (e.g., mosses) as tiny plants with no stomatal

regulations that grow on modelled hummock and hollow grid cells (Dimitrov et al., 2011).
Model input for moss population is usually larger and hence intra-specific competition for lights
and nutrients is greater so that individual moss plant and moss belowground growth (i.e. root like
structures for water and nutrient uptake) are smaller (Eq. C21b). Shallower belowground growth
of simulated mosses in *ecosys* means the water uptake of mosses are mostly confined to the near
surface peat layers. When WT deepens past a threshold level, inadequate capillary rise (Eqs. D9,
D12) causes near-surface peat desiccation, thereby reducing $\psi_s$ and increasing $\Omega_s$ (Eq. B9) of
those layers (Fig. 1). It causes a reduction in moss canopy water potential ($\psi_c$) while
equilibrating moss evaporation with moss $U_w$ (Eq. B6). Reduced moss $\psi_c$ causes a reduction in
moss carboxylation rate (Eqs. C3, C6a) and moss GPP (Fig. 1) (Eq. C1).
**2.2. Modelling experiment**
**2.2.1. Study site**

The peatland eco-hydrology and biogeochemistry algorithms in *ecosys* were tested in this

study against measurements of WTD and ecosystem net $CO_2$ fluxes at a flux station of the
Fluxnet-Canada Research Network established at the WPL (latitude: 54.95°N, longitude:
112.47°W). The study site is a moderately nutrient-rich treed fen peatland within the Central
Mixed-wood Sub-region of Boreal Alberta, Canada. Peat depth around the flux station was about
2 m. This peatland is dominated by stunted trees of black spruce (*Picea mariana*) and tamarack
(*Larix laricina*) with an average canopy height of 3 m. High abundance of a shrub species *Betula*
*pumila* (dwarf birch), and the presence of a wide range of mosses e.g., *Sphagnum* spp., feather
moss, and brown moss characterize the under-storey vegetation of WPL. The topographic,
climatic, edaphic and vegetative characteristics of this site were described in more details by
Syed et al. (2006).
**2.2.2. Field data sets**

*Ecosys* model inputs of half hourly weather variables i.e. incoming shortwave and

longwave radiation, air temperature ($T_a$), wind speed, precipitation and relative humidity during
2003-2009 were measured by Syed et al. (2006) and Flanagan and Syed (2011) at the
micrometeorological station installed at the WPL. To test the adequacy of WTD simulation in
*ecosys*, modelled outputs of hourly WTD were tested against WTD measured at the WPL with
respect to average hummock surface by Flanagan and Syed (2011). To examine how well *ecosys*
simulated net ecosystem $CO_2$ exchange at the WPL, we tested hourly modelled net ecosystem
$CO_2$ fluxes against hourly averaged measurements (average of two half-hourly) collected by
Syed et al. (2006) and Flanagan and Syed (2011) by using eddy covariance (EC) micro-
meteorological approach. Each of these EC-measured net $CO_2$ fluxes consisted of an eddy flux
and a storage flux (Syed et al. 2006). Erroneous flux measurements due to stable air conditions
were screened out with the use of a minimum friction velocity ($u^*$) threshold of 0.15 m s$^{-1}$ (Syed
et al. 2006). The net $CO_2$ fluxes that survived the quality control were used to derive half hourly
$R_e$ (=nighttime net $CO_2$ fluxes) and GPP (=daytime net $CO_2$ fluxes – $R_e$) (Barr et al., 2004; Syed
et al., 2006). To derive daily, seasonal and annual estimates of GPP, and $R_e$, the data gaps
resulting from the quality control were filled based on empirical relationships between soil
temperature at a shallow depth (0.05 m) and measured half hourly $R_e$, and between incoming
shortwave radiation and measured half hourly GPP using 15-days moving windows. The gap-
filled $R_e$ and GPP were then summed up for each half hour to fill NEP data gaps to derive daily,
seasonal and annual estimates of EC-gap filled NEP (Barr et al., 2004; Syed et al., 2006).
Soil $CO_2$ fluxes measured by automated chambers can provide a valuable supplement to
EC $CO_2$ fluxes in testing modelled respiration by providing more continuous measurements than
EC. So, we tested our modelled outputs against half-hourly automated chamber measurements by
Cai et al. (2010) at the WPL. These $CO_2$ flux measurements were carried out during ice-free
periods (May-October) of 2005 and 2006 over both hummocks and hollows by using a total of 9
non-steady state automatic transparent chambers (Cai et al., 2010). Along with soil respiration
these chamber $CO_2$ fluxes included fixation and autotrophic respiration from dwarf shrubs, herbs
and mosses (Cai et al., 2010).
Modelled WTD effects on peatland biogeochemistry and hence on peatland nutrient and
carbon cycling were also corroborated against leaf nitrogen concentrations, foliar N to P ratios, N
mineralization, and rooting depths biometrically measured at either our site or at sites that had
similar peat substrates, hydrology and/or plant functional types. Needles of black spruce and
tamarack, and leaves of dwarf birch were sampled over our study site during mid-summer of
2004 for foliar nutrient content analyses (Syed et al., 2006). Leaf nitrogen contents were
analyzed on $N_2$ gas that were generated from reduction of dried leaf tissues in an elemental
analyzer and quantified using a gas isotope ratio mass spectrometer (Syed et al., 2006). Leaf
phosphorus contents were analyzed on black spruce and tamrack needle leaf tissues that were
dry-ashed and then digested using a dilute $HNO_3$ and HCl mixture and then quantified using an
Inductively Coupled Plasma (ICP) spectroscopic analysis technique (Syed et al., 2006).
**2.2.3. Model run**
*Ecosys* model run to simulate WTD effects on net $CO_2$ exchange of WPL had a
hummock and a hollow grid cell that exchanged water, heat, carbon and nutrients (N, P) between
them and with surrounding vertical and lateral boundaries (Fig. 2). The hollow grid cell had near
surface peat layer that was 0.3 m thinner than the hummock cell representing a hummock-hollow
surface difference of 0.3 m observed in the field (Long, 2008) (Fig. 2). Any depth with respect to
the modelled hollow surface would thus be 0.3 m shallower than the depth with respect to the
modelled hummock surface.
Peat organic and chemical properties at different depths of the WPL were represented in
*ecosys* by inputs from measurements either at the site (e.g., Syed et al., 2006; Flanagan and Syed,
2011) or at similar nearby sites (e.g., Rippy and Nelson, 2007) (Fig. 2). *Ecosys* was run for a spin
up period of 1961-2002 under repeating 7-year sequences of hourly weather data (shortwave and
longwave radiation, air temperature, wind speed, humidity and precipitation) recorded at the site
from 2003 to 2009. There was a drying trend observed from 2003 to 2009 due to diminishing
precipitation that caused WTD drawdown in the watershed in which WPL is located, which
lowered the WT of this fen peatland (Flanagan and Syed, 2011). To accommodate the gradual
drying effects of catchment hydrology on modelled fen peatland WTD, we set the $WTD_x$ at
different levels based on the annual wetness of weather, e.g., shallow, intermediate, and deep
($WTD_x$=0.19, 0.35 and 0.72 m below the hummock surface, or 0.11 m above and 0.05 and 0.42
m below the hollow surface) (Fig. 2). There was no exchange of water through lower model
boundary to represent the presence of nearly impermeable clay sediment underlying the peat
(Syed et al., 2006) (Fig. 2). Variations in peat surface with WTD variations, which is an
important hydrologic self-regulation of boreal peatlands (Dise, 2009), was not represented in this
version of *ecosys*.

At the start of the spin up run, the hummock grid cell was seeded with an evergreen

needle leaf and a deciduous needle leaf over-storey plant functional types (PFT) to represent the
black spruce and tamarack trees at the WPL. The hollow grid cell was seeded with only the
deciduous needle leaf over-storey PFTs since the black spruce trees at the WPL only grew on the
raised areas. Each of the modelled hummock and the hollow was also seeded with a deciduous
broadleaved vascular (to represent dwarf birch) and a non-vascular (to represent mosses) under-
storey PFTs. The planting densities were such that the population densities of the black spruce,
tamarack, dwarf birch and moss PFTs were 0.16, 0.14, 0.3, and 500 m$^{-2}$ respectively at the end of
the spin up run so as to best represent field vegetation (Syed et al., 2006; Mezbahuddin et al.,
2016). To include wetland adaptation, we used a root porosity ($\theta_{pr}$) value of 0.1 for the two over-
storey PFTs and a higher $\theta_{pr}$ value of 0.3 for the under-storey vascular PFT to represent better
wetland adaptation in the under-storey than the over-storey PFTs. These $\theta_{pr}$ values were used in
calculating root $O_2$ transport through aerenchyma (Eq. D45) and did not change with
waterlogging throughout the model run. These $\theta_{pr}$ values were representatives of root porosities
measured for various northern boreal peatland plant species (Cronk and Fennessy, 2001). Non-
symbiotic $N_2$ fixation through association of cyanobacteria and mosses are also reported for
Canadian boreal forests (Markham, 2009). This was represented in *ecosys* as $N_2$ fixation by non-
symbiotic heterotrophic diazotrophs (Eq. A27) in the moss canopy. Further details about *ecosys*
model set up to represent the hydrological, physical and ecological characteristics of WPL can be
found in Mezbahuddin et al. (2016).
When the modelled ecosystem attained stable values of net ecosystem $CO_2$ exchange at
the end of the spin-up run, we continued the spin up run into a simulation run from 2003 to 2009
by using a real-time weather sequence. We tested our outputs from 2004-2009 of the simulation
run against the available site measurements of WTD, net EC $CO_2$ fluxes and net chamber $CO_2$
fluxes over those years.
**2.2.4. Model validation**
To examine the adequacy of modelling WTD effects on canopy, root and soil $CO_2$ fluxes
which were summed for net ecosystem $CO_2$ exchange at the WPL, we spatially averaged hourly
net $CO_2$ fluxes modelled over the hummock and the hollow to represent a 50:50 hummock-
hollow ratio and then regressed against hourly EC measured net ecosystem $CO_2$ fluxes for each
year from 2004-2009 with varying WTD. Each of these hourly EC measured net ecosystem $CO_2$
fluxes used in these regressions is an average of two half-hourly net $CO_2$ fluxes measured at a
friction velocity ($u^*$) greater than 0.15 m s$^{-1}$ that survived quality control procedure (Sec. 2.2.2).
Model performance was evaluated from regression intercepts ($a \rightarrow 0$), slopes ($b \rightarrow 1$), coefficients
of determination ($R^2 \rightarrow 1$), and root means squares for errors (RMSE$\rightarrow 0$) for each study year to
test whether there was any systematic divergence between the modelled and EC measured $CO_2$
fluxes.
Similar regressions were performed between modelled and automated chamber measured
net $CO_2$ fluxes for ice free periods (May-October) of 2005 and 2006 to further test the robustness
of modelled soil respiration under contrasting WTD conditions. Each of the half-hourly
measured chamber net $CO_2$ fluxes included soil respiration, and fixation and autotrophic
respiration from understorey vegetation (e.g., shrubs, herbs and mosses). So, we combined
modelled soil respiration with modelled fixation and autotrophic respiration from understorey
PFTs for comparison against these chamber measured net $CO_2$ fluxes. We also averaged net $CO_2$
flux measurements from all of the 9 chambers for each half hour to accommodate the variations
in those fluxes due to microtopography (e.g., hummock vs. hollow). Two half hourly averaged
values of net $CO_2$ fluxes were then averaged again to get hourly mean net chamber $CO_2$ fluxes
for comparison against modelled hourly sums of soil and understorey fluxes averaged over
modelled hummock and hollow. Model performance was evaluated from regression intercepts
($a \rightarrow 0$), slopes ($b \rightarrow 1$), coefficients of determination ($R^2 \rightarrow 1$), and root means squares for errors
(RMSE$\rightarrow$0) for each of 2005 and 2006.
**2.2.5. Sensitivity of modelled peatland $CO_2$ exchange to artificial drainage**
Large areas of northern boreal peatlands in Canada have been drained primarily for
increased forest and agricultural production since plant productivity in pristine peatlands are
known to be constrained by shallow WTD (Choi et al., 2007). Drainage and resultant WTD
drawdown can affect both GPP and $R_e$ on a short-term basis and the vegetation composition on a
longer time scale thereby changing overall net $CO_2$ exchange trajectories of a peatland. To
predict short-term effects of drainage on WTD and hence ecosystem net $CO_2$ exchange of WPL,
we extended our simulation run into a projection run consisting two 7-yr cycles by using
repeated weather sequences of 2003-2009.  While doing so, we forced a stepwise drawdown in
$WTD_x$ by 1.0 and 2.0 m from that used in spin-up and simulation runs (Fig. 2) in the first
(drainage cycle 1) and the second cycle (drainage cycle 2) respectively. This projection run
would give us a further insight about how the northern boreal peatland of Western Canada would
be affected by further WTD drawdown as a result of drier and warmer weather as well as a
disturbance such as drainage. It would also provide us with a test of how sensitive the modelled
C processes were to the changes in model lateral boundary condition as defined by $WTD_x$ in
*ecosys*.

**3. Results**

**3.1. Model performance in simulating diurnal variations in ecosystem net $CO_2$ fluxes**

Variations in precipitation can cause change in WTD and consequent variation in diurnal

net $CO_2$ exchange across years. *Ecosys* simulated hourly EC-measured net $CO_2$ fluxes well over
2004-2008 with varying precipitation (Table 1a). On a year-to-year basis, regressions of hourly
modelled vs. EC-measured net ecosystem $CO_2$ fluxes gave intercepts within 0.1 µmol m$^{-2}$ s$^{-1}$ of
zero, and slopes within 0.1 of one, indicating minimal bias in modelled outputs during each year
from 2004-2008 (Table 1a). On a growing season (May-August) basis, regressions of modelled
on measured hourly net $CO_2$ fluxes yielded larger positive intercepts from 2004-2009 (Table 1b).
The larger intercepts were predominantly caused by modelled overestimation of growing season
day-time $CO_2$ fluxes. This overestimation was offset by modelled overestimation of night-time
$CO_2$ fluxes during the winter thus yielding smaller intercepts from throughout-the-year
regressions of modelled vs. EC measured fluxes (Tables 1a vs. b). Values for coefficients of
determination ($R^2$) were ~ 0.8 ($P < 0.001$) for all years from both throughout-the-year and
growing season regressions (Tables 1a, b). RMSEs were < 2.0 and ~2.5 µmol m$^{-2}$ s$^{-1}$ for whole
year regressions from 2004-2008 (Table 1a) and for growing season regressions from 2004-2009
(Table 1b) respectively. Much of the variations in EC measured $CO_2$ fluxes that was not
explained by the modelled fluxes could be attributed to a random error of ~20% in EC
methodology (Wesely and Hart, 1985). This attribution was further corroborated by root mean
squares for random errors (RMSRE) in EC measurements, calculated for forests with similar
$CO_2$ fluxes from Richardson et al. (2006) that were similar to RMSE (Tables 1a, b). The similar
values of RMSE and RMSRE also indicated that further constraint in model testing could not be
achieved without further precision in EC measurements.

Regressions of modelled vs. chamber measured net $CO_2$ fluxes gave $R^2$ of ~0.7 for ice-

free periods (May-October) of 2005 and 2006 indicated that the variations in soil respiration, and
the fixation and aboveground autotrophic respiration due to WTD drawdown were modelled well
(Table 1c). Smaller intercepts from those regressions meant lower model biases in simulating
soil and understorey $CO_2$ fluxes under deepening WT (Table 1c). Although the slope was within
0.1 of one in 2005, it was a bit smaller in 2006 indicating lower modelled vs. chamber measured
soil and understorey net $CO_2$ fluxes in 2006 (Table 1c). It was because some of the nighttime
chamber fluxes in warmer nights of summer 2006 were as large as the EC measured ecosystem
net $CO_2$ fluxes corresponding to those same hours which could not be modelled to their full
extent. RMSE lower than RMSRE meant the errors in modelling soil and understorey $CO_2$ fluxes
were within the limit of random errors due to chamber measurements (Table 1c). It further
indicated the robustness of modelled outputs for soil and understorey $CO_2$ fluxes under different
WTD conditions (e.g., shallower in 2005 vs. deeper in 2006) (Table 1c).
**3.2. Seasonality in WTD and net ecosystem $CO_2$ exchange**

*Ecosys* simulated the seasonal and interannual variations in WTD from 2004 to 2009 well

at the WPL (Figs. 3b, d, f, h, j, l) (Mezbahuddin et al., 2016). Seasonality in net $CO_2$ exchange at
the WPL was predominantly governed by that in temperature which controlled the seasonality in
phenology and GPP as well as that in $R_e$. *Ecosys* simulated the seasonality in phenology and
hence GPP, and $R_e$ well during a gradual growing season WTD drawdown from 2004 to 2009
which was apparent by good agreements between modelled vs. EC-gap filled daily NEP (Fig. 3)
and modelled vs. EC-measured hourly net $CO_2$ fluxes (Table 1). Modelled NEP throughout the
winters of most of the years were more negative than the EC-gap filled NEP indicating larger
modelled $CO_2$ effluxes than EC-gap filled fluxes during the winter (Fig. 3). The onset of
photosynthesis at the WPL varied interannually depending upon spring temperature which was
also modelled well by *ecosys*. For instance, *ecosys* modelled a smaller early growing season
(May) GPP and hence NEP in 2004 with a cooler spring than 2005 which was also apparent in
daily EC-gap filled NEP (Figs. 3a vs. c).
**3.3. WTD effects on diurnal net ecosystem $CO_2$ exchange**

WTD variation can affect diurnal net $CO_2$ exchange by affecting peat $O_2$ status and

consequently root and microbial $O_2$ and nutrient availability, growth and uptake thereby
influencing $CO_2$ fixation and respiration. *Ecosys* simulated WTD effects on diurnal net $CO_2$
exchange at the WPL well over three 10-day periods with comparable weather conditions
(radiation and air temperature) during late growing seasons (August) of 2005, 2006 and 2008
(Fig. 4). A WTD drawdown from August 2005 to August 2006 in *ecosys* caused a reduction in
peat water contents and a consequent increase in $O_2$ influxes from atmosphere into the peat that
eventually caused an increase in modelled soil $CO_2$ effluxes (Fig. 5c). This stimulation of soil
respiration was corroborated by modelled vs. chamber measured (Cai et al., 2010) night-time soil
$CO_2$ fluxes and understorey autotrophic respiration ($R_a$) in August 2006 with deeper WTD which
were larger than those in 2005 with shallower WTD (Fig. 5b). Larger modelled soil $CO_2$ effluxes
in 2006 contributed to the larger modelled ecosystem $CO_2$ effluxes ($R_e$) that was also apparent in
night-time EC $CO_2$ fluxes in 2006 which were larger than those in 2005 (Fig. 5a).

Continued WTD drawdown into the late growing season of 2008 (Fig. 4c) sustained

improved peat oxygenation and hence larger modelled soil $CO_2$ effluxes (Fig. 5c). Consequently,
modelled night-time net ecosystem $CO_2$ fluxes, and soil and understorey $CO_2$ fluxes in 2006 and
in 2008 were similarly larger than those in 2005 which was corroborated well by EC measured
night-time fluxes during 2006 and 2008 vs. 2005 (Figs. 5a-b). Although night-time modelled and
EC measured net ecosystem $CO_2$ fluxes in 2008 were larger than those in 2005, the day-time
modelled and EC measured $CO_2$ fluxes in 2008 did not decline with respect to those in 2005
(Fig. 5a). Similar day-time fluxes in 2005 and 2008 despite larger night-time fluxes in 2008 than
in 2005 indicated a greater late growing season $CO_2$ fixation in 2008 with deeper WTD than in
2005 with shallower WTD.
Beside WTD, temperature variation also profoundly affected ecosystem net $CO_2$
exchange at the WPL. For a given WTD condition warmer weather caused increases in $R_e$ at the
WPL (Figs. 4b-c and 5a-b). Night-time modelled, and EC and chamber measured ecosystem,
soil, and understorey $CO_2$ fluxes, in warmer nights of day 214, 220 and 222 were larger than
those in cooler nights of day 221, 224 and 218 in 2005, 2006 and 2008 respectively (Figs. 4b and
5a-b). However, for a given temperature modelled and EC-gap filled night-time ecosystem $CO_2$
fluxes, and modelled and chamber measured night-time soil and understorey $CO_2$ fluxes were
larger under deeper WT conditions in 2006 and 2008 than under shallower WT condition in 2005
(denoted by the grey arrows in Figs. 4b and 5a-b). It showed net WTD drawdown effect on $R_e$
(separated from temperature effect) and hence on NEP.
The degree of $R_e$ stimulation due to warming was also influenced by WTD at the WPL.
The warming events in early to mid-August of 2006 and 2008, when WT was deeper than in late
July of 2005, caused gradual increases in modelled, and EC and chamber measured night-time
ecosystem, soil and understorey $CO_2$ effluxes (=$R_e$) (Figs. 6h, i, k, l). This $R_e$ stimulation due to
warming under deeper WT contributed to declines in modelled and EC-gap filled July-August
NEP in 2006 and 2008 (Figs. 3e, i). Lack of similar stimulation in $R_e$ with warming under
shallower WT in 2005 did not yield a similarly evident stimulation of either modelled or EC or
chamber measured ecosystem, soil and understorey night-time $CO_2$ effluxes (Figs. 6g, j vs. h, i,
k, l) which resulted in the absence of decline in July-August NEP as occurred in 2006 and 2008
(Figs. 3c vs. e, i). Greater warming driven $R_e$ stimulation under deeper WT further indicated the
importance of WTD in mediating potential effects of future warmer climates on boreal peatland
NEP.
**3.4. Interannual variations in WTD and net ecosystem productivity**
The effects of WTD drawdown on modelled and EC-gap filled diurnal net ecosystem
$CO_2$ exchange also contributed to the effects of interannual variation in WTD on that of NEP.
*Ecosys* simulated a site measured gradual drawdown of average growing season (May-August)
WTD well from 2004 to 2009 (Fig. 7d) (Mezbahuddin et al., 2016). A small WTD drawdown
simulated a large increase in growing season GPP from 2004 to 2005 as corroborated by similar
increase in EC-derived GPP (Fig. 7b). This increase in GPP was also contributed by a larger GPP
in May 2005 which was warmer than May 2004. This small WTD drawdown, however, did not
raise either modelled or EC-derived growing season $R_e$ from 2004 to 2005 (Fig. 7c). June and
July of 2005 was cooler than 2004 by over 2°C which caused cooler soil that reduced $R_e$ in 2005.
Reduction in $R_e$ in June-July 2005 due to cooler soil more than fully offset the increase in $R_e$ due
to the small WTD drawdown and resulted in modelled and EC-derived $R_e$ that were smaller in
the growing season of 2005 than in 2004 (Fig. 7c). Larger GPP and smaller $R_e$ gave rise to
modelled and EC-gap filled growing season NEP estimates that were larger in 2005 than those in
2004 (Fig. 7a).
WTD drawdown from 2005 to 2006 raised both modelled and EC-derived growing
season GPP and $R_e$ (Figs. 7b-c). Warmer growing season in 2006 caused warmer soil that further
contributed to the increase in modelled and EC-derived growing season $R_e$ from 2005 with
shallower WT to 2006 with deeper WT (Figs. 5, 6 and 7c-d). An increase in growing season $R_e$
that was greater than the increase in growing season GPP caused a decline in modelled and EC-
derived growing season NEP from 2005 to 2006 (Fig. 7a). Continued growing season WTD
drawdown from 2006 to 2008 caused similar increases in modelled growing season GPP and $R_e$
and hence no significant change in modelled growing season NEP (Figs. 7a-d). With this
continued WTD drawdown from 2006 to 2008, however, EC-derived growing season GPP
increased more than EC-derived growing season $R_e$ that resulted a larger EC-derived NEP in the
growing season of 2008 than in 2006 (Figs. 7a-d). A further drawdown in WTD from the
growing season of 2008 to that of 2009 caused reductions in both modelled and EC-derived
growing season GPP and $R_e$ (Figs. 7a-d). Reductions in GPP and $R_e$ from 2008 to 2009 was also
contributed by lower $T_a$ in 2009 than in 2008 that caused cooler canopies and soil (Figs. 7b-d).
The reduction in EC-derived growing season GPP was larger than that in EC-derived growing
season $R_e$ thereby causing a decrease in growing season EC-gap filled NEP from 2008 to 2009
(Figs. 7a- c). On the contrary, the reduction in modelled growing season GPP was smaller than
the reduction in modelled $R_e$ that yielded an increase in modelled growing season NEP from
2008 to 2009 (Figs. 7a-c).

Modelled and EC-derived estimates of growing season GPP and $R_e$ in 2009 were larger

than those in 2004 despite similar mean $T_a$ in those years (Figs. 7a-d). It suggested that increases
in growing season GPP and $R_e$ from 2004 to 2009 was a net effect of the deepening of average
growing season WT (Figs. 7a-d). It was further corroborated by polynomial regressions of
modelled growing season estimates of GPP and $R_e$ against modelled average growing season
WTD, and similar regressions of EC-derived growing season GPP and $R_e$ against site measured
average growing season WTD (Figs. 8a-c). These relationships showed that there were increases
in modelled and EC-derived growing season GPP and $R_e$ with deepening of the growing season
WT from 2004 to 2008 after which further WTD drawdown in 2009 started to cause slight
declines in both GPP and $R_e$ (Figs. 8b-c). Neither modelled nor EC-gap filled estimates of
growing season NEP yielded significant regressions when regressed against modelled and
measured growing season WTD respectively (Fig. 8a). It indicated that similar increases in
modelled and EC-derived growing season estimates of GPP and $R_e$ with deepening of WT left no
net effects of WTD drawdown on either modelled or EC-derived growing season NEP (Figs. 7a-
d and 8a).

Similar to the growing season trend, drawdown of both measured and modelled WTD

averaged over the ice free periods (May-October) from 2004 to 2008 generally stimulated annual
modelled and EC-derived GPP and $R_e$ (Figs. 7f, g, h and 8e, f). Similar increases in both
modelled and EC-derived annual GPP and $R_e$ with WTD drawdown left no net WTD effects on
modelled and EC-gap filled annual NEP (Figs. 7e, 8d). Although modelled WTD effects on GPP,
$R_e$ and hence NEP corroborated well by EC-derived GPP, $R_e$ and EC-gap filled NEP, the
modelled values for growing season and annual GPP and $R_e$ were consistently higher than the
EC-derived estimates of those throughout the study period (Figs. 7b, c, f, g and 8b, c, e, f).

Increased GPP with WTD drawdown (Figs. 7b, f and 8b, e) was modelled predominantly

through increased root growth and uptake of nutrients and consequently improved leaf nutrient
status and hence more rapid $CO_2$ fixation in vascular PFTs. Under shallow WT during the
growing season of 2004, roots in modelled black spruce and tamarack PFTs hardly grew below
0.35 m from the hummock surface. Modelled root densities of both black spruce and tamarack
were higher by 2-3 orders of magnitude in the top 0.19 m of the hummock (data not shown). A
WTD drawdown by 0.35 m from the growing season of 2004 to that of 2009 caused increase in
maximum modelled rooting depth in both PFTs (Table 2). Increased root growth in modelled
vascular PFTs augmented root surface area for nutrient uptake under deeper WT in the growing
season of 2009 than in 2004. Increased root surface area along with increased nutrient
availability due to more rapid mineralization with improved aeration as a result of WTD
drawdown from 2004 to 2009 caused improved root nutrient uptake in modelled vascular PFTs.
Increased root growth, nutrient availability and hence uptake due to WTD drawdown from the
growing season of 2004 to that of 2009 caused an increase in modelled foliar N concentrations in
black spruce, tamarack and dwarf birch PFTs driving the increases in GPP modelled over this
period (Figs. 7b, f and 8b, e) (Table 2).
**3.5. Simulated drainage effects on WTD and NEP**

Artificial drainage can drastically alter WTD in a peatland that can cause dramatic

changes in peatland NEP by shifting the balance between GPP and $R_e$. Projected growing season
WT was deeper by ~0.5 m and ~0.55 m respectively from those in the real-time simulation in
drainage cycles 1 and 2 in all the years from 2004 to 2009 (Fig. 9a). Modelled growing season
GPP increased with drainage-induced WTD drawdown up to ~0.5 m below the hollow surface
(~0.8 m below the hummock surface) below which GPP decreased (Figs. 9c, f). The WTD
drawdown affected modelled vascular and non-vascular growing season GPP quite differently.
Modelled growing season vascular GPP increased with WTD drawdown before it plateaued and
eventually decreased when WTD fell below ~0.6 m from the hollow surface (~0.9 m below the
hummock surface) (Figs. 10a, c, e). On the contrary, modelled non-vascular growing season GPP
continued to decrease with WTD drawdown below ~0.1 m from the hollow surface (~0.4 m
below the hummock surface) (Figs. 10a-b, d).

WTD drawdown due to simulated drainage not only affected modelled growing season

GPP but also affected, and was affected by, the associated change in transpiration from vascular
canopies. Deeper $WTD_x$ in drainage cycle 1 caused larger hydraulic gradients and greater lateral
discharge thereby deepening the WT with respect to that in the real-time simulation (Figs. 9a).
Larger GPP throughout the growing seasons of 2004-2007 in the drainage cycle 1 than in the
real-time simulation caused a greater vertical water loss through rapid transpiration from
vascular canopies that further contributed to this deepening of WT (Fig. 9c). However, greater
lateral water discharge in drainage cycle 2 caused by deeper $WTD_x$ did not deepen the modelled
growing season WT much below that in cycle 1 (Fig. 9a). The larger lateral water loss through
discharge in drainage cycle 2 than in cycle 1 was mostly offset by slower vertical water losses
due to vascular plant water stress as indicated by smaller GPP in the drainage cycle 2 (Fig. 9c).
The changing feedbacks between WTD, and GPP and plant water relations in *ecosys* also
indicated the ability of the model to simulate hydrological self-regulation which is an important
characteristic of peatland eco-hydrology (Dise, 2009).

Modelled growing season $R_e$ continued to increase with projected drainage driven WTD

drawdown (Figs. 9d, g). Reductions in modelled growing season $R_e$ from drainage cycle 1 to 2
during 2006-2009 indicated $R_e$ inhibition due to desiccation of near surface peat layers and
surface residues (Fig. 9d). Overall GPP increased more than $R_e$ with drainage driven initial WTD
drawdown that caused a small increase in modelled growing season NEP (Figs. 9 b, e).
Continued drainage driven WTD drawdown, however, caused declines in GPP particularly in
model years of 2008 and 2009 (Figs. 9c). This decrease in GPP was also accompanied by
increased $R_e$ thereby causing a decrease in NEP when WT fell below a threshold of about ~0.45
m from the hollow surface, particularly during the drier years (Figs. 9b-g). This projected
drainage effect on WTD and NEP may be transient. Long-term manipulation of WTD may
produce different trajectories of WTD effects on C processes and plant water relations in
northern boreal peatlands via vegetation adaptation and succession (Strack et al., 2006; Munir et
al., 2014).
**4. Discussion**
**4.1. Modelling WTD effects on northern boreal peatland NEP**

Hourly modelled, EC measured, and chamber measured net ecosystem, and soil and

understorey $CO_2$ fluxes, and modelled and EC-derived seasonal and annual NEP, GPP and $R_e$
estimates showed that WTD drawdown raised both GPP and $R_e$ at the WPL (Figs. 3-8). Similar
increases in GPP and $R_e$ with WTD drawdown yielded no net effect of WTD drawdown on NEP
at the WPL during 2004-2009 (Figs. 7-8). Four central hypotheses which outlined how coupled
eco-hydrology and biogeochemistry algorithms in *ecosys* would simulate and explain the
mechanisms of these WTD effects on $R_e$, GPP and NEP at the boreal fen peatland under study
are examined in details in the following sections of 4.1.1 to 4.1.4.
**4.1.1. Hypothesis 1: Increase in $R_e$ with WTD drawdown**

Shallow WTD in *ecosys* caused shallow aerobic zone above WT and thicker anaerobic

zone below the WT. In the shallow aerobic zone, peat $O_2$ concentration $[O_{2s}]$ was well above the
Michaelis-Menten constant for $O_2$ reduction ($K_m$=0.064 g m$^{-3}$) and hence DOC oxidation and
consequent microbial uptake and growth in *ecosys* was not much limited by $[O_{2s}]$ (Eqs. A17a,
C14c). On the contrary, $[O_{2s}]$ in the thicker anaerobic zone below the WT was well below $K_m$ so
that DOC oxidation was coupled with DOC reduction by anaerobic heterotrophic fermenters,
which yielded much less energy (4.4 kJ g$^{-1}$ C) than did DOC oxidation coupled with $O_2$
reduction (37.5 kJ g$^{-1}$ C) (Fig. 1) (Eq. A21). Lower energy yields in the thicker anaerobic zone
resulted in slower microbial growth (Eq. A25) and $R_h$ (Eq. A13). Since the anaerobic zone in
*ecosys* was thicker than the aerobic zone under shallow WT, lower modelled $R_h$ in the anaerobic
zone contributed to reduced modelled soil respiration and hence $R_e$ that was corroborated by EC
and chamber measurements at the WPL (Figs. 5-8) (Tables 1 and 2).

WTD drawdown in *ecosys* caused peat pore drainage and increased $\theta_g$ thereby deepening

of the aerobic zone. It raised $D_g$ (Eq. D44) and increased $O_2$ influxes into the peat (Fig. 5c) (Eqs.
D42-D43). Increased $O_2$ influxes enhanced $[O_{2s}]$ and stimulated $R_h$ (Eqs. A13, A20), soil
respiration and hence $R_e$ (Figs. 5-8). Rapid mineralization of DON and DOP due to improved
$[O_{2s}]$ under deeper WT also raised aqueous concentrations of $NH_4^+$, $NO_3^-$ and $H_2PO_4^-$ (Eqs.
C23a, c, e) that increased microbial nutrient availability, uptake (Eq. A22) and growth (Eq. A29)
and further enhanced $R_e$ (Fig. 1) (Figs. 5-8). Modelled $R_e$ stimulation by improved peat
oxygenation due to WTD drawdown was corroborated well by EC and chamber measurements at
the site (Figs. 5-8) (Tables 1 and 2). However, the chamber measured nighttime net $CO_2$ fluxes
during warmer nights of 2006 with deeper WT were sometimes as large as corresponding EC-
measured net ecosystem $CO_2$ fluxes (Figs. 5-6). Those very large chamber $CO_2$ effluxes could
not be modelled to their full extent and consequently modelled vs. chamber $CO_2$ flux regression
yielded a slope lower than $1\pm0.1$ in 2006 (Table 1c). Although modelled rate of increase in $R_e$
with each 0.1 m of WTD drawdown was larger than EC-derived rate, it is still comparable with
rates reported for other similar peatlands (Table 2). Kotowska (2013) carried out chamber based
field measurements and laboratory incubation experiments in a moderately rich fen very close to
our study site which reported that a WTD drawdown driven stimulation of aerobic microbial
decomposition contributed to increased $R_e$. Mäkiranta et al. (2009) also found rapid microbial
decomposition in a Finish peatland due to thicker aerobic zone and consequently larger amounts
of decomposable organic matter exposed to aerobic oxidation.

Apart from WTD, peat warming in *ecosys* also increased rates of decomposition (Eq. A1)

through an Arrhenius function (Eq. A6) and increased $R_h$ and $R_e$ (Figs. 4-7). Warming effect on
decomposition in *ecosys* was also modified by WTD. For a similar warming, greater thermal
diffusivity in peat with deeper WT and consequent smaller water contents caused greater peat
warming (Eqs. D34, D36). It enabled larger simulation of $R_e$ during warming periods in 2006
and 2008 with deeper WT than in 2005 (Fig. 6). Increased stimulation of peat decomposition by
warming under deeper WT was also modelled by Grant et al. (2012) using the same model
*ecosys* over a northern fen peatland at Wisconsin, USA and by Ise et al. (2008) using a land
surface scheme named ED-RAMS (Ecosystem Demography Model version 2 integrated with the
Regional Atmospheric Modeling System) coupled with a soil biogeochemical model across
several shallow and deep peat deposits in Manitoba, Canada.
**4.1.2. Hypothesis 2: Increase in GPP with WTD drawdown**

Modelled WTD variations influenced GPP by controlling root and microbial $O_2$

availability, energy yields, root and microbial growth and decomposition, rates of mineralization
and hence root nutrient availability and uptake (Fig. 1). Wet soils under shallow WT caused low
$O_2$ diffusion (Fig. 5c) (Eqs. D42-D44) into the peat and consequent low $[O_{2s}]$ meant that root $O_2$
demand had to be mostly met by $[O_{2r}]$. *Ecosys* inputs for root porosity ($\theta_{pr} = 0.1$) that governed
$O_2$ transport through aerenchyma (Eq. D45) and hence maintained $[O_{2r}]$ was not enough to meet
the root $O_2$ demand in saturated soil by the two over-storey tree PFTs i.e. black spruce and
tamarack, causing shallow root systems to be simulated in these two tree PFTs under shallow
WTD (Sec. 3.4). The under-storey shrub PFT (dwarf birch) had a higher root porosity ($\theta_{pr}=0.3$)
and hence had deeper rooting under shallow WT than the two tree PFTs (Sec. 3.4). Shallow
rooting in the tree PFTs reduced root surface area for nutrient uptake. Root nutrient uptake (Eqs.
C23b, d, f) in all the PFTs was also constrained by low nutrient availability due to smaller
aqueous concentrations of $NH_4^+$, $NO_3^-$ and/or $H_2PO_4^-$ (Eqs. C23a, c, e) resulting from slower
mineralization (Eq. A26) of DON and DOP (Eq. A7) because of low $[O_{2s}]$ in the wet soils under
shallow WT (Fig. 1). Slower root growth and nutrient uptake caused lower foliar $\sigma_N$ and/or $\sigma_P$
with respect to foliar $\sigma_C$ (Eq. C11) that slowed the rates of carboxylation (Eq. C6) and hence
reduced vascular GPP (Eq. C1) under shallow WT.
WTD drawdown enhanced $O_2$ diffusion (Fig. 5c) (Eqs. D42-D44) and raised $[O_{2s}]$ so that
root $O_2$ demand in all the three vascular PFTs was almost entirely met by $[O_{2s}]$. Consequently
roots in all the PFTs could grow deeper which increased the root surface for nutrient uptake
(Table 2) (Sec. 3.4). Increase in modelled rooting depth due to WTD drawdown was
corroborated well by studies on same PFTs as in our study grown on similar peatlands very close
to the study site (Table 2). Murphy et al. (2009) found a significant increase in tree fine root
production with WTD drawdown by 0.15-0.2 m during a WTD manipulation study in a Finish
peatland. Beside improved root growth, greater $[O_{2s}]$ under deeper WT also enhanced rates of
mineralization (Eq. A26) of DON and DOP (Eq. A7) that raised aqueous concentrations of $NH_4^+$,
$NO_3^-$ and/or $H_2PO_4^-$ and hence facilitated root nutrient availability and uptake (Fig. 1). Enhanced
root nutrient uptake increased foliar $\sigma_N$ and/or $\sigma_P$ with respect to foliar $\sigma_C$ (Eq. C11) that
hastened the rates of carboxylation (Eq. C6) and hence raised vascular GPP (Eq. C1) under
deeper WT.
The three modelled vascular PFT were predominantly N limited as indicated by mass-
based modelled foliar N to P ratios that matched well with site-measured mass-based foliar N to
P ratios (Table 2). Mass-based modelled and measured foliar N to P ratio in all the PFTs were
less than 16:1 indicating that the vegetation at the WPL was N limited (Aerts and Chapin III,
1999). Since the modelled PFTs were predominantly N limited, increases in foliar N
concentrations as a result of improved root nutrient availability, growth and nutrient uptake with
WTD drawdown enhanced modelled carboxylation rates and hence modelled GPP. In a similar
fen peatland close to our study site, Choi et al. (2007) found an increase in peat $NO_3^-$ -N due to
enhanced mineralization and nitrification stimulated by a WTD drawdown which improved
foliar N status, and hence increased radial tree growth of black spruce and tamarack (Table 2).
Macdonald and Lieffers (1990) also found improved foliar N concentrations in black spruce and
tamarack trees that enhanced net photosynthetic C assimilation rates by those tree species in a
northern Alberta moderately rich fen (Table 2). The rates of increases in foliar N concentrations
in black spruce and tamarack trees due to WTD drawdown as reported in those studies are
comparable with those in our modelled outputs (Table 2). Although modelled rate of increase in
GPP with each 0.1 m of WTD drawdown was larger than EC-derived rate, it is still comparable
with rates reported for other similar peatlands (Table 2).
**4.1.3. Hypothesis 3: Microbial water stress on $R_e$ due to WT deepening below a threshold**
**WTD**

When modelled WTD fell below a threshold of ~0.3 m from the hollow surface (~0.6 m

below the hummock surface), desiccation of the surface residue layer and near surface shallow
peat layers reduced microbial access to substrate for decomposition (Eq. A15) which enabled
simulation of reduced $R_h$ in those layers. When reduction in surface residue and near-surface $R_h$
more than fully offset the increase in deeper $R_h$, net ecosystem $R_h$ decreased. The offsetting
effect on $R_h$ partly contributed to simulated decrease in growing season $R_e$ (=$R_h$+$R_a$) from 2008
to 2009 with WTD drawdown that was corroborated by a similar decrease in EC-derived $R_e$ (Fig.
7c). Greater reductions in $R_h$ in desiccated surface residue and near surface peat layers also
caused the reductions in growing season $R_e$ in drainage cycle 2 from those in cycle 1 during
2007-2009 in the simulated drainage study (Fig. 9d). Similar to our study, Peichl et al. (2014)
found reductions in $R_e$ when WTD fell below a threshold of 0.25 m from the peat surface in a
Swedish fen which could be partially attributed to reduction in near surface $R_h$ due to
desiccation. Mettrop et al. (2014) in a controlled incubation experiment found that the rates of
microbial respiration in a nutrient rich Dutch fen initially increased with peat drying and
consequent improved aeration. But excessive drying and consequent peat desiccation in their
study reduced microbial respiration efficiency, growth and biomass. Dimitrov et al. (2010a)
while modelling $CO_2$ exchange of a northern temperate bog using *ecosys* showed that a decrease
in desiccated near surface peat respiration partially offset increased deeper peat respiration when
WT deepened below a threshold of 0.6-0.7 m from the hummock surface.
**4.1.4. Hypothesis 4: Plant water stress on GPP due to WT deepening below a threshold**
**WTD**

Modelled WTD drawdown below a threshold level also caused rapid peat pore drainage

and low moisture contents in the near surface peat layers which were colonized by most of the
vascular root systems and all of the belowground biomasses of non-vascular mosses (Eqs. D9-
D29). When WTD fell below ~0.1 m from the hollow surface (~0.4 m below the hummock
surface), vertical recharge through capillary rise from the WT was not adequate to maintain near
surface peat moisture. It reduced peat water potential ($\psi_s$) and raised peat hydraulic resistance
($\Omega_s$) (Eq. B9) that supressed root and moss water uptake ($U_w$) (Eq. B6) from desiccated near
surface peat layers (Fig. 1). Since moss $U_w$ entirely depended upon moisture supply from the
near surface layers, reduction in $U_w$ from desiccation of these layers caused reduction in moss
canopy water potential ($\psi_c$) and hence moss GPP (Fig. 1) (Eqs. C1, C4) (Mezbahuddin et al.,
2016). Reduction in root $U_w$ from desiccated near surface layers, however, was offset by
increased root $U_w$ (Eq. B6) from deeper wetter layers, which had higher $\psi_s$ and lower $\Omega_s$, due to
deeper root growth facilitated by enhanced aeration. It enabled the vascular PFTs in *ecosys* to
sustain $\psi_c$, canopy turgor potential ($\psi_t$) (Eq. B4), stomatal conductance ($g_c$) (Eqs. B2, C4) and
hence to sustain increased GPP (Eq. C1) due to higher root nutrient availability and uptake (Fig.
1) (Mezbahuddin et al., 2016). Increased vascular GPP and consequent greater vascular plant
growth further imposed limitations of water, nutrient and light to the modelled non-vascular
PFTs due to interspecific competition and greater shading from the overstorey vascular PFTs.
However, increases in vascular GPP due to enhanced plant nutrient status more than fully offset
the suppression in moss GPP due to moss drying, and greater shading and competition from the
overstorey, thereby causing a net increase in modelled GPP with WTD drawdown (Figs. 7b and
10b-c). This simulation of increased vascular dominance over moss with deepening of WT was
corroborated by several WTD manipulation studies (e.g., Moore et al., 2006, Munir et al., 2014)
in similar peatlands in Alberta that reported increased tree, shrub and herb growths over mosses
with WTD drawdown. However, increase in projected vascular GPP eventually plateaued and it
started to decline when WT fell below ~0.6 m from the hollow surface (~0.9 m below the
hummock surface) (Figs. 10c, e). It was because deeper root $U_w$ (Eq. B6) could no longer offset
suppression of near-surface root $U_w$ when WT fell below threshold WTD, thereby causing lower
$\psi_c$, $\psi_t$ (Eq. B4), $g_c$ (Eqs. B2, C4) and slower $CO_2$ fixation (Eq. C6) (Fig. 1).

These threshold WTD effects on modelled vascular and non-vascular plant water

relations were validated well by testing modelled vs. site measured hourly energy fluxes (latent
and sensible heat) and Bowen ratios, and modelled vs. site measured daily soil moisture at
different depths throughout 2004-2009 as described in Mezbahuddin et al. (2016). Riutta et al.
(2007) measured a reduction in moss productivity due to water limitation when WTD fell below
~0.15 m from the surface in a Finish fen peatland. However, they reported a sustained vascular
GPP during that period indicating no vascular water stress (Riutta et al., 2007). Peichl et al.
(2014) measured a reduction in moss GPP due to moss drying caused by insufficient moisture
supply through capillary rise when WTD fell below 0.25 m from the surface in a Swedish fen.
Reductions in moss GPP due to decreased moss canopy water potentials were also modelled by
Dimitrov et al. (2011) using the same model *ecosys* when WTD fell below 0.3 m from the
hummock surface of a Canadian temperate bog. They, however, found no vascular plant water
stress and hence no reduction in vascular GPP during that period. Similarly, Kuiper et al. (2014)
found reductions in moss productivity with peat drying while vascular productivity sustained in a
simulated drought experiment on a Danish peat.
Continued deepening of WT can also cause vascular plant water stress and hence
reductions in vascular GPP as projected in our drainage simulation (Figs. 10c, e). It can also be
corroborated by field measurements across various northern boreal fen peatlands in Canada and
Sweden. Sonnentag et al. (2010) found a reduction in stomatal conductance ($g_c$) of a canopy that
included tamarack and dwarf birch and a consequent decline in GPP when WT fell below 0.3 m
from the ridge surface at a fen peatland in Saskatchewan. Peichl et al. (2014) also found a
reduction in vascular GPP due to plant water stress when WTD fell below 0.25 m from the
surface in a Swedish fen. The WTD threshold for reductions in vascular GPP in those two field
studies were shallower than that in our modelled projection i.e. ~0.6 m from the hollow surface
(~0.9 m below the hummock surface) (Figs. 10a, c, e) thereby indicating different vertical
rooting patterns determined by specific interactions between hydrologic properties and rooting.
Lafleur et al. (2005) and Schwärzel et al. (2006) found much deeper WTD thresholds for
reductions in vascular transpiration that could negatively affect vascular GPP over a Canadian
pristine bog and a German drained fen respectively. Those WTD thresholds were ~0.65 and ~0.9
m below the surface for the pristine and drained peatland respectively, further indicating the
importance of root-hydrology interactions and the resultant root adaptations, growth and uptake
in determining WTD effects on vascular GPP across peatlands.
**4.2. Divergences between modelled and EC-derived annual GPP, $R_e$ and NEP**
Modelled seasonal and annual GPP and $R_e$ were consistently larger than EC-derived
estimates of GPP and $R_e$ during 2004-2009 (Figs. 7b, c, f, g). Although modelled annual GPP
and $R_e$ were larger than the EC-derived estimates, modelled annual NEP were consistently lower
than the EC gap-filled annual NEP (Fig. 7e). Modelled annual NEP were smaller than EC-
derived estimates because modelled $R_e$ were larger than EC-derived estimates by margins bigger
than by what modelled GPP were larger than EC-derived estimates (Figs. 7f-g). Modelled $R_e$
were larger than EC-derived $R_e$ estimates mainly due to the presence of gap filled night-time $CO_2$
fluxes (=$R_e$) in EC-derived estimates which were smaller than corresponding modelled values. It
was apparent in negative intercepts that resulted from regressions of modelled vs. gap-filled net
$CO_2$ fluxes (Table S1 in supplementary material). Gap filled $R_e$ fluxes were calculated from soil
temperature ($T_s$) at a shallow depth (0.05 m) (Sec. 2.2.2). During night-time and in the winter,
peat at this shallow depth rapidly cooled down and yielded smaller night-time gap-filled $CO_2$
fluxes (Figs. 3, 5 and 6). On the contrary, corresponding modelled $CO_2$ effluxes were affected by
not only the cooler shallow peat layers but also the warmer deeper peat layers and thus were
larger than the gap-filled fluxes (e.g., Figs. 5a, 6h-i). Like modelled $CO_2$ effluxes, chamber
measured $CO_2$ effluxes in cooler nights also did not decline as rapidly as did the corresponding
gap-filled $CO_2$ fluxes as night progressed which further indicated the likely contribution of gap-
filling artifact to $CO_2$ effluxes that were smaller than corresponding modelled fluxes (e.g., Figs.
5b vs. a, 6j-k vs. g-h).
Systematic uncertainties embedded in EC methodology could also have contributed to
EC-derived annual and growing season $R_e$ estimates which were smaller than the modelled
values (Figs. 7c, g). The major uncertainty in the EC methodology is the possible
underestimation of nighttime EC $CO_2$ flux measurements due to poor turbulent mixing under
stable air conditions (Goulden et al., 1997; Miller et al., 2004). On the contrary, modelled
biological production of $CO_2$ by plant and microbial respiration was independent of turbulent
mixing which could thus contribute to modelled $R_e$ that were larger than EC-derived estimates.
Complete energy balance closure in the model as opposed to incomplete (~75%) energy
balance closure in EC measurements would also give rise to modelled evapotranspiration and
GPP values that were larger than EC-derived estimates (Figs. 7b, f) (Mezbahuddin et al., 2016).
Modelled GPP influenced modelled $R_e$ through root exudation and litter fall (Fig. 1).  Therefore,
modelled GPP that were larger than EC-derived estimates would have further contributed to
modelled growing season and annual $R_e$ estimates that were larger than EC-derived estimates
(Figs. 7c, g).
Modelled GPP and hence $R_e$ can also be larger than EC-derived estimates due to
uncertainties in model inputs for soil organic N, N deposition, $N_2$ fixation and any other sources
of N inputs into the modelled ecosystem. In *ecosys*, plant productivity is governed by foliar N
status which is constrained by root N availability and uptake. Our input for organic N into each
modelled peat layer was measured for corresponding depth at the site (Fig. 2). To simulate N
deposition, background wet deposition rates of 0.5 mg ammonium-N, and 0.25 mg nitrate-N per
litre of precipitation which were reported for the study area were used as model inputs. However,
from visual field observations, it was evident that there was a significant amount of nutrient
inflow with the lateral water influxes into this fen peatland from the surrounding upland forests
which was not quantified. To mimic this lateral nutrient inflow, we doubled the background wet
deposition of $NH_4^+$ and $NO_3^-$ as reported for the area and used these as surrogates of lateral
nutrient inflow into the modelled ecosystem. *Ecosys* also included a $N_2$ fixing algorithm which
simulated symbiotic $N_2$ fixation in moss canopies that was reported for the boreal forests (Sec.
2.2.3). We tested the adequacy of these N inputs into the model by comparing modelled leaf N
concentrations against those measured in the field. The modelled foliar N concentrations for the
vascular PFTs corroborated well against site measurements (Table 2). To further examine the
contribution of uncertainty due to model inputs towards the divergence between modelled and
EC-derived seasonal and annual GPP and $R_e$, we performed a sensitivity test where we had a
parallel run without doubling the background N wet deposition rates in the model, hence
simulating no lateral N influx into the modelled ecosystem. Unlike the run with lateral N inflow,
the parallel run without lateral N inflow simulated GPP and $R_e$ which were very close to the EC-
derived estimates. However, the regressions between hourly modelled net $CO_2$ fluxes from the
parallel run and EC-measured hourly net $CO_2$ fluxes gave slopes of ~0.8 indicating under-
simulation of the EC-measured fluxes in the parallel run with no lateral N inflow.
**5. Conclusions**

Our modelling study showed that, when adequately coupled into algorithms of a process-

based ecosystem model, the existing knowledge of peatland eco-hydrological and peat
biogeochemical processes could explain underlying mechanisms that governed WTD effects on
net ecosystem $CO_2$ exchange of a boreal fen. Testing of our hypotheses against EC-measured net
$CO_2$ fluxes, automated chamber fluxes and other biometric measurements at the site revealed that
a drier weather driven WTD drawdown at this boreal fen raised both $R_e$ and GPP due to
improved aeration that facilitated 1) microbial and root $O_2$ availability, energy yields, growth and
decomposition which raised microbial and root respiration; and 2) rapid nitrogen mineralization,
and consequently increased root nitrogen availability and uptake that improved leaf nitrogen
status and hence raised carboxylation (Figs. 1, 7-8) (Table 2). Similar increases in $R_e$ and GPP
with WTD drawdown to a certain depth caused no net WTD drawdown effect on NEP (Fig. 8).
Modelled drainage projection, however, showed that further WTD drawdown caused by either
drainage or climate change induced drying would cause plant water stress and reduce GPP and
hence NEP of this boreal fen (Figs. 9-10). This study further reconciled and mechanistically
explained the WTD effects on seasonal and annual GPP, $R_e$ and hence NEP of this boreal fen
which was previously speculated from EC-derived estimates. However, although modelled $CO_2$
fluxes were validated well by the EC and chamber measured net $CO_2$ fluxes, modelled values of
annual and seasonal GPP and $R_e$ were consistently larger than the EC-derived estimates (Table 1)
(Figs. 7-8) (Sec. 4.2). These discrepancies between modelled and EC-derived GPP and $R_e$
estimates also raised a potential research question of whether or not to use more robust process-
based estimates of these peatland C balance components instead of empirically modelled EC-
derived estimates that might not include some of the above discussed offsetting feedbacks in
peatland eco-hydrology and biogeochemistry.

The model algorithms that were used in this study represented coupled feedbacks among

ecosystem processes that governed carbon, water, energy, and nutrient (N, P) cycling in a
peatland ecosystem. These feedbacks were thus not parameterized for this particular boreal fen
peatland site. Instead, the modelled boreal fen was simulated from peatland specific model inputs
for weather, soil, and vegetation properties that had physical meaning and were quantifiable at
the site (Fig. 2). These modelled process level interactions were also validated by corroborating
modelled outputs against site measurements. On the contrary, most of the current peatland C
models use scalar functions to represent these feedbacks and so those model algorithms have to
be parameterized for each peatland site. Therefore, the modelling approach as described in this
study should be more robust than the scalar feedback approach while assessing WTD effects on
peatland C balance under contrasting peat types, climates and hydrology, or under unknown
future climates. These process level feedbacks are also scalable once the peatlands of interest are
defined within the modelled landscapes by scalable model inputs for weather; peat hydrological,
physical, and biological properties; and plant functional types (Sec. 2.2.3). Current global land
surface models either lack or have very poor representation of these feedbacks which is thus far
limiting our large scale predictive capacity on WTD effects on boreal peatland C stocks. This
modelling exercise would thus provide valuable information to improve representation of these
feedbacks into next generation land surface models. Therefore, the insights gained from this
modelling study should be a significant contribution to our understanding and apprehension of
how peatlands would behave with changing hydrology under future drier and warmer climates.

## Code availability

**Code availability**
The *ecosys* model codes are listed in equation forms and sufficiently described in the
supplementary material. The model codes that were written in FORTRAN will also be available
on request from either symon.mezbahuddin@gov.ab.ca or rgrant@ualberta.ca.
**Data availability**

Field data that were used to validate model outputs are available at

http://fluxnet.ornl.gov/site/292.
**Author contribution**

960       M. Mezbahuddin contributed to the model code modification and development, designing

modelling experiment, simulation, validation, and analyses of modelled outputs. R. F. Grant is
the original developer of the model *ecosys* and also contributed into simulation design and model
runs. L. B. Flanagan was site principal investigator who led the collection, and quality control of
the field data that were used to validate model outputs. M. Mezbahuddin wrote the manuscript
with significant contributions from R. F. Grant and L. B. Flanagan.
**Acknowledgments**

Computing facilities for the modelling project was provided by Compute Canada,

Westgrid, and University of Alberta. Funding for the modelling project was provided by several
research awards from Faculty of Graduate Studies and Research and Department of Renewable
Resources of University of Alberta and a Natural Sciences and Engineering Research Council
(NSERC) of Canada discovery grant. The field research was carried out as part of the Fluxnet-
Canada Research Network and the Canadian Carbon Program and was funded by grants to
Lawrence B. Flanagan from NSERC, Canadian Foundation for Climate and Atmospheric
Sciences, and BIOCAP Canada.

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

 **Figure captions**

**Fig. 1.** Schematic diagram of *ecosys* algorithms representing coupled key eco-physiological and
biogeochemical (aerobic and anaerobic) processes, and plant water relations of a typical boreal
fen peatland ecosystem that are affected by water table depth (WTD) fluctuation. $\Psi_a$, $\Psi_c$, and $\Psi_s$
=atmospheric, canopy and soil water potentials; $\Psi_r$=vascular root or non-vascular belowground
water potential; $r_c$=canopy stomatal resistance [=1/canopy stomatal conductance ($g_c$)]; $\Omega_s$=soil
hydraulic resistance; $\Omega_r$=hydraulic resistance to water flow through plants; OM=organic matter;
DOC, DON, DOP=dissolved organic carbon, nitrogen, and phosphorus; POC = particulate
organic C; and POM = particulate organic matter
**Fig. 2.** Layout for *ecosys* model run to represent biological, chemical and hydrological
characteristics of a Western Canadian fen peatland. Figure is not drawn to scale. $D_{humm}$ = depth
to the bottom of a layer from the hummock surface; $D_{holl}$ = depth to the bottom of a layer from
the hollow surface; TOC = total organic C (Flanagan and Syed, 2011); TN = total nitrogen
(Flanagan and Syed, 2011); TP = total phosphorus (Flanagan and Syed, 2011); CEC = Cation
exchange capacity (Rippy and Nelson, 2007); the value for pH was obtained from Syed et al.
(2006); $WTD_x$ = external reference water table depth representing average water table depth of
the adjacent ecosystem; $L_t$ = distance from modelled grid cells to the adjacent watershed over
which lateral discharge / recharge occurs
**Fig. 3. (a, c, e, g, i, k)** 3-day moving averages of modelled and EC-gap filled net ecosystem
productivity (NEP) (Flanagan and Syed, 2011), and **(b, d, f, h, j, l)** hourly modelled and half
hourly measured water table depth (WTD) (Syed et al., 2006; Cai et al., 2010; Long et al., 2010;
Flanagan and Syed, 2011) from 2004 to 2009 at a Western Canadian fen peatland. A positive
NEP means the ecosystem is a carbon sink and a negative NEP means the ecosystem is a carbon
source. A negative WTD represents a depth below hummock/hollow surface and a positive WTD
represents a depth above hummock/hollow surface
**Fig. 4.** Half hourly measured **(a)** incoming shortwave radiation, and **(b)** air temperature ($T_a$); and
**(c)** hourly modelled and half hourly measured water table depth (WTD) (Syed et al., 2006; Cai et
al.; 2010, Long et al.; 2010, Flanagan and Syed, 2011) during August of 2005, 2006 and 2008 at
a Western Canadian fen peatland. A negative WTD represents a depth below hummock/hollow
surface and a positive WTD represents a depth above hummock/hollow surface. Grey arrows
indicate nights with similar temperatures
**Fig. 5. (a)** Half hourly EC-gap filled (Flanagan and Syed, 2011) and hourly modelled ecosystem
net $CO_2$ fluxes, **(b)** half hourly automated chamber measured (Cai et al., 2010) and hourly
modelled understorey and soil $CO_2$ fluxes, and **(c)** hourly modelled soil $CO_2$ and $O_2$ fluxes
during August of 2005, 2006 and 2008 at a Western Canadian fen peatland. No chamber $CO_2$
flux measurement was available for 2008. Bars represent standard errors of means of chamber
$CO_2$ fluxes ($n=9$). A negative flux represents an upward flux or a flux out of the ecosystem and a
positive flux represents a downward flux or a flux into the ecosystem. Grey arrows indicate
nights with similar temperatures (Fig. 4)
**Fig. 6. (a-c)** Half hourly observed air temperature ($T_a$), **(d-f)** hourly modelled and half hourly
observed water table depth (WTD) (Syed et al., 2006; Cai et al., 2010; Long et al., 2010;
Flanagan and Syed, 2011), **(g-i)** half hourly EC-gap filled (Flanagan and Syed, 2011) and hourly
modelled ecosystem net $CO_2$ fluxes, **(j-l)** half hourly automated chamber measured (Cai et al.,
2010) and hourly modelled understorey and soil $CO_2$ fluxes during July-August of 2005, 2006
and 2008 at a Western Canadian fen peatland. No chamber $CO_2$ flux measurement was available
for 2008. Bars represent standard errors of means of chamber $CO_2$ fluxes ($n$=9). A negative flux
represents an upward flux or a flux out of the ecosystem and a positive flux represents a
downward flux or a flux into the ecosystem. A negative WTD represents a depth below
hummock/hollow surface and a positive WTD represents a depth above hummock/hollow
surface
**Fig. 7.** Modelled and EC-derived (Flanagan and Syed, 2011) growing season (May-August)
sums of **(a)** net ecosystem productivity (NEP), **(b)** gross primary productivity (GPP), and **(c)**
ecosystem respiration ($R_e$) during 2004-2009; **(d)** observed mean growing season air temperature
($T_a$) and measured and modelled average growing season water table depth (WTD) during 2004-
2009; Modelled and EC-derived (Flanagan and Syed, 2011) annual sums of **(e)** NEP, **(f)** GPP,
and **(g)** $R_e$ during 2004-2008; and **(h)** observed mean annual $T_a$ and measured and modelled
average WTD during ice free periods (May-October) of 2004-2008 at a Western Canadian fen
peatland. A negative WTD represents a depth below hollow surface and a positive WTD
represents a depth above hollow surface. A positive NEP means the ecosystem is a carbon sink.
Annual modelled vs. EC-gap filled NEP, GPP, $R_e$ estimates for 2009 were not compared due to
the lack of flux measurements from September to December in that year.
**Fig. 8.** Regressions ($P$<0.001) of growing season (May-August) sums of modelled and EC-
derived (Flanagan and Syed, 2011) **(a)** net ecosystem productivity (NEP), **(b)** gross primary
productivity (GPP), and **(c)** ecosystem respiration ($R_e$) on growing season averages of modelled
and observed water table depth (WTD) during 2004-2009; and regressions ($P$<0.001) of annual
sums of modelled and EC-derived (Flanagan and Syed, 2011) **(d)** NEP, **(e)** GPP and **(f)** $R_e$ on
average modelled and measured WTD during ice free periods (May-October) of 2004-2008 at a
Western Canadian fen peatland. A negative WTD represents a depth below hollow surface and a
positive WTD represents a depth above hollow surface. A positive NEP means the ecosystem is
a carbon sink
**Fig. 9. (a)** Observed, simulated (real-time simulation) and projected (drainage simulation)
average growing season (May-August) water table depth (WTD); EC-derived, simulated and
projected growing season sums of **(b)** net ecosystem productivity (NEP), **(c)** gross primary
productivity (GPP), and **(d)** ecosystem respiration ($R_e$); and regressions ($P<0.001$) of simulated
and projected sums of **(e)** NEP, **(f)** GPP, and **(g)** $R_e$ on simulated and projected average growing
season WTD during 2004-2009 at a Western Canadian fen peatland. A negative WTD represents
a depth below hollow surface and a positive WTD represents a depth above hollow surface. A
positive NEP means the ecosystem is a C sink
**Fig. 10.** Simulated (real-time simulation) and projected (drainage simulation) **(a)** average
growing season (May-August) water table depth (WTD), **(b)** growing season sums of non-
vascular (moss) gross primary productivity (GPP), and **(c)** growing season sums of vascular
GPP; and regressions ($P<0.001$) of simulated and projected sums of **(d)** non-vascular GPP, and
**(e)** vascular GPP on simulated and projected average growing season WTD during 2004-2009 at
a Western Canadian fen peatland. A negative WTD represents a depth below hollow surface and
a positive WTD represents a depth above hollow surface

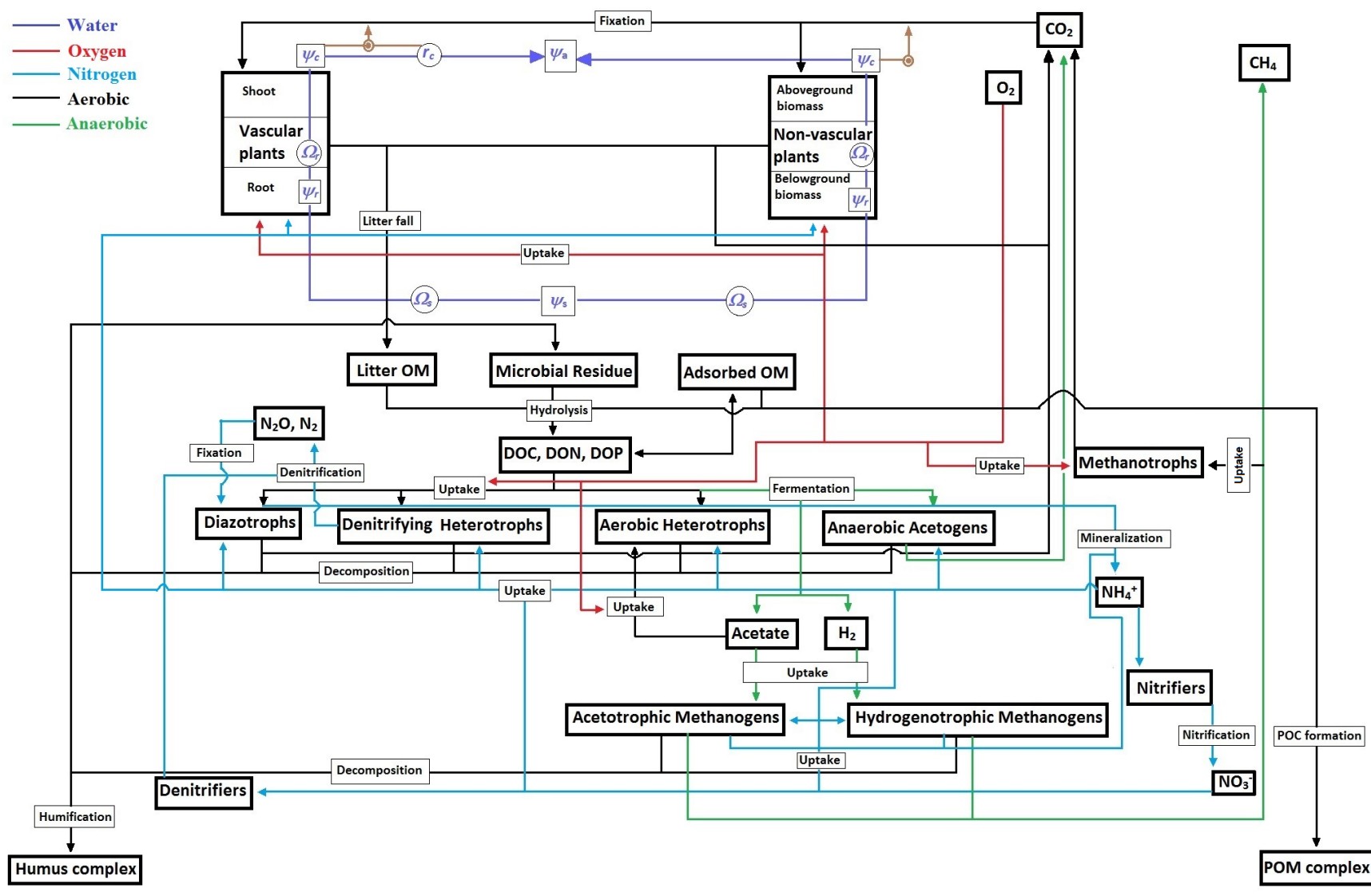


**Fig. 1.**

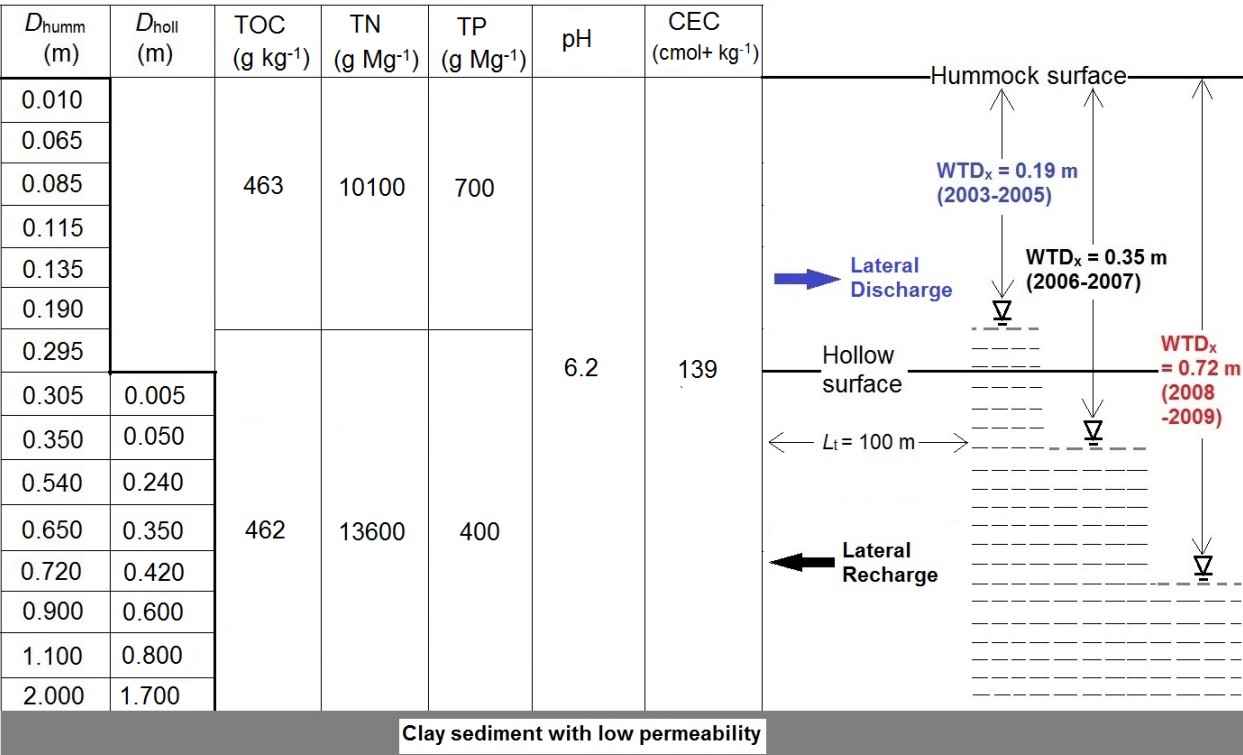


**Fig. 2.**

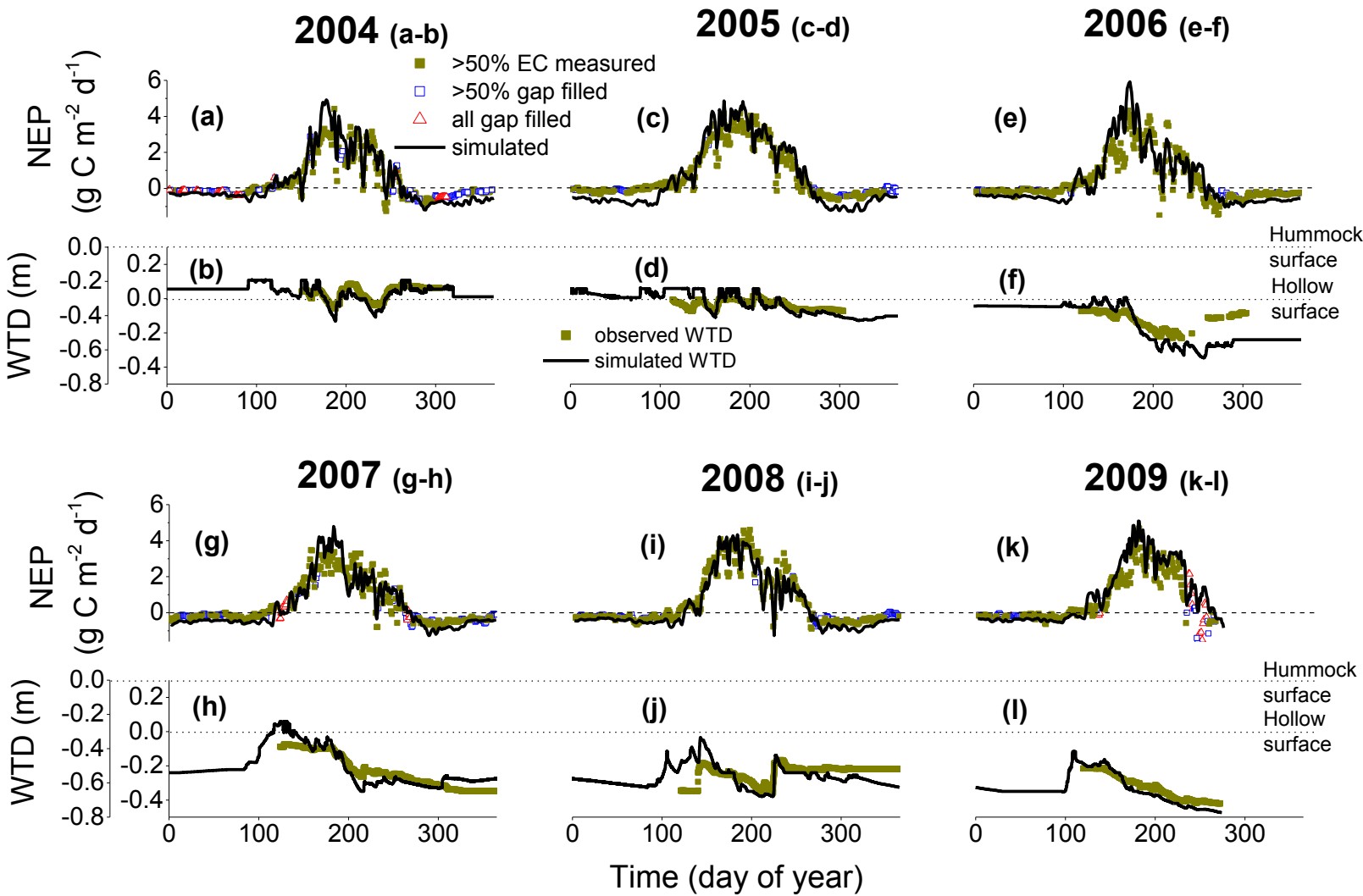


**Fig. 3.**

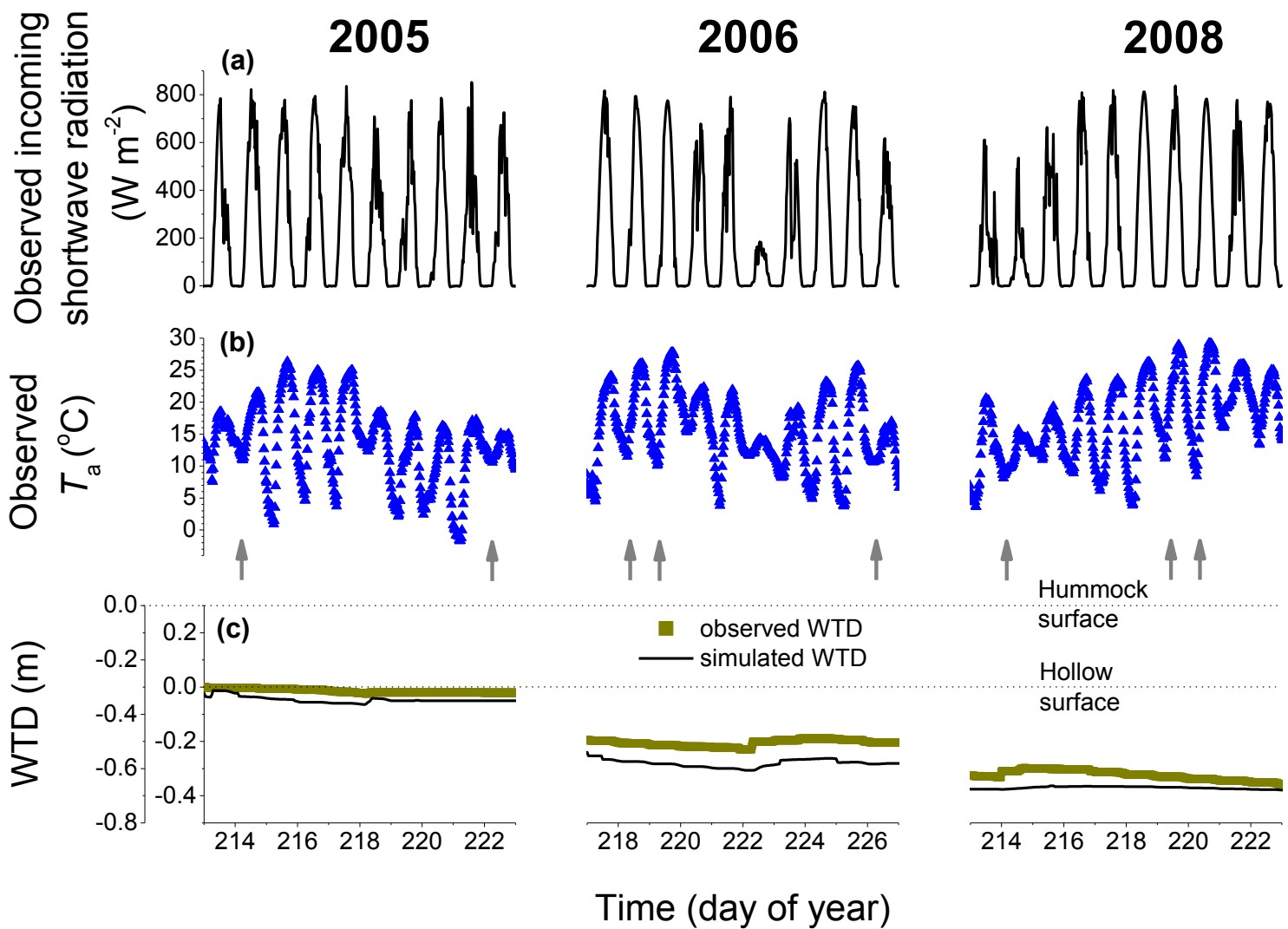


Fig. 4.


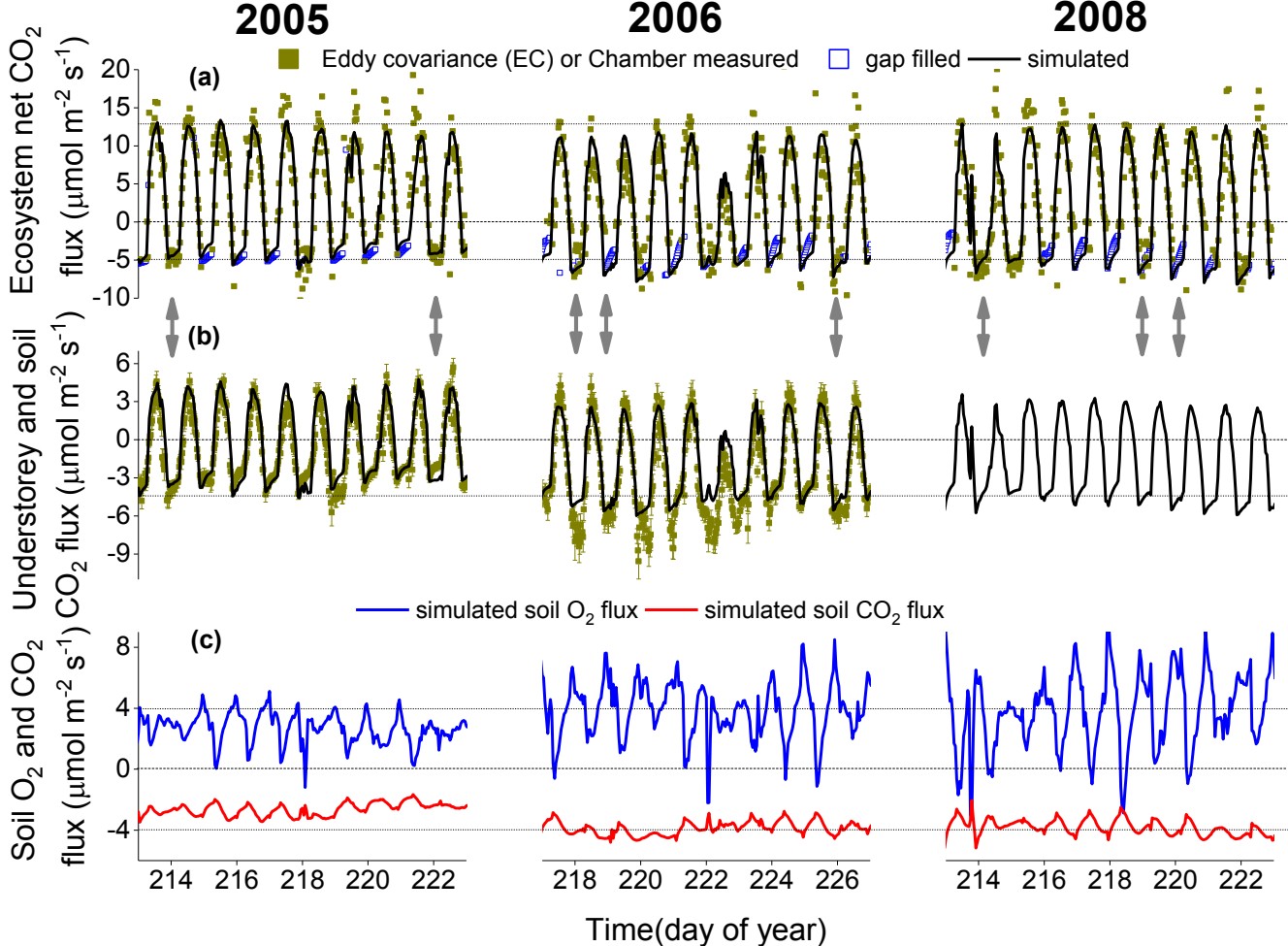


**Fig. 5.**

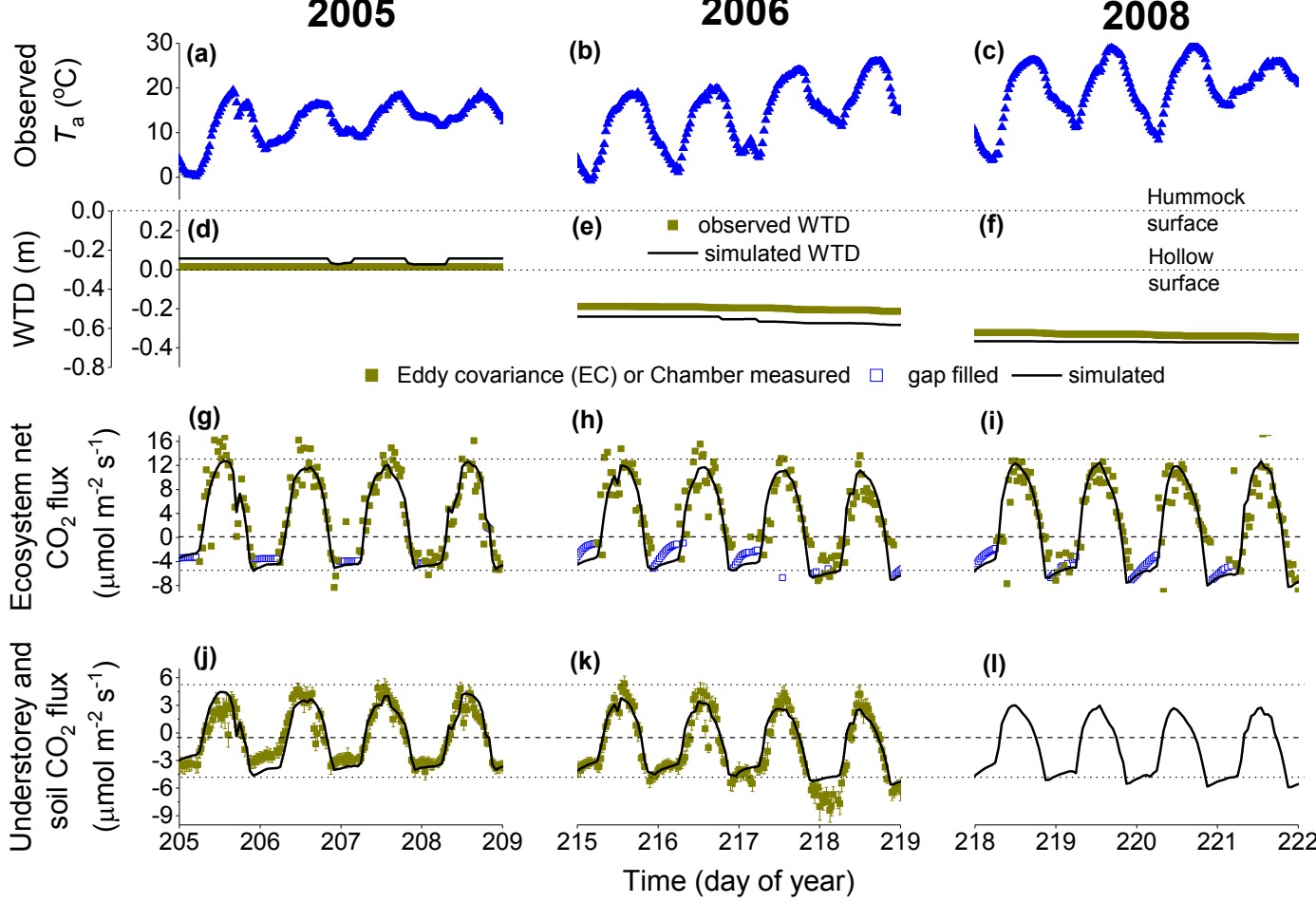

**Fig. 6.**

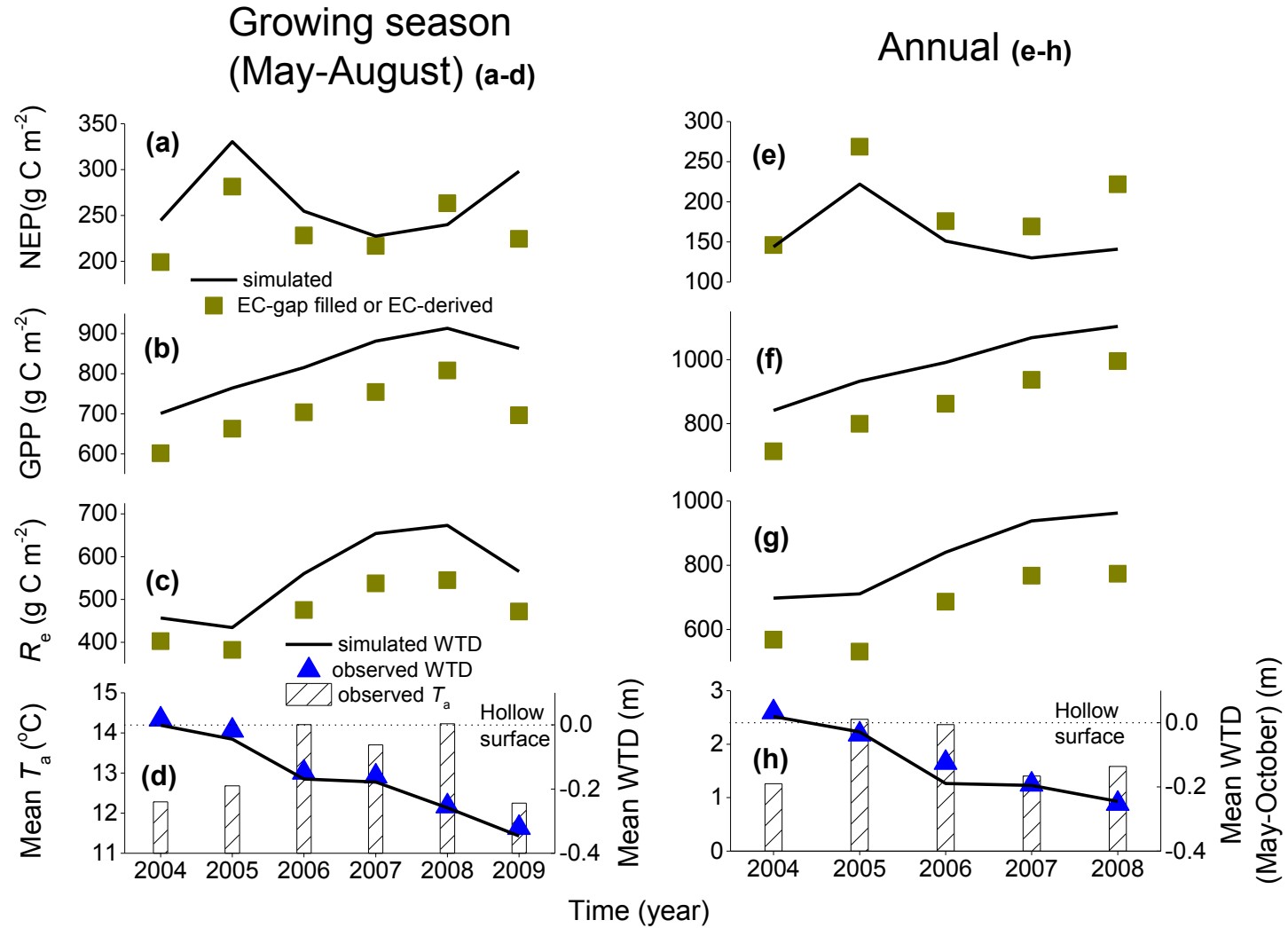


**Fig. 7.**

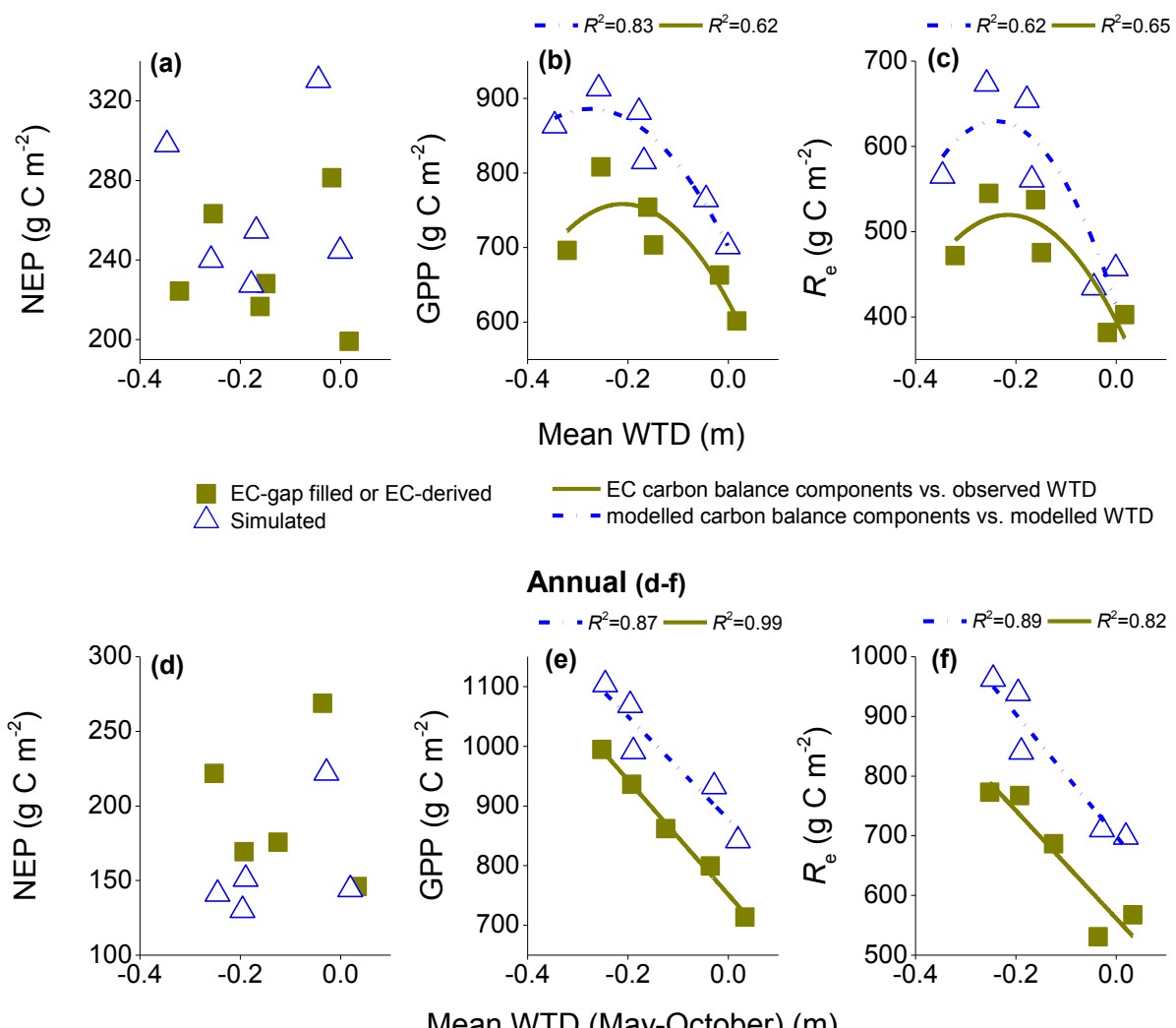


**Fig. 8.**

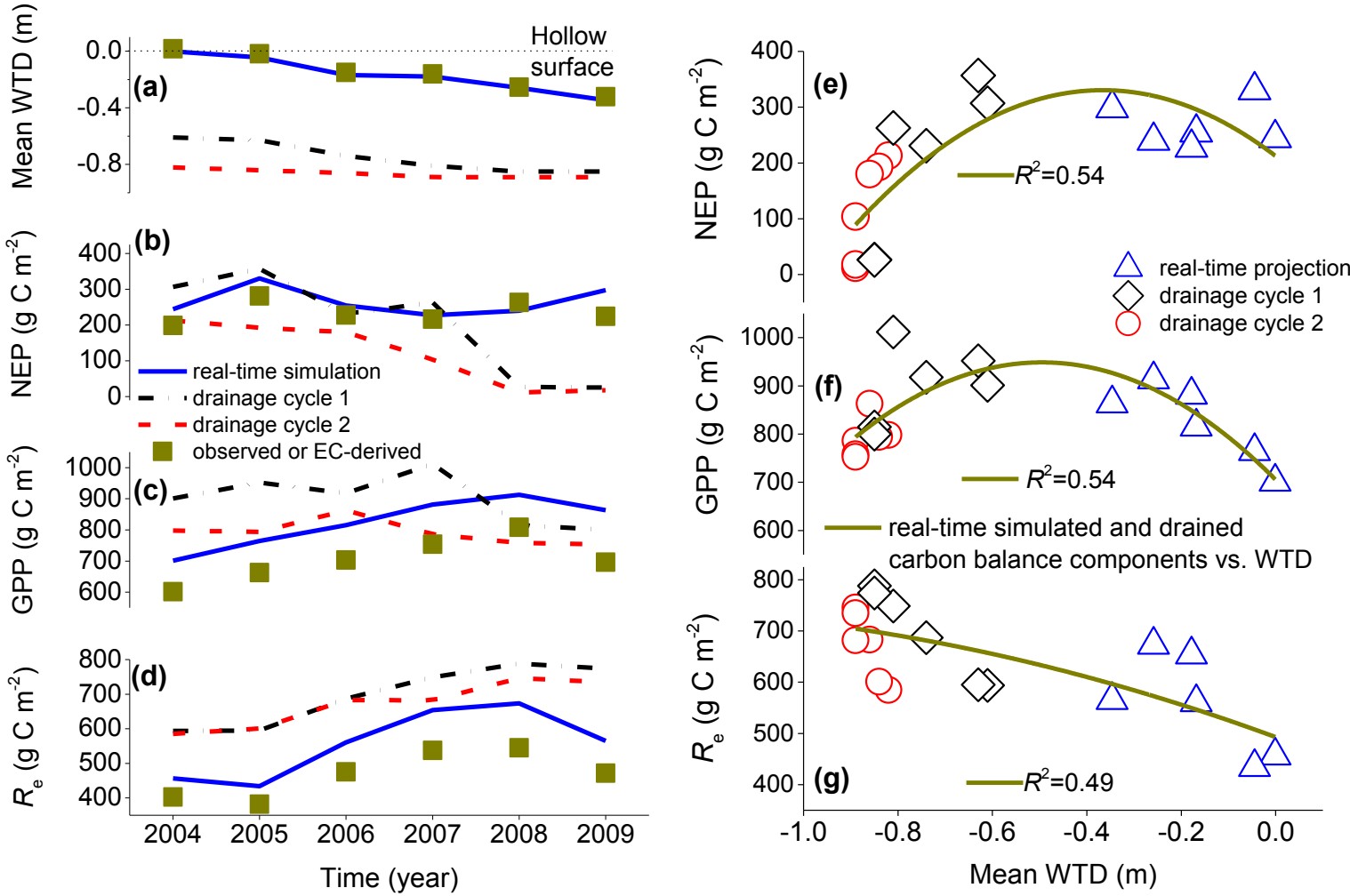


**Fig. 9.**

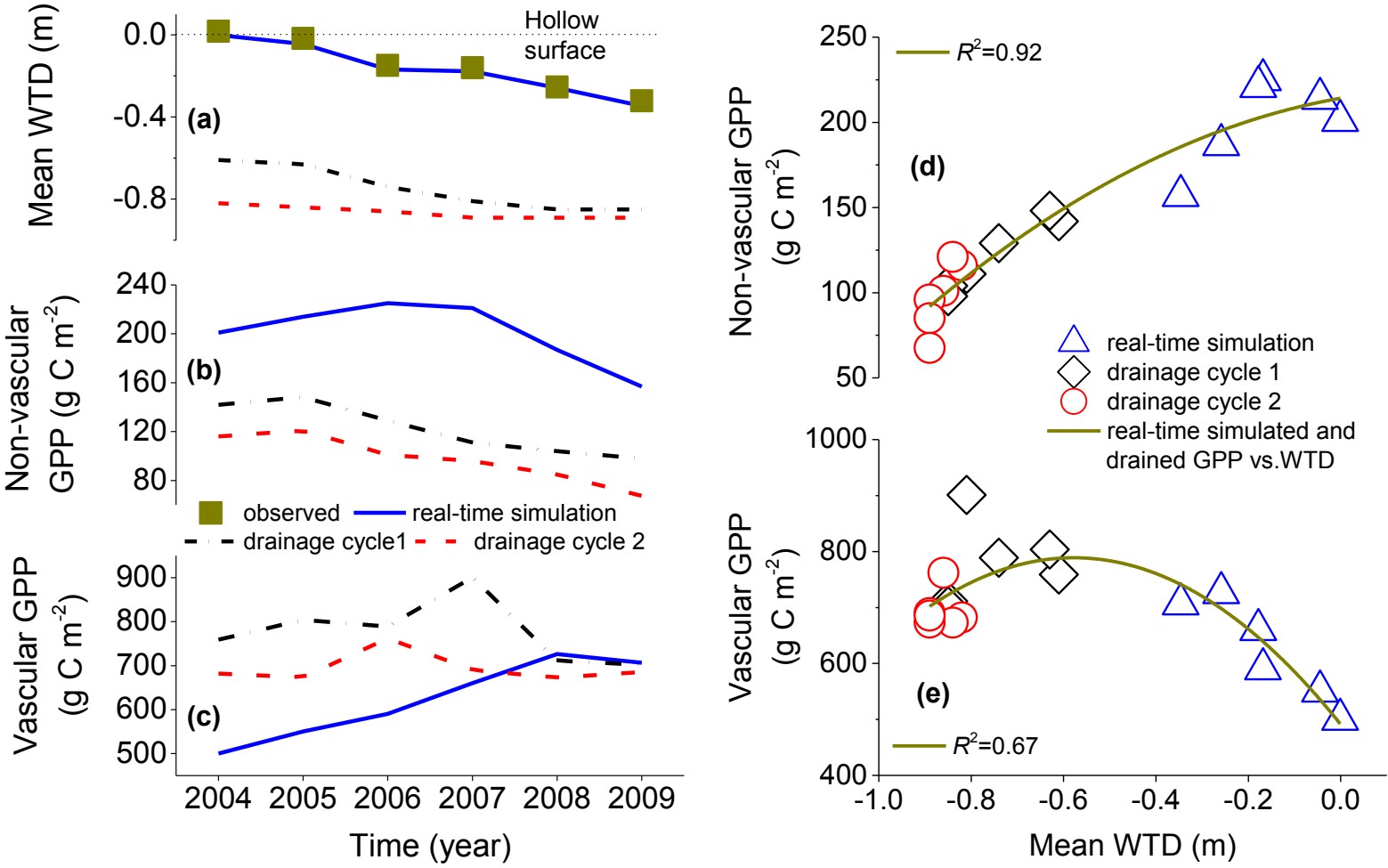


**Fig. 10.**

**Table 1.** Statistics from regressions between hourly modelled and measured net $CO_2$ fluxes from 2004-2009 at a Western Canadian fen peatland

**(a)** Regressions of modelled vs. EC measured (recorded at $u* > 0.15$ m s$^{-1}$) net ecosystem $CO_2$ fluxes over whole years of 2004-2008[a]

| Year | Total annual precipitation (mm) | $n$ | $a$ | $b$ | $R^2$ | RMSE ($\mu$mol m$^{-2}$ s$^{-1}$) | RMSRE ($\mu$mol m$^{-2}$ s$^{-1}$) |
|---|---|---|---|---|---|---|---|
| 2004 | 553 | 5034 | 0.08±0.03 | 1.10±0.01 | 0.81 | 1.58 | 1.92 |
| 2005 | 387 | 5953 | 0.07±0.03 | 1.03±0.01 | 0.82 | 1.68 | 1.99 |
| 2006 | 465 | 6012 | 0.07±0.03 | 1.08±0.01 | 0.79 | 1.68 | 1.98 |
| 2007 | 431 | 5385 | 0.06±0.03 | 0.99±0.01 | 0.79 | 1.83 | 2.09 |
| 2008 | 494 | 5843 | -0.01±0.02 | 0.98±0.01 | 0.84 | 1.63 | 2.02 |

**(b)** Regressions of modelled vs. EC measured (recorded at $u* > 0.15$ m s$^{-1}$) net ecosystem $CO_2$ fluxes over growing seasons (May-August) of 2004-2009

| Year | Total growing season precipitation (mm) | $n$ | $a$ | $b$ | $R^2$ | RMSE ($\mu$mol m$^{-2}$ s$^{-1}$) | RMSRE ($\mu$mol m$^{-2}$ s$^{-1}$) |
|---|---|---|---|---|---|---|---|
| 2004 | 287 | 2043 | 0.55±0.07 | 1.05±0.01 | 0.78 | 2.27 | 2.55 |
| 2005 | 276 | 2200 | 0.82±0.07 | 0.98±0.01 | 0.79 | 2.50 | 2.74 |
| 2006 | 253 | 2107 | 0.48±0.07 | 1.06±0.01 | 0.78 | 2.36 | 2.76 |
| 2007 | 237 | 1822 | 0.65±0.07 | 0.93±0.01 | 0.75 | 2.91 | 3.06 |
| 2008 | 276 | 2070 | 0.32±0.07 | 0.96±0.01 | 0.82 | 2.45 | 2.85 |
| 2009 | 138 | 1870 | 0.76±0.08 | 1.01±0.01 | 0.81 | 2.27 | 2.83 |

**(c)** Regressions of modelled vs. measured chamber net $CO_2$ fluxes (understorey vegetation and soil $CO_2$ fluxes) over ice free periods (May-October) of 2005-2006

| Year | Mean May-October WTD (m) | | $n$ | $a$ | $b$ | $R^2$ | RMSE (μmol m$^{-2}$ s$^{-1}$) | RMSRE (μmol m$^{-2}$ s$^{-1}$) |
|------|---------|--------|-----|-----|-----|-------|------|-------|
| | Modelled | Measured | | | | | | |
| 2005 | 0.33 | 0.34 | 3285 | 0.43±0.02 | 1.05±0.01 | 0.68 | 1.19 | 2.38 |
| 2006 | 0.48 | 0.42 | 3855 | 0.31±0.03 | 0.85±0.01 | 0.71 | 1.44 | 3.25 |

WTD = water table depth below the hummock surface; (*a, b*) from simple linear regressions of modelled on measured (± standard errors), and $R^2$ = coefficient of determination; RMSE = root mean square for errors from simple linear regressions of measured on simulated; RMSRE= root mean square for random errors in measurements; RMSRE for eddy covariance (EC) measurements were estimated by inputting EC $CO_2$ fluxes recorded at $u*$ (friction velocity) > 0.15 m s$^{-1}$ into algorithms for estimation of random errors due to EC $CO_2$ measurements developed for forests by Richardson et al. (2006); [a] whole year modelled vs. EC net $CO_2$ flux regression for 2009 could not be done due to the lack of flux measurements from September to December in that year.

**Table 2.** Effects of water table depth (WTD) drawdown on components of ecosystem carbon and nutrient cycles of a Western

Canadian fen peatland

| | | Modelled | Eddy covariance-derived/biometrically measured at the site [a] | Values from other studies in similar peatlands |
|---|---|---|---|---|
| Growing season (May to August) mean WTD drawdown from 2004 to 2009 | | from 0.3 m below the hummock surface (at the hollow surface) in 2004 to 0.65 m below the hummock surface (0.35 below the hollow surface) in 2009 | from 0.32 m below the hummock surface (0.02 m below the hollow surface) in 2004 to 0.62 m below the hummock surface (0.32 m below the hollow surface) in 2009 | |
| Rate of increase in annual $R_e$ with each 0.1 m of WTD drawdown | | 0.26 $\mu$mol $CO_2$ m$^{-2}$ s$^{-1}$ | 0.16 $\mu$mol $CO_2$ m$^{-2}$ s$^{-1}$ | 0.32$\pm$0.27 $\mu$mol $CO_2$ m$^{-2}$ s$^{-1}$ [b] |
| Rate of increase in growing season GPP with each 0.1 m of WTD drawdown | | 0.39 $\mu$mol $CO_2$ m$^{-2}$ s$^{-1}$ | 0.22 $\mu$mol $CO_2$ m$^{-2}$ s$^{-1}$ | (1) 0.32$\pm$0.15 $\mu$mol $CO_2$ m$^{-2}$ s$^{-1}$ [b]  (2) 0.47$\pm$0.06 $\mu$mol $CO_2$ m$^{-2}$ s$^{-1}$ [c] |
| | Black Spruce | 14.3 g N kg$^{-1}$ C | 12.4$\pm$0.6 g N kg$^{-1}$ C | |

| | | | | |
|---|---|---|---|---|
| Leaf nitrogen concentration (mid-July 2004) | Tamarack | 32.1 g N kg$^{-1}$ C | 32.6±1.4 g N kg$^{-1}$ C | |
| | Dwarf Birch | 37.4 g N kg$^{-1}$ C | 41.4±1.8 g N kg$^{-1}$ C | |
| Leaf nitrogen (N) to phosphorus (P) ratio (mid-July 2004) | Black Spruce | 6.6:1 | 7.1:1 | |
| | Tamarack | 5.2:1 | 6.3:1 | |
| | Dwarf Birch | 4.8:1 | | |
| Increase in foliar nitrogen concentrations with WTD drawdown | Black Spruce | from 14.3 to 17.2 g N kg$^{-1}$ C with the growing season WTD drawdown from 0.3 m below the hummock surface in 2004 to 0.65 m below the hummock surface in 2009 | | (1) from 20.8±0.6 to 27±0.4 g N kg$^{-1}$ C with a WTD drawdown from 0.24 to 0.7 m below peat surface [d] <br><br> (2) from 18.7±0.05 to 21.3±0.06 g N kg$^{-1}$ C for a WTD drawdown by 0.4-0.5 m [e] |
| | Tamarack | from 32.1 to 36.7 g N kg$^{-1}$ C with the growing season WTD drawdown from 0.3 m below the hummock surface in 2004 to 0.65 m below the hummock surface in 2009 | | (1) from 41.4±0.4 to 66.2±1.2 g N kg$^{-1}$ C for a WTD drawdown from 0.24 to 0.7 m below peat surface [d] <br><br> (2) from 35.9±0.1 to 41.7±0.18 g N kg$^{-1}$ C for a WTD drawdown by 0.4-0.5 m [e] |
| | Dwarf Birch | from 37.4 to 45.1 g N kg$^{-1}$ C with the growing season WTD drawdown from 0.3 m below the hummock surface in 2004 to 0.65 | | |

| | | | |
|---|---|---|---|
| | | m below the hummock surface in 2009 | |
| Increase in maximum rooting depth with WTD drawdown | Black spruce | from 0.35 to 0.65 m below the hummock surface with the growing season WTD drawdown from 0.3 m below the hummock surface in 2004 to 0.65 m below the hummock surface in 2009 | from 0.1 to 0.6 m below hummock surface with a WTD drawdown from 0.1 to 0.7 m below hummock surface [f] |
| | Tamarack | from 0.35 to 0.65 m below the hummock surface with the growing season WTD drawdown from 0.3 m below the hummock surface in 2004 to 0.65 m below the hummock surface in 2009 | from 0.2 to 0.6 m below hummock surface with a WTD drawdown from 0.1 to 0.6 m below hummock surface [f] |

GPP=gross primary productivity; $R_e$ = ecosystem respiration; [a] Syed et al. (2006), Flanagan and Syed (2011); [b] Peichl et al. (2012) for a Swedish fen; [c] Ballantyne et al. (2014) for a Michigan fen peatland complex that has similar peat and plant functional types as our study site; [d] Choi et al. (2007) for a central Alberta fen peatland located ~350 km to the southwest of the study site; [e] Macdonald and Lieffers (1990) for a northern Alberta fen peatland located ~250 km to the northwest of the study site; [f] Lieffers and Rothwell (1987) for a northern Alberta fen peatland located ~250 km to the northwest of the study site