# Peer review of "Coupled eco-hydrology and biogeochemistry algorithms enable simulation of water table"

_Biogeosciences, 2017_

## Referee Comment (RC1) · Anonymous Referee #1 · 19 May 2017

This paper compares the results of the ecosystem biophysical model ecosys against field measurements of environmental variables (primarily water table depth – WTD) and carbon fluxes measured by eddy covariance (EC) at a treed peatland in western Canada over a 5 year time frame when WT was decreasing at the site. The model, ecosys, is a very sophisticated tool and has been widely applied in the past against many different ecosystem types, with success. It is fair to say that is among the top ranked platforms for simulating ecosystem functioning. With that said, the purpose of this particular paper is a bit foggy. The EC flux measurements from this site, including the time series over which the WTD had declined, have been clearly reported in

previous literature, as has been cited in this study. Therefore, is the purpose of this study 1) to simply to test if ecosys can simulate the trend in measured EC fluxes over the study period, or 2) to use ecosys to explain the behaviour of the EC-fluxes, which cannot be obtained from most common EC and environmental measurements? The paper seems to do a bit of both, but the main objective is not clear. However, given the extensive testing of ecosys at other peatlands and other ecosystems, the former seems to be quite a weak objective. The latter is more scientifically interesting, but that is not the way the paper is set out. There are 4 operating hypotheses (not repeated here). These are not stated in terms of what is reflected in the EC-derived measurements, which seems to be the main thrust of the paper from the rest of the introduction, but rather in terms of biophysical processes that will take place in the model. Hence, it is a source of some of the confusion about the purpose of this paper. It would be nice to see an attempt to improve the introduction with a clearer purpose. Another concern about the present manuscript is that there is a lot of attention to how ecosys performs in simulating the WTD. It seems to me that this topic was adequately covered in the a previous paper, Mezbahuddin et al. (2016), so why do we need the emphasis here. I have a small worry about the comparison of ecosys modelled fluxes against gap-filled data (especially nighttime (Re) fluxes – section 3.3). Since the gap-filling is a model itself - now we are comparing one model against another. I realize there is a discussion of how this may have affected the comparison, but that does little to convince readers that the comparison of modelled and measured data is sound. Why not just compare half hours where measured data were available to test ecosys, if that is the point of the paper (see above). Section 4.2 Divergence between modeled and EC-derived fluxes makes some interesting points, but as it stands very little of this has been tested or analyzed in any detail, so we really don't know what the source of the discrepancy is. It would be nice to see some attempt or suggestions as to how to hone in on the most likely causes of the discrepancy. You seem to suggest the E-derived measurements are wrong, which may well be, but I am not sure that the model is not without fault. The Conclusion section is not really a conclusion. First, much of it is simply a re-statement
of the main findings. Second, it suffers the same problems as the objective of the study namely, not really being clear. The final statements about the value and application of the ecosys model, while possibly true, seem a little self-serving.

Finally, the manuscript needs a good editing, although the writing is such that it is understandable, there are many awkward statements/phrases, issues with tense, or grammatical errors that could be addressed to improve the manuscripts readability. I have pointed out some of these in the minor points below, but there are several others.

Overall, this could be quite a useful contribution, especially if cast in the role of using ecosys to help understand the pattern and responses of EC-derived fluxes over time, something that is hard to get from just EC and environmental measurements, rather than just another test of the ecosys algorithms at another peatland site.

Minor Issues: 1. Line 81-82 & line 87, Lafleur et al. (2005) reference is inappropriate here, they discuss Re not GPP. Also relevant on lines 908-911. 2. Lines 95-97, as above this reference does not discuss a threshold for WTD and GPP, perhaps another reference by this author? 3. Lines 98-107, the start of this paragraph is poorly worded. First one does not start a new paragraph with the word 'therefore". The sentence beginning "So, to adequately predict ... " is awkward and doesn't quite read right. As with the next sentence – the phrase "do not have prognostic WTD dynamics that prevent simulation" is confusing and awkward. 4. Lines 314-317, these two sentences that describe the simulated mosses are very difficult to understand, some revision for clarity is needed. Should be put in terms of what a real moss is and where it grows. 5. Line 447, the word diurnal here is incorrect: Table 1 compares instantaneous half hour fluxes. Diurnal suggests some course of measurements over the daytime. 6. Line 457-58, the sentence here about 2009 is not needed here; simply add it as a foot note to the Table. 7. Line 478-81, this sentence is somewhat heuristic; the Figure certainly does not show these components. I think it is adequate just to say the model simulated measured WTD well. 8. Line 536, word 'also' is not needed 9. Line 537, what does 'lt' refer to? 10. Line 538-41, this long sentence is somewhat awkward and doesn't really
say anything new. 11. Lines 559-61, the sentence here seems to be missing a word or words, does not read well. 12. Lines 55-570, you seem to miss an opportunity here. You describe how warming does not stimulate Re when water table was high in 2005, and how it is stimulated by warming in low WT years (2006 and 2008), yet given the sophistication in ecosys there is not real explanation of why this works the way it does, what is the biogeochemical functioning that does or does not stimulate Re under low and high WTs respectively? Further down you describe the mechanisms associated with GPP, why not the same with Re? 13. Line 632-33, this was mentioned above (#6), no need to repeat it here. 14. Lines 664-69 Section 3.5, You state this drainage experiment "... would 667 also serve as a climate change analog in providing us insight into how potential WTD drawdown 668 under future drier and warmer climates would affect boreal peatland GPP, Re and hence NEP." I don't see how, as the atmospheric changes of higher temperatures and perhaps higher VPD are not included. I think it is fair to say this simulation represents the effects of WT drawdown only. 15. Lines 673-74. I don't think you need this sentence, it is rather obvious that changes would occur and you have a lot of words following to describe them. 16. Lines 683-696, is this enhanced evapotranspiration coming from the tree cover or ground vegetation or both? 17. Lines 848-51, should note here that his Dimitrov et al. 2010 study was on a temperate bog not a fen.

---

## Referee Comment (RC2) · Anonymous Referee #2 · 26 May 2017

General Comments The manuscript addresses impacts on CO2 fluxes from changes in water table depth by using the ecosys model. The model is tested with eddy covariance and chamber CO2 fluxes from a boreal peatland field site. The manuscript is dense throughout and requires very careful attention on the part of the reader to follow along. While the ecosys model is complex, how this paper is written exacerbates the complexity of the model. Right now, this paper would be an incredibly useful guide to someone wanting to run the ecosys model themselves, but lacks a clear and story that is supported by the results of this work.

The major issue I have with this work is the relative complexity of the ecosys model next to the small amount of observed data that the model is compared with. Since ecosys has so many moving parts "under the hood", I can't say that I'm surprised at all to see it match data as well as it does. A good fit to observed data is not a new finding itself, and in a broader sense, the research questions aren't new. In fact, there is a good amount of overlap with Mezbahuddin et al. (2016), as brought up by Referee 1. So, I'm stuck reading through a dense description of a complex model, and at the end, it's compared with limited amounts of data that itself is modeled. The main conclusions seem to be focused on internal modeled variables within the ecosys model that have zero comparison to data. The major conclusions are changes in modeled O2 diffusion, N mineralization rates, nutrient availability, microbial concentrations, plant functional type GPP. These results, as currently presented, are simply not supported by comparing to net CO2 fluxes.

The authors state in the conclusions that "These modelling hypotheses were also corroborated by various field, laboratory and modelling studies over similar peatlands (Sect. 4.1)" but the reader is left to dig out bits of information through the entire discussion section. At a bare minimum, for me to trust the conclusions of this work, the authors must provide a clear and succinct comparison of their model parameters to literature values in a table/graph, including error analysis. Also, asking the reader to trust your conclusions because they match literature is fine, but there is a major issue when the story of the paper is that inter-site variation of peatland sites is high.

Specific Comments To expand on the above general comment, the eddy covariance and chamber data is not explained well enough in this paper. Let me be clear, that doesn't mean that I have issues with the data itself, just how it is presented and used here. Referee 1 brought up the issue of comparing model output to gap filled data, which is comparing a model to another model. That is absolutely an issue in this work, and I second what Referee 1 highlights as a major issue, but I'll go further. There needs to be more discussion of the data, how it was gap-filled, possible sources of
error and what that means when compared to the model results. I know this is a modeling work, but with very limited observational data to compare the model results with, simply saying in two sentences "To examine how well ecosys simulated net ecosystem CO2 exchange at the WPL, we tested hourly modelled net ecosystem CO2 fluxes against those measured by using eddy covariance (EC) micro-meteorological approach by Syed et al. (2006) and Flanagan and Syed (2011). Quality control, and gap-filling of EC measured net CO2 fluxes, and partitioning of EC-gap filled net CO2 fluxes into GPP and Re were done by Syed et al. (2006) and Flanagan and Syed (2011)" is not enough when the first line says "we used observational data" and the second says "please read those other papers for their methods". Now, when we get to the details of the chamber fluxes, there are slightly more details, but again, not nearly enough. Again, you partition net CO2 using a model, but don't explain anything beyond that. You average over 9 chambers, but don't say why or what that means? How much error is introduced here? What is the range of observed fluxes? The reader doesn't know, so again going back to my main issue, with very limited observational data, ecosys modeled results look good at the surface, but the work is limited in how much the reader can trust the results of an over-parameterized and under-tested model.

Once we move past the issue of how the observational data is described, we move through a lot of model descriptions and results that are very, very dense. I'm very happy to see what looks like the full set of equations that go into ecosys in the appendixes, but the reader is left with only very dense blocks of text to try to figure out what parts of the model are important and why. I would suggest keeping the entire set of equations in the appendix, but moving the main equations used here into the text of the manuscript. Then, the reader doesn't have to dig out the equations for context and, more importantly, it would be easier to focus the story around those few equations. As the reader is starting to get a handle of the main story presented, a major issue comes up again. There isn't enough data to support the conclusions. Even the highlighted results in the abstract are heavily focused on things like nitrogen dynamics, nutrient mineralization, GPP of plant functional types, all of which are 100 percent internal to

**BGD**
the model without any space in the manuscript devoted to why the read should trust the internal model equations.

Finally, the conclusions presented are simply changes in variables that are internal to the model, without anything to compare them with other than literature values from other studies. As mentioned in the general comment above, the literature values could be a valid check if done well, but as this manuscript is currently written, that needs to be done more formally and not in the discussion. With the suggestion of strengthening the comparison of the modeled conclusions to literature as well as the authors pinning a lot of the trust in their conclusions on said literature values, when the ending of the introduction/justification section is as follows: "Moreover, since hydrological feedbacks to key peatland C processes are highly non-linear and site-specific, testing of ecosys algorithms across contrasting peatlands would also facilitate formation of a modelling platform for scaling up simulations of those feedbacks across peatlands at larger spatial scales i.e., national, regional, continental or global as also recommended by Waddington et al. (2015)" the reader is going to be confused. On one hand, you compare your conclusions to literature and say "look, these results fit with other studies" but the entire paper was setup with the story of "there are lots of variations across peatland sites" throughout the introduction. So, I'm confused and this needs to be cleared up either by heavy editing of the story.

---

## Referee Comment (RC3) · Anonymous Referee #3 · 29 May 2017

This manuscript describes simulations of peatland biogeochemistry using the ecosys model, and compares the results to observations from a flux tower, chambers, and water table observations. The model is used to support a detailed analysis of interactions among soil microbial processes, mosses, vascular plants, and hydrology under different water table regimes. The model does a very good job of replicating measured hydrology and carbon cycling at the site, and the analysis produces some very interesting insights about the peatland response to changes in water table. The explanation of how different components of the peatland (vascular plants, non-vascular plants, aerobic decomposition, etc) interact differently under saturated, unsaturated, and deep

water table conditions was especially interesting, and I think this has great promise to be a useful framework for future analyses in this field.

Overall, I think this was a really nice paper, with novel insights, an impressive model, and clearly written (though dense) presentation. My only major suggestion in that a visualization of the interactions at different water table levels might help to summarize the results in a way that's easier for readers to grasp. Sections 2.1.2 and 2.1.3 lay out a lot of competing responses to water table depth (oxygen availability, decomposition, root growth, nitrogen availability, ...). While the description is pretty clear, I think a visualization would be really helpful. This could be as simple as a table or text box with columns for vascular plants, non-vascular plants, aerobic microbes, etc and rows for different hydrological regimes with the key processes affecting each ecosystem component.

Also, there are a lot of interesting mechanistic explanations in the Discussion, but I think there's an opportunity to tie these to the existing literature a little better. Are these new insights about process interactions that have not been discussed in the past? Or are these known interactions that have not been successfully modeled before? A little discussion of the novelty of the process interpretations and general framework of the results versus the novelty of the model itself might help place the results in a better context.

Specific comments:

Lines 17-22: This sentence is really long and hard to follow. I would suggest rewriting it.

Lines 38-40: This is an important result, and it might help to also briefly explain the process behind it (e.g. more drainage eventually leads to limitation of GPP due to water limitation).

Line 46: The units of g/yr don't seem right.

Lines 316-318: This sentence has some grammatical issues. I suggest rewriting it.

Lines 642-666: These root responses were very interesting. I don't think I've seen this process represented in ecosystem models in the past.

Lines 682-686: It's very interesting how vascular plants have an optimum at an inter-mediate water depth while non-vascular plants don't. Very interesting implications for changes in relative biomass under different conditions.

---

## Author Comment (AC1) · 21 Jul 2017

Referee's comment: This paper compares the results of the ecosystem biophysical model ecosys against field measurements of environmental variables (primarily water table depth – WTD) and carbon fluxes measured by eddy covariance (EC) at a treed peatland in western Canada over a 5 year time frame when WT was decreasing at the site. The model, ecosys, is a very sophisticated tool and has been widely applied in the past against many different ecosystem types, with success. It is fair to say that is among the top ranked platforms for simulating ecosystem functioning. With that said,

the purpose of this particular paper is a bit foggy. The EC flux measurements from this site, including the time series over which the WTD had declined, have been clearly reported in previous literature, as has been cited in this study. Therefore, is the purpose of this study 1) to simply to test if ecosys can simulate the trend in measured EC fluxes over the study period, or 2) to use ecosys to explain the behaviour of the EC-fluxes, which cannot be obtained from most common EC and environmental measurements? The paper seems to do a bit of both, but the main objective is not clear. However, given the extensive testing of ecosys at other peatlands and other ecosystems, the former seems to be quite a weak objective. The latter is more scientifically interesting, but that is not the way the paper is set out.

Authors' response: The objective of this paper was to test whether a coupling of algorithms from independent published research that describe feedbacks among peat biogeochemistry, peatland hydrology and peat forming vegetation would be able to simulate and explain WTD effects on peatland CO2 exchange in a boreal peatland. This testing of algorithms representing interactions between peatland biogeochemistry and hydrology help reconcile our current understanding based on inferences drawn numerically from relationships among EC-gap filled partitioned NEP, GPP, Re and WTD. However, given the non-linearity and peatland-specific responses of WTD-C cycle feedbacks, testing these algorithms in peatlands with contrasting peat type, vegetation, hydrology, climate and weather conditions also have important scientific and practical implications. Current predictive capacity of water table depth (WTD) effects on peat carbon (C) accumulation and degradation is limited by poor representation of peatland biogeochemistry in the peatland C models. So testing ecosys algorithms against measurements in a boreal fen, which is very different from the earlier peatlands where ecosys was tested in terms of climate, hydrology, peat forming substrates and vegetation, should complement those in earlier papers in examining the adequacy and robustness of our predictive capacity of these feedbacks.

The objective and rationale section (lines 121-134) will be edited as much as possible
to clearly lay out the objective, purpose and the implications of the study to remove any confusion.

Referee's comment: There are 4 operating hypotheses (not repeated here). These are not stated in terms of what is reflected in the EC-derived measurements, which seems to be the main thrust of the paper from the rest of the introduction, but rather in terms of biophysical processes that will take place in the model. Hence, it is a source of some of the confusion about the purpose of this paper. It would be nice to see an attempt to improve the introduction with a clearer purpose.

Authors' response: Perhaps the best way to present the model hypotheses is to state the physical and biological processes affected by WTD and how they are modelled, as we have done in the current manuscript. These processes, if accurately modelled, should manifest themselves as increased CO2 effluxes from increased Rh with increasing WTD, offset by increased CO2 influxes with increased N uptake. At some point, further increases in WTD will manifest itself as decreased CO2 effluxes and influxes with greater water stress. These manifestations should then be corroborated by observations by EC and flux chambers and by other eco-physiological measurements such as N status as has been done in the manuscript. The reflection of those hypotheses in EC-measurements has also been somewhat stated in lines 171-175. However, it will be weaved better as suggested. Overall, sections 1.1 and 1.2 in the introduction will be edited as much as possible to clearly lay out the objective, purpose and hypotheses so as to remove any confusion upfront as also mentioned above.

Referee's comment: Another concern about the present manuscript is that there is a lot of attention to how ecosys performs in simulating the WTD. It seems to me that this topic was adequately covered in the previous paper, Mezbahuddin et al. (2016), so why do we need the emphasis here.

Authors' response: The contents in lines 481-501 will be removed as much as possible to eliminate the overlap with Mezbahuddin et al. (2016).
Referee's comment: I have a small worry about the comparison of ecosys modelled fluxes against gap-filled data (especially nighttime (Re) fluxes – section 3.3). Since the gap-filling is a model itself – now we are comparing one model against another. I realize there is a discussion of how this may have affected the comparison, but that does little to convince readers that the comparison of modelled and measured data is sound. Why not just compare half hours where measured data were available to test ecosys, if that is the point of the paper (see above).

Authors' response: The real test of modelled outputs of hourly net CO2 fluxes was against EC-measured hourly net CO2 fluxes (no gap-filled fluxes) (first five rows of both Table 1 and table 2). Since the model is hourly time-step, we averaged two half-hourly measured EC net CO2 fluxes (no gap-filled fluxes) to test the modelled fluxes against with (lines 417-423). If any or both of the two half-hourly fluxes was gap-filled, the average hourly net CO2 fluxes were termed as "gap-filled".

Daily, growing season and annual aggregates of EC NEPs include number of gapfilled net CO2 fluxes. The sole reason of regressing modelled results against gap-filled CO2 fluxes was to examine how much of the deviation between modelled and EC gapfilled estimates of growing season and annual NEP, and between modelled and ECpartitioned GPP and Re were contributed by the gap-filled fluxes (Tables 1 and 2; and lines 920-936). However, since it creates confusion, we could move those regression results for gap-filled vs. modelled net CO2 fluxes to a separate table and could put the table in the appendices.

Referee's comment: Section 4.2 Divergence between modeled and EC-derived fluxes makes some interesting points, but as it stands very little of this has been tested or analyzed in any detail, so we really don't know what the source of the discrepancy is. It would be nice to see some attempt or suggestions as to how to hone in on the most likely causes of the discrepancy. You seem to suggest the EC-derived measurements are wrong, which may well be, but I am not sure that the model is not without fault.
Authors' response: In ecosys model, productivity of the plant functional types (PFT) is governed by leaf nitrogen status which is constrained by root nitrogen availability and uptake. So, the uncertainty in modelled outputs for the productivity of the PFTs depends on the accuracy of model inputs of soil organic nitrogen, wet or dry deposition, fixation and any other sources of nutrient inputs into the ecosystem. The model inputs for organic nitrogen in each peat layers were measured from the site. The background wet and dry deposition rates for NH4+ and NO3- reported for the site area were used as model inputs. However, from field observations, it was evident that there was significant nutrient inflow with the lateral water influxes into this fen peatland from the surrounding upland forests which was not quantified. To mimic this, we doubled the background wet deposition of NH4+ and NO3- reported for the area and used these as a surrogate of lateral nutrient inflow into the modelled ecosystem (lines 371-377). We also included a nitrogen fixing mechanism by lichens which was reported for the boreal forests (lines 405-408). We tested the adequacy of these nitrogen inputs into the model by comparing modelled leaf nitrogen contents against those measured in the field. The modelled leaf nitrogen contents for black spruce, tamrack and dwarf birch PFTs corroborated well against site measured leaf nitrogen contents during the summer of 2004 (lines 659-666).

To further examine the contribution of model uncertainty towards the divergence between modelled vs. EC-derived seasonal and annual GPP and Re, we performed a sensitivity test where we had a parallel run without doubling the background nitrogen wet deposition rates in the model, hence simulating no lateral nitrogen influx. This parallel run gave GPP and Re outputs for the modelled ecosystem that matched better with the EC-derived GPP and Re compared to the current run. However, the regression of the modelled net CO2 fluxes from the parallel run on the EC-measured (excluding gap-filled fluxes) fluxes gave slopes of  $\sim$ 0.8 indicating under-simulation of the EC-measured fluxes in the parallel run with no lateral nitrogen inflow despite better matching EC-derived GPP and Re aggregates.
The above discussion of model uncertainties will now be added to section 4.2.

Referee's comment: The Conclusion section is not really a conclusion. First, much of it is simply a re-statement of the main findings. Second, it suffers the same problems as the objective of the study namely, not really being clear. The final statements about the value and application of the ecosys model, while possibly true, seem a little self-serving.

Authors' response: All of the re-statements will be removed. The conclusion will be edited to link better with the revised statements of objectives. The final statements will be rephrased to remove any confusion.

Referee's comment: Finally, the manuscript needs a good editing, although the writing is such that it is understandable, there are many awkward statements/phrases, issues with tense, or grammatical errors that could be addressed to improve the manuscripts readability. I have pointed out some of these in the minor points below, but there are several others. Overall, this could be quite a useful contribution, especially if cast in the role of using ecosys to help understand the pattern and responses of EC-derived fluxes over time, something that is hard to get from just EC and environmental measurements, rather than just another test of the ecosys algorithms at another peatland site.

Authors' response: The manuscript will be thoroughly revised and edited to remove any sentence structure related or grammatical errors. The objective and rationale section (Sec. 1.1) will be revised to clearly represent the focus of the study which was to simulate and examine the underlying processes affecting WTD – peatland C cycle feedbacks. This would also serve as a reconciliation of the inferences drawn on WTD-C feedbacks in this peatland by using EC-gap filled and partitioned aggregates.

Referee's comment: Minor Issues: 1. Line 81-82 & line 87, Lafleur et al. (2005) reference is inappropriate here, they discuss Re not GPP. Also relevant on lines 908-911. 2. Lines 95-97, as above this reference does not discuss a threshold for WTD and GPP, perhaps another reference by this author?

BGD
Authors' response: The correct reference will be "Lafleur, P. M., Hember, R. A., Admiral, S. W., and Roulet, N. T.: Annual and seasonal variability in evapotranspiration and water table at a shrub-covered bog in southern Ontario, Canada, Hydrol. Process., 19, 3533-3550, 2005." It will be corrected in the revised manuscript.

Referee's comment: 3. Lines 98-107, the start of this paragraph is poorly worded. First one does not start a new paragraph with the word 'therefore". The sentence beginning "So, to adequately predict : : :" is awkward and doesn't quite read right. As with the next sentence – the phrase "do not have prognostic WTD dynamics that prevent simulation" is confusing and awkward.

Authors' response: All of the three sentences will be rephrased.

Referee's comment: 4. Lines 314-317, these two sentences that describe the simulated mosses are very difficult to understand, some revision for clarity is needed. Should be put in terms of what a real moss is and where it grows.

Authors' response: Those sentences will be edited to describe more clearly how moss is simulated in ecosys.

Referee's comment: 5. Line 447, the word diurnal here is incorrect; Table 1 compares instantaneous half hour fluxes. Diurnal suggests some course of measurements over the daytime.

Authors' response: It will be edited as suggested.

Referee's comment: 6. Line 457-58, the sentence here about 2009 is not needed here; simply add it as a foot note to the Table.

Authors' response: The line will be removed from the text and will be added as a footnote to the Table 1 as suggested.

Referee's comment: 7. Line 478-81, this sentence is somewhat heuristic; the Figure certainly does not show these components. I think it is adequate just to say the model

**BGD**
simulated measured WTD well.

Authors' response: The line will be edited as suggested.

Referee's comment: 8. Line 536, word 'also' is not needed.

Authors' response: It will be removed.

Referee's comment: 9. Line 537, what does 'It' refer to?

Authors' response: 'It' referred to 'similar day-time fluxes in 2005 and 2008 despite larger night-time fluxes in 2008 than in 2005' (will be revised).

Referee's comment: 10. Line 538-41, this long sentence is somewhat awkward and doesn't really say anything new.

Authors' response: It will be removed.

Referee's comment: 11. Lines 559-61, the sentence here seems to be missing a word or words, does not read well.

Authors' response: The sentence will be rephrased.

Referee's comment: 12. Lines 55-570, you seem to miss an opportunity here. You describe how warming does not stimulate Re when water table was high in 2005, and how it is stimulated by warming in low WT years (2006 and 2008), yet given the so-phistication in ecosys there is not real explanation of why this works the way it does, what is the biogeochemical functioning that does or does not stimulate Re under low and high WTs respectively? Further down you describe the mechanisms associated with GPP, why not the same with Re?

Authors' response: The mechanisms describing how warming affected Re in shallow vs. deeper WT conditions have already been discussed in lines 764-769.

Referee's comment: 13. Line 632-33, this was mentioned above (#6), no need to repeat it here.
Authors' response: This sentence will be deleted.

Referee's comment: 14. Lines 664-69 Section 3.5, You state this drainage experiment ": : : would 667 also serve as a climate change analog in providing us insight into how potential WTD drawdown 668 under future drier and warmer climates would affect boreal peatland GPP, Re and hence NEP." I don't see how, as the atmospheric changes of higher temperatures and perhaps higher VPD are not included. I think it is fair to say this simulation represents the effects of WT drawdown only.

Authors' response: We did not include rise in temperature and consequent VPD effects in our drainage simulation. Those sentences will be edited as suggested to remove the confusion.

Referee's comment: 15. Lines 673-74, I don't think you need this sentence, it is rather obvious that changes would occur and you have a lot of words following to describe them.

Authors' response: The sentence will be deleted.

Referee's comment: 16. Lines 683-696, is this enhanced evapotranspiration coming from the tree cover or ground vegetation or both?

Authors' response: The enhanced evapotranspiration was coming from the tree cover (will be added to the text).

Referee's comment: 17. Lines 848-51, should note here that his Dimitrov et al. 2010 study was on a temperate bog not a fen.

Authors' response: Yes that was a bog. The sentence will be omitted.

BGD

---

## Author Comment (AC2) · 21 Jul 2017

Referee's comment: General Comments The manuscript addresses impacts on CO2 fluxes from changes in water table depth by using the ecosys model. The model is tested with eddy covariance and chamber CO2 fluxes from a boreal peatland field site. The manuscript is dense throughout and requires very careful attention on the part of the reader to follow along. While the ecosys model is complex, how this paper is written exacerbates the complexity of the model. Right now, this paper would be an incredibly useful guide to someone wanting to run the ecosys model themselves, but

lacks a clear and story that is supported by the results of this work.

Authors' response: Significant effort will be made to edit the story of the manuscript to simplify it as much as possible. The linkage between the findings and the data will be made clearer following the suggestions of the reviewer wherever applicable.

Referee's comment: The major issue I have with this work is the relative complexity of the ecosys model next to the small amount of observed data that the model is compared with. Since ecosys has so many moving parts "under the hood", I can't say that I'm surprised at all to see it match data as well as it does. A good fit to observed data is not a new finding itself, and in a broader sense, the research questions aren't new. In fact, there is a good amount of overlap with Mezbahuddin et al. (2016), as brought up by Referee 1. So, I'm stuck reading through a dense description of a complex model, and at the end, it's compared with limited amounts of data that itself is modeled. The main conclusions seem to be focused on internal modeled variables within the ecosys model that have zero comparison to data. The major conclusions are changes in modeled $O_2$ diffusion, N mineralization rates, nutrient availability, microbial concentrations, plant functional type GPP. These results, as currently presented, are simply not supported by comparing to net $CO_2$ fluxes. The authors state in the conclusions that "These modelling hypotheses were also corroborated by various field, laboratory and modelling studies over similar peatlands (Sect. 4.1)" but the reader is left to dig out bits of information through the entire discussion section. At a bare minimum, for me to trust the conclusions of this work, the authors must provide a clear and succinct comparison of their model parameters to literature values in a table/graph, including error analysis. Also, asking the reader to trust your conclusions because they match literature is fine, but there is a major issue when the story of the paper is that inter-site variation of peatland sites is high.

Authors' response: Northern peatlands are likely to be important in future carbon cycle-climate feedbacks due to their large carbon pools and vulnerability to hydrological change. Current predictive capacity of water table depth (WTD) effects on peat

carbon (C) accumulation and degradation is limited by poor representation of peatland biogeochemistry in the peatland C models. So, the novelty of this research lies upon its effort to test whether a coupling of algorithms from independent published research that describe feedbacks among peat biogeochemistry, peatland hydrology and peat forming vegetation would be able to simulate and explain WTD effects on peatland $CO_2$ exchange in a boreal peatland. This testing of algorithms representing interactions between peatland biogeochemistry and hydrology not only improves our predictive capacity of WTD effects on peatland $CO_2$ exchange, but also help reconcile our current understanding based on inferences drawn numerically from relationships among EC-gap filled partitioned NEP, GPP, Re and WTD.

Section 1 describes how peatland WTD – C cycle feedbacks vary across peatlands depending upon climate, hydrology, peat substrate type, and peatland vegetation (lines 58-119). It does not stop at only portraying the inter-site variations of these feedbacks. It goes further to explore, by using existing literature, how these inter-site variations are mediated by the interactions among peat forming climate and vegetation, peatland hydrology, and peat type (lines 58-119). Following recommendations from previous model inter-comparison studies, it describes how representing interactions among peat biogeochemistry, hydrology and peat vegetation physiology could lead to improved predictive capacity of the effects of these inter-peatland variations of WTD–C process feedbacks without peatland-specific parameterization of model algorithms (lines 101-119). So, the objective of this study was to examine whether a coupling of site-independent model algorithms describing peatland carbon, nitrogen, oxygen, and water cycling which was fed by site-specific measurable inputs with physical meanings would simulate and explain WTD effects on peatland $CO_2$ exchange for the boreal peatland. Physical and biological processes that mediate WTD effects on peatland $CO_2$ exchange, if accurately modelled, should manifest themselves as increased $CO_2$ effluxes from increased Rh with increasing WTD, offset by increased $CO_2$ influxes with increased N uptake. At some point, further increases in WTD will manifest itself as decreased $CO_2$ effluxes and influxes with greater water stress. These manifestations

should then be corroborated by observations by EC and flux chambers and by other eco-physiological measurements such as N status as has been done in our manuscript.

In the current manuscript, the hourly modelled outputs of net ecosystem $CO_2$ fluxes were first tested against hourly net $CO_2$ fluxes measured (excluding the gap-filled values) by eddy covariance (EC) approach over a gradually drying weather period from 2004 to 2009 (first five rows of tables 1 and 2). Then the modelled trend of WTD drawdown effect on net $CO_2$ exchange were examined closely for shorter periods (e.g. 10-day) of gradual WTD drawdown along with EC-measured hourly net ecosystem $CO_2$ fluxes, and chamber measured hourly net understory vegetation and soil $CO_2$ fluxes (Figs. 3-5). The examination of WTD effects on $CO_2$ exchange was then extended to daily, growing season, and annual time-scales along with EC-gap filled NEP and partitioned GPP and Re aggregates (Figs. 2, 6-7). The internal peat biogeochemistry and peatland nutrient cycling modelled in "ecosys" were tested against leaf nitrogen concentrations, N mineralization, rooting depth, GPP, and Re measured at either our site or at sites that had similar peat substrates, hydrology and/or plant functional types (Secs. 3.4., 4.1). The above mentioned model validation ensured that the modelled outputs were tested against 6 years of hourly EC measurements under contrasting conditions e.g. wet vs. dry, cool vs. warm. These tests were further corroborated by observations by flux chambers and by other eco-physiological measurements such as N status. So, the testing of the modelled results were as robust as it could be within the best availability of measurements. However, there is always room for improvement and we will be making the following edits as suggested by the referee:

1) The "Objective and rationale" section (Sec. 1.1) will be edited to make the research question more vivid and to clearly depict the novelty of the research question. The potential of modelling peat biogeochemistry to explain and simulate the effects of site-variability on peatland WTD-C process feedbacks will be described more clearly to avoid any confusion.

2) The comparison among modelled parameters, site measurements, and other literature values of peat biogeochemistry and nutrient cycling will be presented in a separate table instead of the way they are now presented in the texts in Secs. 3.4. and 4.1.

3) The overlaps with Mezbahuddin et al. (2016) has been identified in lines 481-501. Mezbahuddin et al. (2016) talked about hydrological modelling of the same site some of which were relevant to the current manuscript. However, these overlaps will be removed as much as possible.

Referee's comment: Specific Comments to expand on the above general comment, the eddy covariance and chamber data is not explained well enough in this paper. Let me be clear, that doesn't mean that I have issues with the data itself, just how it is presented and used here. Referee 1 brought up the issue of comparing model output to gap filled data, which is comparing a model to another model. That is absolutely an issue in this work, and I second what Referee 1 highlights as a major issue, but I'll go further. There needs to be more discussion of the data, how it was gap-filled, possible sources of error and what that means when compared to the model results. I know this is a modeling work, but with very limited observational data to compare the model results with, simply saying in two sentences "To examine how well ecosys simulated net ecosystem $CO_2$ exchange at the WPL, we tested hourly modelled net ecosystem $CO_2$ fluxes against those measured by using eddy covariance (EC) micro-meteorological approach by Syed et al. (2006) and Flanagan and Syed (2011). Quality control, and gap-filling of EC measured net $CO_2$ fluxes, and partitioning of EC-gap filled net $CO_2$ fluxes into GPP and Re were done by Syed et al. (2006) and Flanagan and Syed (2011)" is not enough when the first line says "we used observational data" and the second says "please read those other papers for their methods". Now, when we get to the details of the chamber fluxes, there are slightly more details, but again, not nearly enough. Again, you partition net $CO_2$ using a model, but don't explain anything beyond that.

Authors' response: Daily, growing season and annual aggregates of EC NEP includes number of gap-filled net $CO_2$ fluxes. The sole reason of regressing modelled results

against gap-filled CO2 fluxes was to examine how much of the deviation between modelled and EC gap-filled estimates of growing season and annual NEP, and between modelled and EC-partitioned GPP and Re were contributed by the gap-filled fluxes (Tables 1 and 2; and lines 920-936). However, since it creates confusion, we could move those regression results for gap-filled vs. modelled net CO2 fluxes to a separate table and could put the table in the appendices.

Methods for screening, gap-filling, and partitioning of EC datasets will be described in sufficient details in sec. 2.2.2.

Referee's comment: You average over 9 chambers, but don't say why or what that means? How much error is introduced here? What is the range of observed fluxes? The reader doesn't know, so again going back to my main issue, with very limited observational data, ecosys modeled results look good at the surface, but the work is limited in how much the reader can trust the results of an over-parameterized and under-tested model.

Authors' response: Those 9 chambers were in place to cover spatial variation due to peatland micro-topography while measuring the net CO2 fluxes from understory vegetation and soil. We averaged those chambers to include overall hummock-hollow variations of those fluxes. The reasoning will be described in sufficient details in sec. 2.2.2. Also, figs. 4 and 5 will be redone to include standard error of means of spatially averaged chamber fluxes so as to represent flux variations among the chambers. Moreover, there will be a separate regression test between modelled understory and soil CO2 fluxes, and the chamber CO2 fluxes for 2005 and 2006 for which we had measurements available. This will further strengthen the robustness of the test of modelled outputs.

Referee's comment: Once we move past the issue of how the observational data is described, we move through a lot of model descriptions and results that are very, very dense. I'm very happy to see what looks like the full set of equations that go into

ecosys in the appendixes, but the reader is left with only very dense blocks of text to try to figure out what parts of the model are important and why. I would suggest keeping the entire set of equations in the appendix, but moving the main equations used here into the text of the manuscript. Then, the reader doesn't have to dig out the equations for context and, more importantly, it would be easier to focus the story around those few equations.

Authors' response: The model development in sec. 2.1. in current manuscript only describes the key equations that are related to the hypotheses. The respective equations that are listed in the appendices are also cited within the text. The current model description is self-explanatory and a reader does not have to always go back to the appendices to understand the processes. However, the citation of the equation within the text makes sure that a reader can go back to the appendices at any time to see details of a particular equation. Few key equations from the appendices could be pasted into and described in sec. 2.1 as suggested. But we feel that it will either make the section even denser or the story could be incomplete if roamed around too few equations. Instead, our preference is to include a flow chart summarizing those key processes and linking the flow chart with the description as suggested by reviewer 3 (please see the figure attached with the authors' response to the comments of reviewer # 3 as example).

Referee's comment: As the reader is starting to get a handle of the main story presented, a major issue comes up again. There isn't enough data to support the conclusions. Even the highlighted results in the abstract are heavily focused on things like nitrogen dynamics, nutrient mineralization, GPP of plant functional types, all of which are 100 percent internal to the model without any space in the manuscript devoted to why the read should trust the internal model equations.

Authors' response: How the modelled outputs were tested against measurements and how the testing would be improved have been discussed in reply to the general comments above. The model equations were derived from independent research which

were rigorously tested in other published studies. The sources of those equations have been listed in the supplementary material. However, a separate brief discussion about the sources and significance of the model equations will be added to section 2.1.

Referee's comment: Finally, the conclusions presented are simply changes in variables that are internal to the model, without anything to compare them with other than literature values from other studies. As mentioned in the general comment above, the literature values could be a valid check if done well, but as this manuscript is currently written, that needs to be done more formally and not in the discussion. With the suggestion of strengthening the comparison of the modeled conclusions to literature as well as the authors pinning a lot of the trust in their conclusions on said literature values, when the ending of the introduction/justification section is as follows: "Moreover, since hydrological feedbacks to key peatland C processes are highly non-linear and site-specific, testing of ecosys algorithms across contrasting peatlands would also facilitate formation of a modelling platform for scaling up simulations of those feedbacks across peatlands at larger spatial scales i.e., national, regional, continental or global as also recommended by Waddington et al. (2015)" the reader is going to be confused. On one hand, you compare your conclusions to literature and say "look, these results fit with other studies" but the entire paper was setup with the story of "there are lots of variations across peatland sites" throughout the introduction. So, I'm confused and this needs to be cleared up either by heavy editing of the story.

Authors' response: The conclusion section will be sufficiently edited to include general conclusions, and the implications of those conclusions. A formal comparison between the modelled outputs and measurements from the same site, and/or similar sites from earlier studies will be done in a separate table as mentioned earlier. The mentioned sentence within the quote will be rephrased to remove any confusion and contradiction with earlier description.

---

## Author Comment (AC3) · 21 Jul 2017

Referee's comment: This manuscript describes simulations of peatland biogeochemistry using the ecosys model, and compares the results to observations from a flux tower, chambers, and water table observations. The model is used to support a detailed analysis of interactions among soil microbial processes, mosses, vascular plants, and hydrology under different water table regimes. The model does a very good job of replicating measured hydrology and carbon cycling at the site, and the analysis produces some very interesting insights about the peatland response to changes in water

table. The explanation of how different components of the peatland (vascular plants, non-vascular plants, aerobic decomposition, etc) interact differently under saturated, unsaturated, and deep water table conditions was especially interesting, and I think this has great promise to be a useful framework for future analyses in this field. Overall, I think this was a really nice paper, with novel insights, an impressive model, and clearly written (though dense) presentation.

Referee's comment: My only major suggestion in that a visualization of the interactions at different water table levels might help to summarize the results in a way that's easier for readers to grasp. Sections 2.1.2 and 2.1.3 lay out a lot of competing responses to water table depth (oxygen availability, decomposition, root growth, nitrogen availability, : : :). While the description is pretty clear, I think a visualization would be really helpful. This could be as simple as a table or text box with columns for vascular plants, non-vascular plants, aerobic microbes, etc and rows for different hydrological regimes with the key processes affecting each ecosystem component.

Authors' response: A flow chart will be added to depict the key biogeochemical and physiological responses to different water regimes (e.g. aerobic vs. anaerobic) that were modelled in ecosys and described in section 2.1. An example of the flow chart is attached herewith as figure 1.

Referee's comment: Also, there are a lot of interesting mechanistic explanations in the Discussion, but I think there's an opportunity to tie these to the existing literature a little better. Are these new insights about process interactions that have not been discussed in the past? Or are these known interactions that have not been successfully modeled before? A little discussion of the novelty of the process interpretations and general framework of the results versus the novelty of the model itself might help place the results in a better context.

Authors' response: The mechanistic explanations in the discussion will be summarized in a table to facilitate the comparison between modelled processes and existing literature. This will be done by comparing modelled outputs of different components of carbon and nutrient cycling against measurements available for the site or for similar sites in the literatures with references. The intention of the mechanistic discussion was to test the adequacy of existing knowledge of our process interpretation to reproduce WTD-C feedbacks in a northern boreal fen which has not been done before. The novelty of the process interpretations vs. modelled processes will be briefly discussed in the discussion section wherever applicable.

Referee's comment: Specific comments: Lines 17-22: This sentence is really long and hard to follow. I would suggest rewriting it.

Authors' response: Will be done.

Referee's comment: Lines 38-40: This is an important result, and it might help to also briefly explain the process behind it (e.g. more drainage eventually leads to limitation of GPP due to water limitation).

Authors' response: Will be done.

Referee's comment: Line 46: The units of g/yr don't seem right.

Authors' response: It should be g m-2 yr-1, will be corrected.

Referee's comment: Lines 316-318: This sentence has some grammatical issues. I suggest rewriting it.

Authors' response: Will be done.

Referee's comment: Lines 642-666: These root responses were very interesting. I don't think I've seen this process represented in ecosystem models in the past. Lines 682-686: It's very interesting how vascular plants have an optimum at an intermediate water depth while non-vascular plants don't. Very interesting implications for changes in relative biomass under different conditions.

[Figure]

**Fig. 1.**

---

## Author Response (AR1)

**Authors' responses to the comments of anonymous Referee #1**

*[All the references to the line numbers, figures and tables in authors' responses are of revised version of the manuscript unless otherwise stated]*

Referee's comment: This paper compares the results of the ecosystem biophysical model ecosys against field measurements of environmental variables (primarily water table depth – WTD) and carbon fluxes measured by eddy covariance (EC) at a treed peatland in western Canada over a 5 year time frame when WT was decreasing at the site. The model, ecosys, is a very sophisticated tool and has been widely applied in the past against many different ecosystem types, with success. It is fair to say that is among the top ranked platforms for simulating ecosystem functioning. With that said, the purpose of this particular paper is a bit foggy. The EC flux measurements from this site, including the time series over which the WTD had declined, have been clearly reported in previous literature, as has been cited in this study. Therefore, is the purpose of this study 1) to simply to test if ecosys can simulate the trend in measured EC fluxes over the study period, or 2) to use ecosys to explain the behaviour of the EC-fluxes, which cannot be obtained from most common EC and environmental measurements? The paper seems to do a bit of both, but the main objective is not clear. However, given the extensive testing of ecosys at other peatlands and other ecosystems, the former seems to be quite a weak objective. The latter is more scientifically interesting, but that is not the way the paper is set out.

*Authors' response:*

*The objective of this paper was to test whether a coupling of algorithms from independent published research that describe feedbacks among peat biogeochemistry, peatland hydrology and peat forming vegetation would be able to simulate and explain WTD effects on peatland $CO_2$ exchange in a boreal peatland. This testing of algorithms representing interactions between peatland biogeochemistry and hydrology help reconcile our current understanding based on inferences drawn numerically from relationships among EC-gap filled partitioned NEP, GPP, $R_e$ and WTD. However, given the non-linearity and peatland-specific responses of WTD-C cycle feedbacks, testing these algorithms in peatlands with contrasting peat type, vegetation, hydrology, climate and weather conditions also have important scientific and practical implications. Current predictive capacity of water table depth (WTD) effects on peat carbon (C) accumulation and degradation is limited by poor representation of peatland biogeochemistry in the peatland C models. So testing ecosys algorithms against measurements in a boreal fen, which is very different from the earlier peatlands where ecosys was tested in terms of climate, hydrology, peat forming substrates and vegetation, should complement those in earlier papers in examining the adequacy and robustness of our predictive capacity of these feedbacks.*

*The objective and rationale section (lines 122-139) are now heavily edited to clearly lay out the objective, purpose and the implications of the study to remove any confusion.*

Referee's comment: There are 4 operating hypotheses (not repeated here). These are not stated in
terms of what is reflected in the EC-derived measurements, which seems to be the main thrust of
the paper from the rest of the introduction, but rather in terms of biophysical processes that will
take place in the model. Hence, it is a source of some of the confusion about the purpose of this
paper. It would be nice to see an attempt to improve the introduction with a clearer purpose.

*Authors' response:*

*Perhaps the best way to present the model hypotheses is to state the physical and*
*biological processes affected by WTD and how they are modelled, as we have done in the*
*current manuscript. These processes, if accurately modelled, should manifest themselves as*
*increased $CO_2$ effluxes from increased $R_h$ with increasing WTD, offset by increased $CO_2$ influxes*
*with increased N uptake. At some point, further increases in WTD will manifest itself as*
*decreased $CO_2$ effluxes and influxes with greater water stress. These manifestations should then*
*be corroborated by observations by EC and flux chambers and by other eco-physiological*
*measurements such as N status as has been done in the manuscript. The reflection of those*
*hypotheses in EC-measurements was also somewhat stated in lines 171-175 of the previous*
*version of the manuscript. However, the objective and rationale (lines 122-139) and hypotheses*
*(lines 141-172) sub-sections within the introduction section are now heavily edited to clearly lay*
*out the objective, purpose and hypotheses so as to remove any confusion upfront as suggested.*

Referee's comment: Another concern about the present manuscript is that there is a lot of
attention to how ecosys performs in simulating the WTD. It seems to me that this topic was
adequately covered in the previous paper, Mezbahuddin et al. (2016), so why do we need the
emphasis here.

*Authors' response:*

*The contents in lines 481-501 of the previous version of the manuscript are now removed*
*to eliminate the overlap with Mezbahuddin et al. (2016) (lines 482-483).*

Referee's comment: I have a small worry about the comparison of ecosys modelled fluxes
against gap-filled data (especially nighttime (Re) fluxes – section 3.3). Since the gap-filling is a
model itself – now we are comparing one model against another. I realize there is a discussion of
how this may have affected the comparison, but that does little to convince readers that the
comparison of modelled and measured data is sound. Why not just compare half hours where
measured data were available to test ecosys, if that is the point of the paper (see above).

*Authors' response:*

*The real test of modelled outputs of hourly net $CO_2$ fluxes was against EC-measured*
*hourly net $CO_2$ fluxes excluding any gap-filled flux. Since the model is hourly time-step, we*
*averaged two half-hourly measured EC net $CO_2$ fluxes (no gap-filled fluxes) to test the modelled*

*fluxes against. If any or both of the two half-hourly fluxes was gap-filled, the average hourly net*
*$CO_2$ fluxes were termed as "gap-filled".*

*Daily, growing season and annual aggregates of EC NEPs include number of gap-filled*
*net $CO_2$ fluxes. The sole reason of regressing modelled results against gap-filled $CO_2$ fluxes was*
*to examine how much of the deviation between modelled and EC gap-filled estimates of growing*
*season and annual NEP, and between modelled and EC-partitioned GPP and $R_e$ were*
*contributed by the gap-filled fluxes. However, since it created confusion, we now moved those*
*regression results for gap-filled vs. modelled net $CO_2$ fluxes to a separate table in the*
*supplementary material and edited the texts accordingly (lines 404-414, 447-467, 840-858)*
*(Table S1 in page 43 of supplementary material).*

Referee's comment: Section 4.2 Divergence between modeled and EC-derived fluxes makes
some interesting points, but as it stands very little of this has been tested or analyzed in any
detail, so we really don't know what the source of the discrepancy is. It would be nice to see
some attempt or suggestions as to how to hone in on the most likely causes of the discrepancy.
You seem to suggest the EC-derived measurements are wrong, which may well be, but I am not
sure that the model is not without fault.

*Authors' response:*

*The likely contribution of uncertainties due to modelling into the divergence between*
*modelled and EC-derived seasonal and annual GPP and $R_e$ estimates are now included in lines*
*873-897 (the texts in those lines are not repeated here).*

Referee's comment: The Conclusion section is not really a conclusion. First, much of it is simply
a re-statement of the main findings. Second, it suffers the same problems as the objective of the
study namely, not really being clear. The final statements about the value and application of the
ecosys model, while possibly true, seem a little self-serving.

*Authors' response:*

*The conclusions section is now heavily edited to remove most of the re-statements, and to*
*link it better with the revised statements of objectives. The final statements are now rephrased to*
*remove any confusion (lines 899-941).*

Referee's comment: Finally, the manuscript needs a good editing, although the writing is such
that it is understandable, there are many awkward statements/phrases, issues with tense, or
grammatical errors that could be addressed to improve the manuscripts readability. I have
pointed out some of these in the minor points below, but there are several others. Overall, this
could be quite a useful contribution, especially if cast in the role of using ecosys to help
understand the pattern and responses of EC-derived fluxes over time, something that is hard to get from just EC and environmental measurements, rather than just another test of the ecosys
algorithms at another peatland site.

*Authors' response:*

*The manuscript is now thoroughly revised and edited to remove any sentence or*
*grammatical errors (e.g., lines 118-120, 498-539, 543-568, 571-580, 583-591, 593-594, 614-*
*617, 627-629, 631-640, 643-649, 655-662, 840-858 etc.). The objective and rationale section*
*(lines 122-139) is now revised to clearly represent the focus of the study which was to examine*
*the underlying processes affecting WTD – peatland C cycle feedbacks. This will also serve as a*
*reconciliation of the inferences drawn on WTD-C feedbacks in this peatland by using EC-gap*
*filled and partitioned aggregates.*

Referee's comment: Minor Issues: 1. Line 81-82 & line 87, Lafleur et al. (2005) reference is
inappropriate here, they discuss Re not GPP. Also relevant on lines 908-911. 2. Lines 95-97, as
above this reference does not discuss a threshold for WTD and GPP, perhaps another reference
by this author?

*Authors' response:*

*The correct reference is now inserted (lines 1027-1029).*

Referee's comment: 3. Lines 98-107, the start of this paragraph is poorly worded. First one does
not start a new paragraph with the word 'therefore". The sentence beginning "So, to adequately
predict : : :" is awkward and doesn't quite read right. As with the next sentence – the phrase "do
not have prognostic WTD dynamics that prevent simulation" is confusing and awkward.

*Authors' response:*

*All of the three sentences are now rephrased (lines 103-109).*

Referee's comment: 4. Lines 314-317, these two sentences that describe the simulated mosses
are very difficult to understand, some revision for clarity is needed. Should be put in terms of
what a real moss is and where it grows.

*Authors' response:*

*Those sentences are now edited to describe more specifically how moss is simulated in*
*ecosys (lines 299-305).*

Referee's comment: 5. Line 447, the word diurnal here is incorrect; Table 1 compares
instantaneous half hour fluxes. Diurnal suggests some course of measurements over the daytime.

*Authors' response:*

*Edited (line 448).*

Referee's comment: 6. Line 457-58, the sentence here about 2009 is not needed here; simply add
it as a foot note to the Table.

*Authors' response:*

*The line is now removed from the text and is added as a footnote to the Table 1.*

Referee's comment: 7. Line 478-81, this sentence is somewhat heuristic; the Figure certainly
does not show these components. I think it is adequate just to say the model simulated measured
WTD well.

*Authors' response:*

*Edited (lines 482-483).*

Referee's comment: 8. Line 536, word 'also' is not needed.

*Authors' response:*

*Removed (line 512).*

Referee's comment: 9. Line 537, what does 'It' refer to?

*Authors' response:*

*'It' referred to 'similar day-time fluxes in 2005 and 2008 despite larger night-time fluxes*
*in 2008 than in 2005', now revised (lines 516-517).*

Referee's comment: 10. Line 538-41, this long sentence is somewhat awkward and doesn't really
say anything new.

*Authors' response:*

*The sentence is now removed.*

Referee's comment: 11. Lines 559-61, the sentence here seems to be missing a word or words,
does not read well.

*Authors' response:*

*The sentence is now rephrased (lines 529-532).*

Referee's comment: 12. Lines 55-570, you seem to miss an opportunity here. You describe how
warming does not stimulate Re when water table was high in 2005, and how it is stimulated by
warming in low WT years (2006 and 2008), yet given the sophistication in ecosys there is not
real explanation of why this works the way it does, what is the biogeochemical functioning that does or does not stimulate Re under low and high WTs respectively? Further down you describe
the mechanisms associated with GPP, why not the same with Re?

*Authors' response:*

*The mechanisms describing how warming affected $R_e$ in shallow vs. deeper WT*
*conditions is discussed in lines 697-702.*

Referee's comment: 13. Line 632-33, this was mentioned above (#6), no need to repeat it here.

*Authors' response:*

*Deleted.*

Referee's comment: 14. Lines 664-69 Section 3.5, You state this drainage experiment ": : :
would 667 also serve as a climate change analog in providing us insight into how potential WTD
drawdown 668 under future drier and warmer climates would affect boreal peatland GPP, Re and
hence NEP." I don't see how, as the atmospheric changes of higher temperatures and perhaps
higher VPD are not included. I think it is fair to say this simulation represents the effects of WT
drawdown only.

*Authors' response:*

*We did not include rise in temperature and consequent VPD effects in our drainage*
*simulation. Those sentences are now edited as suggested to remove the confusion (lines 38-39,*
*911-912).*

Referee's comment: 15. Lines 673-74, I don't think you need this sentence, it is rather obvious
that changes would occur and you have a lot of words following to describe them.

*Authors' response:*

*Deleted.*

Referee's comment: 16. Lines 683-696, is this enhanced evapotranspiration coming from the tree
cover or ground vegetation or both?

*Authors' response:*

*The enhanced evapotranspiration was coming from the tree cover. Now explicitly*
*mentioned in the text (lines 628, 632).*

Referee's comment: 17. Lines 848-51, should note here that his Dimitrov et al. 2010 study was
on a temperate bog not a fen.

*Authors' response:*

*It is now mentioned within the text (line 774).*

**Authors' responses to the comments of anonymous Referee #2**

General Comments

Referee's comment: The manuscript addresses impacts on $CO_2$ fluxes from changes in water table depth by using the ecosys model. The model is tested with eddy covariance and chamber $CO_2$ fluxes from a boreal peatland field site. The manuscript is dense throughout and requires very careful attention on the part of the reader to follow along. While the ecosys model is complex, how this paper is written exacerbates the complexity of the model. Right now, this paper would be an incredibly useful guide to someone wanting to run the ecosys model themselves, but lacks a clear and story that is supported by the results of this work. The major issue I have with this work is the relative complexity of the ecosys model next to the small amount of observed data that the model is compared with. Since ecosys has so many moving parts "under the hood", I can't say that I'm surprised at all to see it match data as well as it does. A good fit to observed data is not a new finding itself, and in a broader sense, the research questions aren't new. In fact, there is a good amount of overlap with Mezbahuddin et al. (2016), as brought up by Referee 1. So, I'm stuck reading through a dense description of a complex model, and at the end, it's compared with limited amounts of data that itself is modeled. The main conclusions seem to be focused on internal modeled variables within the ecosys model that have zero comparison to data. The major conclusions are changes in modeled $O_2$ diffusion, N mineralization rates, nutrient availability, microbial concentrations, plant functional type GPP. These results, as currently presented, are simply not supported by comparing to net $CO_2$ fluxes. The authors state in the conclusions that "These modelling hypotheses were also corroborated by various field, laboratory and modelling studies over similar peatlands (Sect. 4.1)" but the reader is left to dig out bits of information through the entire discussion section. At a bare minimum, for me to trust the conclusions of this work, the authors must provide a clear and succinct comparison of their model parameters to literature values in a table/graph, including error analysis. Also, asking the reader to trust your conclusions because they match literature is fine, but there is a major issue when the story of the paper is that inter-site variation of peatland sites is high.

*Authors' response: To facilitate point-by-point responses, we addressed the general comments by separating and re-organizing those into smaller segments under the following points:*

(1) The manuscript is dense throughout and requires very careful attention on the part of the reader to follow along.

*Authors' response:*

*Significant editing of the manuscript is done to simplify the story as much as possible (e.g., lines 118-120, 122-172, 498-539, 543-568, 571-580, 583-591, 593-594, 614-617, 627-629, 631-640, 643-649, 655-662, 840-858, 899-941 etc.). The main body of the revised version of the manuscript is about 700 words smaller than the earlier version of the manuscript.*

(2) Right now, this paper would be an incredibly useful guide to someone wanting to run the
ecosys model themselves, but lacks a clear and story that is supported by the results of this
work….The major issue I have with this work is the relative complexity of the ecosys model
next to the small amount of observed data that the model is compared with. Since ecosys has
so many moving parts "under the hood", I can't say that I'm surprised at all to see it match
data as well as it does…. So, I'm stuck reading through a dense description of a complex
model, and at the end, it's compared with limited amounts of data that itself is modeled….
These results, as currently presented, are simply not supported by comparing to net $CO_2$
fluxes.

*Authors' response:*

*Hourly modelled outputs of net ecosystem $CO_2$ fluxes have already been tested against hourly*
*net $CO_2$ fluxes measured (excluding the gap-filled values) by eddy covariance (EC) approach over*
*a gradually drying weather period from 2004 to 2009 (Tables 1a, b). Modelled hourly understorey*
*vegetation and soil $CO_2$ fluxes are also tested against hourly automated chamber measured $CO_2$*
*fluxes over two years with contrasting WTD conditions i.e., 2005 with shallower WTD vs. 2006*
*with deeper WTD (Table 1c). To further constrain and explain modelled WTD effects on GPP and*
*$R_e$, modelled daytime and nighttime net $CO_2$ exchange have already been examined closely for*
*shorter periods (e.g. 10-day) with contrasting WTD along with EC-measured net ecosystem $CO_2$*
*fluxes, and chamber measured net understory vegetation and soil $CO_2$ fluxes (Figs. 4-6). The*
*examination of modelled WTD effects on $CO_2$ exchange have already been extended to daily,*
*growing season, and annual time-scales along with EC-gap filled NEP and partitioned GPP and*
*$R_e$ (Figs. 3, 7,8).*
*The internal peat biogeochemistry and peatland nutrient cycling modelled in ecosys are now*
*tested against leaf nitrogen concentrations, N mineralization, rooting depth, GPP, and $R_e$*
*measured/estimated at either our site or at sites that had similar peat substrates, hydrology and/or*
*plant functional types (Table 2) (lines 744-749). So, the testing of the modelled results should be*
*as robust as it could be within the best availability of measurements.*

(3) A good fit to observed data is not a new finding itself, and in a broader sense, the research
questions aren't new.

*Authors' response:*

*Northern peatlands are likely to be important in future carbon cycle-climate feedbacks due*
*to their large carbon pools and vulnerability to hydrological change. Current predictive capacity*
*on water table depth (WTD) effects on peat carbon (C) accumulation and degradation is limited*
*by poor representation of peatland biogeochemistry in the peatland C models. So, the novelty of*
*this research lies upon its effort to test whether a coupling of algorithms from independent*
*published research that describe feedbacks among peat biogeochemistry, peatland hydrology*
*and peat forming vegetation would be able to simulate and explain WTD effects on peatland $CO_2$*
*exchange in a boreal peatland. This testing of algorithms representing interactions between*

*peatland biogeochemistry and hydrology not only improves our predictive capacity of WTD*
*effects on peatland $CO_2$ exchange, but also help reconcile our current understanding based on*
*inferences drawn numerically from relationships among EC-gap filled partitioned NEP, GPP, $R_e$*
*and WTD.*

*Previous model inter-comparison studies showed the need for representing interactions*
*among peat biogeochemistry, hydrology and peat vegetation physiology to improve predictive*
*capacity of WTD –C process feedbacks without peatland-specific parameterization of model*
*algorithms (lines 103-120). So, the objective of this study was to examine whether a coupling of*
*site-independent model algorithms describing peatland carbon, nitrogen, oxygen, and water*
*cycling which was fed by site-specific measurable inputs with physical meanings would simulate*
*and explain WTD effects on peatland $CO_2$ exchange for the boreal peatland. The objective and*
*rationale section is now heavily edited to more clearly state the research objective (lines 122-*
*139).*

(4) In fact, there is a good amount of overlap with Mezbahuddin et al. (2016), as brought up by
Referee 1.

*Authors' response:*

*The contents in lines 481-501 of the previous version of the manuscript are now removed to*
*eliminate the overlap with Mezbahuddin et al. (2016) (lines 482-483).*

(5) The main conclusions seem to be focused on internal modeled variables within the ecosys
model that have zero comparison to data. The major conclusions are changes in modeled $O_2$
diffusion, N mineralization rates, nutrient availability, microbial concentrations, plant
functional type GPP……. The authors state in the conclusions that "These modelling
hypotheses were also corroborated by various field, laboratory and modelling studies over
similar peatlands (Sect. 4.1)" but the reader is left to dig out bits of information through the
entire discussion section. At a bare minimum, for me to trust the conclusions of this work, the
authors must provide a clear and succinct comparison of their model parameters to literature
values in a table/graph, including error analysis.

*Authors' response:*

*The comparison of above mentioned modelled outputs are now formally and succinctly done*
*in table 2 against corresponding measurements at the site or at peatlands that had similar peat*
*substrates, hydrology and/or plant functional types.*

(6) Also, asking the reader to trust your conclusions because they match literature is fine, but
there is a major issue when the story of the paper is that inter-site variation of peatland sites
is high.

*Authors' response:*

*Our main tests of the modelled outputs have been against EC, chamber and biometric*
*measurements at the site (Figs. 3-7) (Tables 1, 2). Since the inter-site variations of feedbacks*
*between WTD and peatland C cycle are mediated by the interactions among peat forming*
*climate and vegetation, peatland hydrology, and peat type (lines 60-104), we have always made*
*sure that the sites we are comparing our modelled processes against are similar to our sites in*
*any combination of those characteristics (e.g., footnotes in table 2). However, we have also*
*discussed our results against some contrasting peatlands so as to highlight the underlying*
*mechanisms that lead to inter-site variations in these feedbacks (e.g., lines 832-837). The*
*conclusion section is heavily edited to remove any confusion related to inter-site variation of*
*these feedbacks (lines 921-940).*

Specific Comments:

Referee's comment: Specific Comments to expand on the above general comment, the eddy
covariance and chamber data is not explained well enough in this paper. Let me be clear, that
doesn't mean that I have issues with the data itself, just how it is presented and used here.
Referee 1 brought up the issue of comparing model output to gap filled data, which is comparing
a model to another model. That is absolutely an issue in this work, and I second what Referee 1
highlights as a major issue, but I'll go further. There needs to be more discussion of the data,
how it was gap-filled, possible sources of error and what that means when compared to the
model results. I know this is a modeling work, but with very limited observational data to
compare the model results with, simply saying in two sentences "To examine how well ecosys
simulated net ecosystem $CO_2$ exchange at the WPL, we tested hourly modelled net ecosystem
$CO_2$ fluxes against those measured by using eddy covariance (EC) micro-meteorological
approach by Syed et al. (2006) and Flanagan and Syed (2011). Quality control, and gap-filling of
EC measured net $CO_2$ fluxes, and partitioning of EC-gap filled net $CO_2$ fluxes into GPP and Re
were done by Syed et al. (2006) and Flanagan and Syed (2011)" is not enough when the first line
says "we used observational data" and the second says "please read those other papers for their
methods". Now, when we get to the details of the chamber fluxes, there are slightly more details,
but again, not nearly enough. Again, you partition net $CO_2$ using a model, but don't explain
anything beyond that.

*Authors' response:*

*Daily, growing season and annual aggregates of EC NEP includes number of gap-filled*
*net $CO_2$ fluxes. The sole reason of regressing modelled results against gap-filled $CO_2$ fluxes was*
*to examine how much of the deviation between modelled and EC gap-filled estimates of growing*
*season and annual NEP, and between modelled and EC-partitioned GPP and $R_e$ were*
*contributed by the gap-filled fluxes. However, since it created confusion, we have moved any*
*comparison between modelled and gap-filled fluxes to a separate table in supplementary*

*materials (Table S1 in page 43 of supplementary material) and edited the texts accordingly (lines*
*402-412, 445-465, 845-857). Tables 1a and 1b in the revised version of the manuscript now only*
*include tests between modelled and EC-measured (excluding gap-filled fluxes) net ecosystem*
*$CO_2$ fluxes (lines 402-412, 445-465).*

*Methods for screening, gap-filling, and partitioning of EC datasets are now*
*comprehensively discussed (lines 334-343).*

Referee's comment: You average over 9 chambers, but don't say why or what that means? How
much error is introduced here? What is the range of observed fluxes? The reader doesn't know,
so again going back to my main issue, with very limited observational data, ecosys modeled
results look good at the surface, but the work is limited in how much the reader can trust the
results of an over-parameterized and under-tested model.

*Authors' response:*

*Those 9 chambers were in place to cover spatial variation due to peatland micro-*
*topography while measuring the net $CO_2$ fluxes from understorey vegetation and soil. We*
*averaged fluxes from those chambers to include overall hummock-hollow variations into the*
*averaged $CO_2$ fluxes. Chamber flux measurements are now discussed in sufficient details (lines*
*344-351, 413-426). Figures 4 and 5 in the previous version of the manuscript (Figs. 5b and 6j,k*
*in the revised version of the manuscript) are now redone to include standard error of means of*
*spatially averaged chamber fluxes to represent flux variations among the chambers. Moreover, a*
*separate regression test with error analysis between modelled understory and soil $CO_2$ fluxes,*
*and the chamber $CO_2$ fluxes are now included for 2005 and 2006 with contrasting WTD (Table*
*1c) (lines 413-426, 681-687). This test further strengthens the robustness of the modelled*
*outputs.*

Referee's comment: Once we move past the issue of how the observational data is described, we
move through a lot of model descriptions and results that are very, very dense. I'm very happy to
see what looks like the full set of equations that go into ecosys in the appendixes, but the reader
is left with only very dense blocks of text to try to figure out what parts of the model are
important and why. I would suggest keeping the entire set of equations in the appendix, but
moving the main equations used here into the text of the manuscript. Then, the reader doesn't
have to dig out the equations for context and, more importantly, it would be easier to focus the
story around those few equations.

*Authors' response:*

*The model development section describes the key equations that are related to the four*
*operating hypotheses. The respective equations that are listed in the appendices are also cited*
*within the text. The current model description is self-explanatory and a reader does not have to*
*always go back to the appendices to understand the processes. However, the citation of the*

*equation within the text makes sure that a reader can go back to the appendices at any time to*
*see details of a particular equation. Few key equations from the appendices could be pasted into*
*and described in this section as suggested. However, we have felt that it would either make the*
*section even denser or the story could be incomplete if roamed around only a few equations. So,*
*instead of bringing equations into the model development section, we have preferred to include a*
*visualization in the form of a flow chart summarizing those key processes and linking the flow*
*chart with the description as suggested by reviewer 3 (Fig. 1) (lines 204, 207, 209, 215, 216,*
*233, 237, 240, 241, 248, 250, 277, 279, 286, 291, 297, 309, 311, 672, 683, 711, 723, 735, 786,*
*788, 793, 807 and 870).*

Referee's comment: As the reader is starting to get a handle of the main story presented, a major
issue comes up again. There isn't enough data to support the conclusions. Even the highlighted
results in the abstract are heavily focused on things like nitrogen dynamics, nutrient
mineralization, GPP of plant functional types, all of which are 100 percent internal to the model
without any space in the manuscript devoted to why the read should trust the internal model
equations.

*Authors' response:*

*The internal peat biogeochemistry and peatland nutrient cycling modelled in ecosys are*
*tested against leaf nitrogen concentrations, N mineralization, rooting depth, GPP, and $R_e$*
*measured/estimated at either our site or at sites that had similar peat substrates, hydrology*
*and/or plant functional types. Those comparisons are summarized in a table (Table 2) and are*
*described in the texts (e.g., lines 689-694, 737-753 etc.). WTD effects on modelled vascular vs.*
*non-vascular plant water relations and hence productivities were tested against site measured*
*hourly latent heat fluxes, sensible heat fluxes and Bowen ratios, and daily soil moisture contents*
*at different depths. These tests of modelled vascular vs. non-vascular plant water relations which*
*were described in Mezbahuddin et al. (2016) are now briefly cited (806-809). The model*
*equations were derived from independent research which were rigorously tested in other*
*published studies. The sources of each of those equations are listed in the supplementary*
*material. However, sources and the significance of the model equations and the novelty of the*
*modelled process interpretations are now briefly discussed (179-188, 900-903).*

Referee's comment: Finally, the conclusions presented are simply changes in variables that are
internal to the model, without anything to compare them with other than literature values from
other studies. As mentioned in the general comment above, the literature values could be a valid
check if done well, but as this manuscript is currently written, that needs to be done more
formally and not in the discussion. With the suggestion of strengthening the comparison of the
modeled conclusions to literature as well as the authors pinning a lot of the trust in their
conclusions on said literature values, when the ending of the introduction/justification section is
as follows: "Moreover, since hydrological feedbacks to key peatland C processes are highly non-
linear and site-specific, testing of ecosys algorithms across contrasting peatlands would also facilitate formation of a modelling platform for scaling up simulations of those feedbacks across
peatlands at larger spatial scales i.e., national, regional, continental or global as also
recommended by Waddington et al. (2015)" the reader is going to be confused. On one hand,
you compare your conclusions to literature and say "look, these results fit with other studies" but
the entire paper was setup with the story of "there are lots of variations across peatland sites"
throughout the introduction. So, I'm confused and this needs to be cleared up either by heavy
editing of the story.

*Authors' response:*

*The conclusion section is now heavily edited to include general conclusions, and the*
*implications of those conclusions (lines 898-940). A formal succinct comparison between the*
*modelled outputs and measurements from the same site, and similar sites from earlier studies is*
*done in a separate table (Table 2). The mentioned sentence within the quote is now rephrased to*
*remove any confusion and contradiction with earlier description (932-938).*

**Authors' responses to the comments of anonymous Referee #3**

Referee's comment: This manuscript describes simulations of peatland biogeochemistry using the ecosys model, and compares the results to observations from a flux tower, chambers, and water table observations. The model is used to support a detailed analysis of interactions among soil microbial processes, mosses, vascular plants, and hydrology under different water table regimes. The model does a very good job of replicating measured hydrology and carbon cycling at the site, and the analysis produces some very interesting insights about the peatland response to changes in water table. The explanation of how different components of the peatland (vascular plants, non-vascular plants, aerobic decomposition, etc) interact differently under saturated, unsaturated, and deep water table conditions was especially interesting, and I think this has great promise to be a useful framework for future analyses in this field. Overall, I think this was a really nice paper, with novel insights, an impressive model, and clearly written (though dense) presentation.

Referee's comment: My only major suggestion in that a visualization of the interactions at different water table levels might help to summarize the results in a way that's easier for readers to grasp. Sections 2.1.2 and 2.1.3 lay out a lot of competing responses to water table depth (oxygen availability, decomposition, root growth, nitrogen availability, : : :). While the description is pretty clear, I think a visualization would be really helpful. This could be as simple as a table or text box with columns for vascular plants, non-vascular plants, aerobic microbes, etc and rows for different hydrological regimes with the key processes affecting each ecosystem component.

*Authors' response:*

*A flow chart is now added as figure 1 in the revised version of the manuscript to depict the key biogeochemical and physiological effects on plant and microbial functional types (e.g., aerobic vs. anaerobic microbial processes, nitrogen and water uptake etc.) as mediated by different water table depth regimes that are modelled in ecosys. This figure are also adequately linked to the text (lines 204, 207, 209, 215, 216, 233, 237, 240, 241, 248, 250, 277, 279, 286, 291, 297, 309, 311, 672, 683, 711, 723, 735, 786, 788, 793, 807 and 870).*

Referee's comment: Also, there are a lot of interesting mechanistic explanations in the Discussion, but I think there's an opportunity to tie these to the existing literature a little better. Are these new insights about process interactions that have not been discussed in the past? Or are these known interactions that have not been successfully modeled before? A little discussion of the novelty of the process interpretations and general framework of the results versus the novelty of the model itself might help place the results in a better context.

*Authors' response:*

*The mechanistic explanations in the discussion is now summarized in table 2 of the*
*revised version of the manuscript to facilitate the comparison between modelled processes and*
*existing literature. The intention of the mechanistic discussion was to test the adequacy of*
*existing knowledge of our process interpretation to reproduce WTD-C feedbacks in a northern*
*boreal fen which has not been done before. The novelty of the process interpretations vs.*
*modelled processes are now discussed (179-188, 900-903).*

Referee's comment: Specific comments: Lines 17-22: This sentence is really long and hard to
follow. I would suggest rewriting it.

*Authors' response:*

*The sentence is now rephrased (lines 17-22).*

Referee's comment: Lines 38-40: This is an important result, and it might help to also briefly
explain the process behind it (e.g. more drainage eventually leads to limitation of GPP due to
water limitation).

*Authors' response:*

*Done (lines 42-43).*

Referee's comment: Line 46: The units of g/yr don't seem right.

*Authors' response:*

*Corrected (line 49).*

Referee's comment: Lines 316-318: This sentence has some grammatical issues. I suggest
rewriting it.

*Authors' response:*

*The sentence is no longer needed due to rephrasing of the whole paragraph (lines 299-*
*309), and so it has been removed.*

Referee's comment: Lines 642-666: These root responses were very interesting. I don't think
I've seen this process represented in ecosystem models in the past. Lines 682-686: It's very
interesting how vascular plants have an optimum at an intermediate water depth while non-
vascular plants don't. Very interesting implications for changes in relative biomass under
different conditions.

*Authors' response:*

*Yes, it is very interesting and another unique contribution of this study. We have also*
*cited field studies that reported relative domination of vascular over non-vascular species with*
*WTD drawdown in nearby similar peatlands (lines 798-799).*

[revised manuscript text omitted]

current study held moisture content close to saturation and did not drain until $\psi_m$ fell below

0.004 MPa (Mezbahuddin et al., 2016). Consequently, the Campbell type (Campbell, 1974)

power function in *ecosys*, that was used to simulate peat moisture retention scheme in those previous simulations, significantly underestimated peat moisture content in this boreal fen (Mezbahuddin et al., 2016). However, substitution of the existing power function in *ecosys* with a sigmoidal logistic function (van Genuchten, 1980) significantly improved simulation of peat moisture retention over a wide range of moisture conditions in this boreal fen peatland (Mezbahuddin et al., 2016). Since peat moisture retention largely mediates peat oxygenation and hence peatland biogeochemistry, plant water relations, and $CO_2$ fixation, testing of *ecosys*

algorithms with the inclusion of the improved moisture retention scheme against the measurements at the WPL would offer a further test of robustness of process level modelling of hydrological feedbacks to peatland $CO_2$ exchange. Moreover, WPL differed from those other peatlands, where *ecosys* algorithms of eco-hydrological and biogeochemical feedbacks were previously tested, either in climate (e.g., boreal vs. temperate vs. tropical), or hydrology (e.g., fen vs. bog), or peatland vegetation (e.g., non-vascular vs. vascular), or peat forming substrates (e.g., moss vs. sedge vs. woody peats), or depth of peat deposits, or in any combination of these peatland characteristics (Dimitrov et al., 2011; Grant et al., 2012; Mezbahuddin et al., 2014,

2015, 2016). Since these peatland characteristics predominantly govern hydrological regulations to peatland C processes, further testing of *ecosys* algorithms at the WPL would thus test the versatility and robustness of coupled ecology, hydrology and biogeochemistry algorithms in simulating and explaining WTD effects on net ecosystem $CO_2$ exchange across contrasting peatlands. Moreover, since hydrological feedbacks to key peatland C processes are highly non- linear and site-specific, testing of *ecosys* algorithms across contrasting peatlands would also

**1.2. Hypotheses**

[revised manuscript text omitted]

Model performance was evaluated from regression intercepts ($a\rightarrow0$), slopes ($b\rightarrow1$), coefficients of determination ($R^2\rightarrow1$), and root means squares for errors (RMSE$\rightarrow0$) for each study year to test whether there was any systematic divergence between the modelled and EC measured $CO_2$

fluxes.

Similar regressions were performed between Therefore we compared modelled and automated chamber measured net $CO_2$ fluxes for ice free periods (May-October) of 2005 and

2006 to further test the robustness of modelled soil respiration under contrasting WTD

conditions. Each of the half-hourly measured chamber net $CO_2$ fluxes included soil respiration, and fixation and autotrophic respiration from understorey vegetation (e.g., shrubs, herbs and mosses)from understorey PFTs (e.g., shrub and moss). So, we combined with modelled soil respiration with modelled fixation and autotrophic respiration from understorey PFTs for comparison against these chamber measured net $CO_2$ fluxes measured at the WPL. For this purposeWe also averaged net $CO_2$ flux measurements from all of theose 9 chambers were averaged for each half hour to accommodate the variations in those fluxes due to microtopography (e.g., hummock vs. hollow). Two half hourly averaged values of net $CO_2$

fluxes were then averaged again to get hourly mean net chamber $CO_2$ fluxes for comparison against modelled hourly sums of soil and understorey fluxes averaged over modelled hummock and hollow. Model performance was evaluated from regression intercepts ($a\rightarrow0$), slopes ($b\rightarrow1$), coefficients of determination ($R^2\rightarrow1$), and root means squares for errors (RMSE$\rightarrow0$) for each of

2005 and 2006.and compared against average soil and understorey $CO_2$ fluxes modelled over the hummock and the hollow.of modelled vs. gap-filled net $CO_2$ fluxes were also performed to test

[revised manuscript text omitted]

$CO_2$ effluxes in mid-August of 2006 contributed to the larger modelled ecosystem $CO_2$ effluxes ($R_e$) as apparent in larger modelled night-time fluxes in the late growing season of 2006 than in that of 2005 whichthat was well corroborated also apparent in by night-time EC $CO_2$ fluxes in

2006 which were larger than those in 2005 during those periods (Fig. 54a). The stimulation of $R_e$

as a result of WTD drawdown was further corroborated by larger sums of night-time soil $CO_2$

fluxes and understorey autotrophic respiration ($R_a$) as measured by Cai et al. (2010) using automated chambers and modelled by *ecosys* in late growing season of 2006 with deeper WTD

than in that of 2005 with shallower WTD (Fig. 4b).

Further Continued WTD drawdown into the late growing season of 2008 (Fig. 43c)

caused sustained improved peat oxygenation and hence larger modelled soil $CO_2$ effluxes in the model (Fig. 54c). It contributed to similarly largerConsequently, modelled night-time net ecosystem $CO_2$ fluxes, and soil and understorey $CO_2$ fluxes in the late growing seasons of 2006

and in 2008 were similarly larger than those in 2005 that which was were also well corroborated well by EC measured night-time fluxes during those periods2006 and 2008 vs. 2005 (Figs. 54a- b). Although Consequently, the sums of modelled night-time soil $CO_2$ fluxes and understorey $R_a$

in late growing season of 2008 were similarly larger as in 2006 with respect to those in that period of 2005 (Fig. 4b). We did not have any chamber measurements available for 2008 to corroborate the modelled outputs. Despite larger night-time modelled and EC measured net ecosystem CO$_2$ fluxes in  2008 were larger than those in 2005, the day- time modelled and EC measured  CO$_2$ fluxes in 2008

did not decline with respect to those in 2005 (Fig. 54a). Similar day-time fluxes in 2005 and

2008 despite larger night-time fluxes in 2008 than in 2005 indicated a greater late growing season CO$_2$ fixation in 2008 with deeper  WTD than in 2005

with shallower WTD.

Beside WTD, temperature variation  also profoundly affect ecosystem net CO$_2$ exchange at the WPL. For a given WTD condition warmer weather caused increases in

$R_e$ at the WPL (Figs. 34b-c and 54a-b). Night-time modelled, and EC  and chamber measured ecosystem, soil, and understorey CO$_2$ fluxes in warmer nights of day 214,

[revised manuscript text omitted]

GPP from 2008 to 2009 was less smaller than that the reduction in the EC-derivedmodelled $R_e$

GPP therebythat causing yielded an increase in modelled growing season NEP from 2008 to

2009 (Figs. 76a-c).

Despite the counteracting and offsetting effects of WTD and $T_a$ on GPP and $R_e$, larger mModelled and EC-derived estimates of growing season GPP and $R_e$ in 2009 were larger than those in 2004 with despite similar mean $T_a$ in those years (Figs. 7a-d). It suggested that both modelled and EC-derived growing season GPP and $R_e$ increasedincreases in growing season

GPP and $R_e$ with the deepening of average growing season WT from 2004 to 2009 was a net effect of the deepening of average growing season WT at the WPL (Figs. 76a-d). It was further corroborated by polynomial regressions of modelled growing season estimates of GPP and $R_e$ on against modelled average growing season WTD, and similar regressions of EC-derived growing season GPP and $R_e$ on against site measured average growing season WTD (Figs. 87a-c). These relationships showed that there were increases in modelled and EC-derived growing season GPP

and $R_e$ with deepening of the growing season WT from 2004 to 2008 after which further WTD

drawdown in 2009 started to cause slight declines in both GPP and $R_e$ (Figs. 87b-c). Neither modelled nor EC-gap filled estimates of growing season NEP yielded significant regressions when regressed on against modelled and measured growing season WTD respectively (Fig. 87a).

It indicated that similar increases in modelled and EC-derived growing season estimates of GPP

and $R_e$ with deepening of WT along with some counteracting effects of $T_a$ left no net effects of

WTD drawdown on either modelled or EC-derived growing season NEP (Figs. 76 and 87a).

WTD effects on growing season GPP and $R_e$ and hence NEP from 2004 to 2009 as measured at the WPL and modelled by *ecosys* were also consistent at an annual time scale from

2004 to 2008 (Figs. 6e-h). Similar to the growing season trend, drawdown of both measured and modelled WTD averaged over the ice free periods (May-October) from 2004 to 2008 generally stimulated annual modelled and EC-derived GPP and $R_e$ (Figs. 76f, g, h and 87e, f). The deepening of WT also raised modelled and EC-derived annual $R_e$ from 2005 to 2008 as was in the case of growing season $R_e$ (Figs. 6g-h and 7f). Similar increases in both modelled and EC- derived annual GPP and $R_e$ with WTD drawdown left no net WTD effects on modelled and EC- gap filled annual NEP (Figs. 7e, 87d). We did not include the year 2009 while examining the effects of WTD drawdown on interannual variation of GPP, $R_e$ and NEP due to the lack of EC

$CO_2$ flux measurements from September to December in 2009 (Figs. 6e-h and 7d-f). Even

Although, modelled WTD effects on GPP, $R_e$ and hence NEP were corroborated well by EC- derived GPP, $R_e$ and EC-gap filled NEP, the modelled growing season and annual GPP and $R_e$

were consistently higher than the EC-derived estimates of those from throughout 2004 to

2009the study period (Figs. 67-87).

Increased GPP with WTD drawdown (Figs. 67b, f and 87b, e) was modelled by ecosys predominantly through increased root growth and uptake of nutrients and consequently improved leaf nutrient status and hence more rapid $CO_2$ fixation in vascular PFTs. Under shallow WT

during the growing season of 2004, roots in modelled black spruce and tamarack PFTs hardly grew below 0.35 m from the hummock surface (black spruce was not planted in the hollow) and the roots of modelled tamarack PFT were mostly confined within 0.35 m below the hummock surface and 0.05 m below the hollow surface. Modelled root densities of both black spruce and tamarack were higher by 2-3 orders of magnitude in the top 0.19 m of the hummock (data not shown). A WTD drawdown from by ~0.05-4 m above the hollow surface (~0.25 m below the hummock surface) in the growing season of 2004 to ~0.35 m below the hollow surface (~0.65 m below the hummock surface) in the growing season of 2009from the growing season of 2004 to that of 2009 caused  increase in maximum modelled rooting depth

in both PFTs (Table 2). Increased root growth in modelled vascular PFTs augmented root surface area for nutrient uptake under deeper WT in the growing season of 2009 than in 2004. Increased root surface area along with increased nutrient availability due to more rapid mineralization with improved aeration as a result of WTD drawdown from 2004 to 2009 caused improved root nutrient uptake in modelled vascular PFTs. Increased root growth, nutrient availability and hence uptake due to WTD

drawdown from the growing season of 2004 to that of 2009  caused an increase in modelled foliar N concentrations in black spruce, tamarack and dwarf birch PFTs driving the increases in GPP

modelled over this period (Figs. 6b, f and 7b, e) (Table 2).

.

**3.5. Simulated drainage effects on WTD and NEP**

Artificial drainage can drastically alter  WTD in a peatland that can cause dramatic changes in peatland NEP by shifting the balance between GPP and $R_e$.

Projected  growing season WT was deeper by by ~0.5 m and ~0.55 m respectively from those in the real-time simulation in drainage cycles 1 and 2 in all the years from 2004 to 2009 (Fig. 98a).

The projected drawdown of growing season WTD in *ecosys* caused changes in modelled growing season NEP, GPP and $R_e$. 
[revised manuscript text omitted]

 . Mass-based modelled and measured foliar N to P ratio in all the PFTs were less than 16:1 indicating that the  vegetation at the WPL  wasmore N  limited (Aerts and Chapin III, 1999). Since the modelled PFTs were predominantly N limited, increases in foliar N concentrations as a result of improved root nutrient availability, growth and nutrient uptake with WTD drawdown enhanced modelled carboxylation rates and hence modelled GPP. In a similar fen peatland close to our study site, Choi et al. (2007)  found an increase in peat $NO_3^-$ -N due to enhanced mineralization and nitrification stimulated by a WTD drawdown  ~~surface that caused increases in foliar N concentrations from ~21 to ~27 g kg$^{-1}$ C (assuming 50%of dry matter as organic C) in black spruce and ~41 to ~66 g kg$^{-1}$C in tamarack in a CentralAlbertan fen peatlandIncreases in foliar N concentrations due to enhanced rootnutrient availability and uptake with WTD drawdown in their study also caused significantlygreater radial tree growth.in a WTD manipulation studythat WTD drawdown by ~0.45 m raisedfrom ~19 to ~21g kg$^{-1}$ C (assuming 50% of dry matter as organic C)~36 to ~42 g kg$^{-1}$C inin similar peatlandsSect. 3.4~~
[revised manuscript text omitted]

Despite larger modelled vs. EC-derived GPP and $R_e$ were larger than the EC-derived estimates., modelled annual NEP wereas consistently lower than the EC gap-filled annual NEP (Fig. 67e).

MSmaller mModelled annual NEP were smaller than EC-derived estimates because smaller than

EC-derived estimates were caused bydue to larger differences in larger modelled andmodelled $R_e$

were larger than EC-derived vs.estimates$R_e$ thanEC-derived $R_e$ than differences beweenby margins bigger- than by what modelled GPP were larger than vs.and EC-derived GPP estimates (Figs. 76f-g). Larger deviation between mModelled $R_e$ and were larger than EC-derived $R_e$

estimates was mainly contributed bydue to the presence of gap filled night-time $CO_2$ fluxes (=$R_e$)

in EC-derived estimates which were smaller than corresponding modelled values. which It was apparent in negative intercepts that resulted from regressions of modelled vs. gap-filled net CO$_2$ fluxes (Table S1 in supplementary material). The gap-filled $R_e$ fluxes were calculated from soil temperature ($T_s$) at a shallow depth (0.05 m) (Sec. 2.2.2). During night-time and in the winter, peat at this shallow depth rapidly cooled down and yielded smaller night-time gap-filled CO$_2$ fluxes (Figs. 3, 5 and 6). On the contrary, corresponding modelled CO$_2$ effluxes were affected by the temperatures of not only the shallow peat layers but also the deeper peat profile that were warmer than the shallower layers and thus were larger than the gap-filled fluxes (e.g., Figs. 5a, 6h-i). Like modelled CO$_2$ effluxes, chamber measured CO$_2$ effluxes in cooler nights also did not decline as rapidly as did the corresponding gap-filled CO$_2$ fluxes as night progressed which further indicated the likely contribution of gap-filling artifact to CO$_2$ effluxes that were smaller than corresponding modelled fluxes (e.g., Figs. 5b vs. 4a, 6j-k vs. 5h-i).

Systematic uncertainties embedded in EC methodology could also have contributed to EC-derived annual and growing season $R_e$ estimates which were smaller than the modelled values (Figs. 7c, g). The major uncertainty in the EC methodology is the possible underestimation of nighttime EC CO$_2$ flux measurements due to poor turbulent mixing under stable air conditions (Goulden et al., 1997; Miller et al., 2004). On the contrary, modelled biological production of CO$_2$ by plant and microbial respiration was independent of turbulent mixing which would thus contribute to  modelled $R_e$ that were larger than EC-derived $R_e$

estimates.

 modelled vs. gap-filled $R_e$  might have also contributed to larger modelled  vs. gap-larger than EC-derived growing season and annual GPP (Figs. 76b, f). as derived from $R_e$.  
[revised manuscript text omitted]
 *ecosys* simulated increases in $R_e$ and GPP with deepening of WT from 2004 to 2009 in the boreal fen at the WPL. Similar increases in $R_e$ and GPP, however, left no net effect of WT deepening on modelled variations in NEP. This modelled trend was corroborated by EC-derived NEP, $R_e$ and GPP (Syed et al., 2006; Flanagan and Syed, 2011) and automated chamber measured NEP and $R_e$ (Cai et al., 2010) at the WPL. The effects of WTD drawdown on $R_e$ and GPP was modelled in *ecosys* by the algorithms representing following processes:

Improved [$O_{2s}$] facilitated by rapid $O_2$ diffusion (Eqs. D42-D44) under deeper WT raised microbial energy yields while oxidizing DOC coupled with $O_2$ reduction (Eq. A21) and hence caused increases in $R_e$. Increased mineralization rates of DON and DOP due to improved [$O_{2s}$] also increased aqueous concentrations of $NH_4^+$, $NO_3^-$ and $H_2PO_4^-$ (Eqs. C23a, c, e) that facilitated microbial nutrient availability, uptake (Eq. A22) and growth (Eq. A29) and hence further enhanced $R_e$.

Increased nutrient availability due to rapid mineralization with WTD drawdown as mentioned above hastened root nutrient (mainly N) availability and uptake (Eqs. C23b, d, f). Root nutrient availability and uptake in *ecosys* were further facilitated by increased root growth stimulated by improved [$O_{2s}$] under deeper WT. Greater root growth and uptake thus caused improved foliar $\sigma_N$ with respect to $\sigma_C$ thereby enhancing $CO_2$ fixation (Eq. C6) and vascular GPP (Eq. C1).

When WT in *ecosys* dropped below -0.3 m from the hollow surface (-0.6 m below the hummock surface), inadequate capillary recharge from WT caused desiccation of near surface peat layers and surface residues (Eqs. D9, D12). Surface and near-surface peat desiccation raised microbial concentrations and reduced microbial access to substrate for decomposition in these desiccated layers (Eq. A15). It enabled *ecosys* to simulate reduction in $R_h$ in the desiccated peats from competitive inhibition of microbial exo-enzymes with increasing concentrations (Figs. 6c, 8d) (Eq. A4).

When WT fell below ~ 0.1 m from the hollow surface (~ 0.4 m below the hummock surface) vertical recharge of near surface peat layers through capillary rise from WT was not enough to sustain moss water uptake thereby causing moss drying and consequent reduced moss GPP (Eqs. C1, C4). However, sustained increases in vascular GPP due to root water uptake from deeper wetter layers more than fully offset the suppression of moss GPP thereby causing a net increase in GPP with WTD drawdown (Figs. 6b and 9b-c).

These modelling hypotheses were also corroborated by various field, laboratory and modelling studies over similar peatlands (Sect. 4.1). Moreover, the projected drainage simulation showed that the increase in vascular GPP due to improved plant nutrient status caused by WTD drawdown would only sustain while WTD remained above a threshold level i.e. ~ 0.6 m below the hollow surface (~ 0.9 m below the hummock surface). When WT fell below this threshold, projected vascular GPP started to decrease with further WTD drawdown thereby causing reductions in ecosystem GPP. Similar WTD threshold effects on vascular GPP were also found in other studies in similar peatlands (Sect. 4.1.2).

Therefore, representing feedbacks between peatland hydrological and C processes in *ecosys* enabled successful simulation of WTD effects on $R_e$, GPP and hence NEP of a northern boreal fen peatland. Projected drainage simulation in our study showed that continued WTD could alter ecosystem C balance of northern boreal peatlands by decreasing GPP and sustaining increased $R_e$ thereby causing declines in NEP. These findings provide us with important insights into how northern boreal peatland C stocks would be affected by likely WTD drawdown under future drier and warmer climates. This study is also reproducible in other peatlands when the model is fed by required physical, hydrological, chemical, biological and ecological inputs those are measurable at the sites (Fig. 1) (Mezbahuddin et al., 2016). Successful simulation of hydrological effects on peatland C processes by *ecosys* for a northern boreal fen peatland in this study along with simulations of feedbacks between hydrology, biogeochemistry and ecology by the same model *ecosys* across other contrasting peatlands (e.g., Dimitrov et al., 2011; Grant et al., 2012; Mezbahuddin et al., 2014, 2015, 2016) would, therefore, provide us with a modelling framework for large scale (e.g., regional/continental/global) peatland C simulations, which currently is one of the most sought after in global terrestrial ecosystem carbon modelling community.

**Code availability**

The *ecosys* model codes are listed in equation forms and sufficiently described in the supplementary material. The model codes that were written in FORTRAN will also be available on request from either symon.mezbahuddin@gov.ab.ca or rgrant@ualberta.ca.

**Data availability**

  Field data that were used to validate model outputs are available at http://fluxnet.ornl.gov/site/292.

**Author contribution**

M. Mezbahuddin contributed to the model code modification and development, designing modelling experiment, simulation, validation, and analyses of modelled outputs. R. F. Grant is the original developer of the model *ecosys* and also contributed into simulation design and model runs. L. B. Flanagan was site principal investigator who led the collection, and quality control of the field data that were used to validate model outputs. M. Mezbahuddin wrote the manuscript with significant contributions from R. F. Grant and L. B. Flanagan.

**Acknowledgments**

Computing facilities for the modelling project was provided by Compute Canada,

Westgrid, and University of Alberta. Funding for the modelling project was provided by several research awards from Faculty of Graduate Studies and Research and Department of Renewable

Resources of University of Alberta and a Natural Sciences and Engineering Research Council (NSERC) of Canada discovery grant. The field research was carried out as part of the Fluxnet-

Canada Research Network and the Canadian Carbon Program and was funded by grants to

Lawrence B. Flanagan from NSERC, Canadian Foundation for Climate and Atmospheric

Sciences, and BIOCAP Canada.

[revised manuscript text omitted]

**Fig. 2̶3.**

[Figure]

**Fig. 43.**

[Figure]

[Figure]

**Fig. 54.**

[Figure]

[Figure]

Field Code Changed

**Fig. 65.**

[Figure]

**Fig. 76.**

[Figure]

**Fig. 8̶7̶.**

[Figure]

**Fig. 98.**

[Figure]

**Fig. 109.**

Table 1: Statistics from regressions between modelled and EC-gap filled net ecosystem $CO_2$ fluxes throughout the years of 2004-2008 at a Western Canadian fen peatland

| Year | Total annual precipitation (mm) | $n$ | $a$ | $b$ | $R^2$ | RMSE ($\mu$mol $m^{-2}s^{-1}$) | RMSRE ($\mu$mol $m^{-2}s^{-1}$) |
|---|---|---|---|---|---|---|---|
| Modelled vs. eddy covariance $CO_2$ fluxes measured at $u^* > 0.15$ ms$^{-1}$ | | | | | | | |
| 2004 | 553 | 5034 | 0.08 | 1.10 | 0.81 | 1.58 | 1.92 |
| 2005 | 387 | 5953 | 0.07 | 1.03 | 0.82 | 1.68 | 1.99 |
| 2006 | 465 | 6012 | 0.07 | 1.08 | 0.79 | 1.68 | 1.98 |
| 2007 | 431 | 5385 | 0.06 | 0.99 | 0.79 | 1.83 | 2.09 |
| 2008 | 494 | 5843 | -0.01 | 0.98 | 0.84 | 1.63 | 2.02 |
| Modelled vs. gap-filled $CO_2$ fluxes | | | | | | | |
| 2004 | 553 | 3750 | -0.13 | 1.20 | 0.89 | 0.64 | |
| 2005 | 387 | 2807 | -0.49 | 1.03 | 0.76 | 0.82 | |
| 2006 | 465 | 2748 | -0.48 | 1.15 | 0.81 | 0.58 | |
| 2007 | 431 | 3375 | -0.36 | 0.97 | 0.74 | 1.23 | |
| 2008 | 494 | 2941 | -0.54 | 1.05 | 0.79 | 0.95 | |

(*a, b*) from simple linear regressions of modelled on measured. $R^2$ = coefficient of determination and RMSE = root mean square for errors from simple linear regressions of measured on simulated. RMSRE= root mean square for random errors in eddy covariance (EC) measurements calculated by inputting EC $CO_2$ fluxes recorded at $u^*$ (friction velocity) > 0.15 m s$^{-1}$ into algorithms for estimation of random errors due to EC $CO_2$ measurements developed for forests by Richardson et al. (2006).

**Table 2:** Statistics from regressions between modelled and EC-gap filled net ecosystem $CO_2$ fluxes during the growing seasons of 2004-2009 at a Western Canadian fen peatland

[revised manuscript text omitted]

---

## Author Response (AR2)

**Authors' responses to comments of reviewer # 2**

Referee's comment:

General Comments

I would like to acknowledge the work that the authors put into this updated manuscript version.

The addition of the new figure one combined with the edits made to the text creates a manuscript that is an improved read throughout and the work put into the this updated version address the majority of my issues from the first version. My main issue of how dense the text of the first manuscript version I feel is resolved.

The second main issue I had with lack of data is nearly resolved and I greatly appreciate the additional information in Table Two. This is a very critical bit of information in evaluating the results presented in this paper. While I'm happy to see this information, I do think there needs to be some explaining in the methods section as to where the data came from, particularly the data collected at the field site. For the data from other field sites, a simple justification of similarity I

think is called for.

*Authors' response:*

*A brief discussion on site measurements for foliar nutrient contents and on other data measured*

*at similar sites which were used for corroborating modelled results in Table 2 are now included*

*(lines 353-363 in the revised manuscript).*

Referee's comment:

A comparison of the site's climate data would be great. Right now, the information in Table Two just comes up.

*Authors' response:*

*Since Table 2 shows the effects of WTD drawdown on carbon and nutrient cycling, we have now*

*included a comparison of both modelled and measured growing season WTD between 2004 and*

*2009 at the beginning. This gives the readers a clear picture of how gradually drier weather*

*from 2004 to 2009 caused a growing season water table drawdown at the very beginning of the*

*Table 2 and hence facilitates a smoother progression to what follows next. We, however, did not*

*include comparison of weather variables such as temperature, precipitation etc. in Table 2 since*

*those are already presented in Table 1 and in Figures 7d, and 7h.*

Referee's comment:

In both Tables One and Two, I would like to see confidence intervals (or similar) of the regression statistics (Table One) and modeled/measured parameters (Table Two). Using approximations or ~ signs is enough here.

*Authors' response:*

*Standard errors are now added to regression statistics in Table 1 and other parameters*

*(wherever applicable) in Table 2 instead of approximation signs (~).*

Referee's comment:

In the Author's Response you say "The internal peat biogeochemistry and peatland nutrient cycling modelled in ecosys are tested against leaf nitrogen concentrations, N mineralization, rooting depth, GPP, and Re measured/estimated at either our site or at sites that had similar peat substrates, hydrology and/or plant functional types." I don't see evidence of statistics showing they were tested, only compared.

*Authors' response:*

*The comparisons between the modelled and measured (either at the site or at similar sites)*

*parameters in Table 2 were basically one to one bases and hence no statistical test was*

*performed on any of those comparisons. These comparisons facilitated further corroboration of*

*the modelled eco-hydrology and biogeochemistry processes that were rigorously tested against*

*hourly fluxes in Table 1 and also compared against daily, seasonal and annual estimates of EC-*

*derived NEP, GPP and $R_e$ in Figures 3 through 8. The confusion is now clarified in lines 353-*

*356 of the revised manuscript.*

[revised manuscript text omitted]